# Budget-Constrained Auctions with Unassured Priors: Strategic Equivalence and Structural Properties

Submission Id: 175

## ABSTRACT

In today's online advertising markets, it is common for advertisers to set long-term budgets. Correspondingly, advertising platforms adopt budget control methods to ensure that advertisers' payments lie within their budgets. Most budget control methods rely on the value distributions of advertisers. However, due to the complex advertising landscape and potential privacy concerns, the platform hardly learns advertisers' true priors. Thus, it is crucial to understand how budget control auction mechanisms perform under unassured priors.

This work answers this problem from multiple aspects. Specifically, we examine five budget-constrained parameterized mechanisms: bid-discount/pacing first-price/second-price auctions and the Bayesian revenue-optimal auction. We consider the unassured prior game among the seller and all buyers induced by these five mechanisms in the stochastic model. We restrict the parameterized mechanisms to satisfy the budget-extracting condition, which maximizes the seller's revenue by extracting buyers' budgets as effectively as possible. Our main result shows that the Bayesian revenue-optimal mechanism and the budget-extracting bid-discount first-price mechanism yield the same set of Nash equilibrium outcomes in the unassured prior game. This implies that simple mechanisms can be as robust as the optimal mechanism under unassured priors in the budget-constrained setting. In the symmetric case, we further show that all these five (budget-extracting) mechanisms share the same set of possible outcomes. We further dig into the structural properties of these mechanisms. We characterize sufficient and necessary conditions on the budget-extracting parameter tuple for bid-discount/pacing first-price auctions. Meanwhile, when buyers do not take strategic behaviors, we exploit the dominance relationships of these mechanisms by revealing their intrinsic structures. In summary, our results establish vast connections among budget-constrained auctions with unassured priors and explore their structural properties, particularly highlighting the advantages of first-price mechanisms.

## CCS CONCEPTS

• **Theory of computation** → **Computational pricing and auctions**; *Market equilibria*; *Algorithmic game theory*.

## KEYWORDS

Budget-Constrained Auctions, Unassured Priors, Strategic Equivalence, Structural Properties

**ACM Reference Format:**

Anonymous Author(s). 2023. Budget-Constrained Auctions with Unassured Priors: Strategic Equivalence and Structural Properties. In *Proceedings of Make sure to enter the correct conference title from your rights confirmation email (Conference acronym 'XX).* ACM, New York, NY, USA, 29 pages. https://doi.org/XXXXXXX.XXXXXXX

## 1 INTRODUCTION

We have witnessed substantial growth in the online advertising market in recent years. Billions of advertising positions are sold every day on various kinds of platforms, including major search engines (e.g., Google [28]) and social media (e.g., Meta [33]). According to statistics, the volume of the global online advertising market is hopeful of reaching 626 billion dollars in 2023 [38]. From a macro perspective, the contents of ads exhibit immense heterogeneity according to different types of ad queries. For example, a new parent is more likely to receive ads promoting baby products, while an older individual may be targeted with advertisements for hearing aids.

To address such heterogeneity, advertising platforms employ auctions to allocate ad spaces. Each advertiser submits a bid she wants to pay for each ad query satisfying certain conditions (e.g., the ad query is from a new parent or an older individual). When a real-time query is received, the platform conducts an auction among all advertisers who have proposed positive bids on the query. As such a process occurs at a significant scale every day, an advertiser's payment can vary drastically. Consequently, major platforms now request advertisers to provide a long-term budget (e.g., for a day, a week, or a month) to mitigate this uncertainty. Correspondingly, the platform's auction mechanisms ensure that each advertiser's payment does not exceed her budget. Such an approach can help advertisers control advertising costs and make long-term plans.

Many works have studied different budget control methods, from either a dynamic view [10, 19, 24] or an equilibrium view [3, 8, 11, 13, 14]. One crucial assumption adopted in these works is that the platform knows the prior value distributions or even the actual values of advertisers. Nevertheless, such an assumption can be unattainable in practice. From an information accessing standpoint, the platform can only obtain an advertiser's *historical bids* rather than her historical values. Consequently, the platform lacks information on her values or priors. Furthermore, the classic methodology of incentive compatibility (IC) embraced by existing works hardly fits with today's advertisers due to two main reasons: (1) The traditional definition of IC does not capture the various constraints faced by advertisers, including budget constraint [3] and

return-on-investment (ROI) constraint [7]. Therefore, we must carefully refine the concept to accommodate more complex circumstances, and such trials always lead to intricate outcomes [6]. The concept becomes even more inadequate when considering that advertisers often cooperate simultaneously with multiple platforms [2, 18]. (2) Advertisers have inherent incentives to hide their true values to cope with the learning behavior of the platform and protect their data privacy. Once the platform has complete knowledge of an advertiser's actual value distribution, price discrimination would inevitably occur, which could be a curse for the advertiser.

With the emergence of the above two phenomena, market designers must face the fact that they may never be able to get advertisers' true values/value priors. Thus, an important problem naturally arises:

*How do unassured priors affect budget control methods in auctions? Specifically, when priors are unassured, how are budget control methods related?*

This paper answers the above problem comprehensively. We study a range of five kinds of budget-constrained auctions, respectively Bayesian revenue-optimal auction (BROA) [3], as well as bid-discount/pacing first-price/second-price auctions (BDFPA, PFPA, BDSPA, PSPA) (See Table 1). In these auction forms, the seller adopts diverse methods to help buyers control the expenditure within their budgets. It is worth noting that pacing is one of the most extensively studied strategies for controlling advertisers' payments [11, 13, 14]. Moreover, bid-discount is a strategy that has been adopted in sponsored search auctions [1, 20, 31] and second-price auctions [25, 35]. While it is natural to incorporate such a strategy into first-price auctions, to the best of our knowledge, this combination has not been explored in previous literature. Meanwhile, the power of bid-discount as a means of budget management remains largely unexplored. We comprehensively compare these five mechanisms from a game-theoretic view and study the structural properties of these mechanisms, particularly focusing on variants of first-price auctions.

## 1.1 Main Contributions

This work presents three main contributions.

*Strategic equivalence among budget-constrained auctions in the unassured prior game.* We examine an *unassured prior game with budget constraints* (abbreviated as unassured prior game) among the seller and buyers within a stochastic setting [3, 5, 14, 29]. Technically, this game is an extension of the private data manipulation (PDM) model [15, 39] to the budget-constrained scenario. In our unassured prior game, the seller first commits to a parameterized auction mechanism, after which buyers report their *bid distributions* and real budgets while keeping their value distributions private. At last, a parameter tuple is calculated based on a predefined rule that considers buyers' bid distributions and budgets, ensuring that each buyer's budget is not exceeded in expectation. This model captures the scenario in budget-constrained auctions where the seller can only access buyers' historical bids rather than their true values.

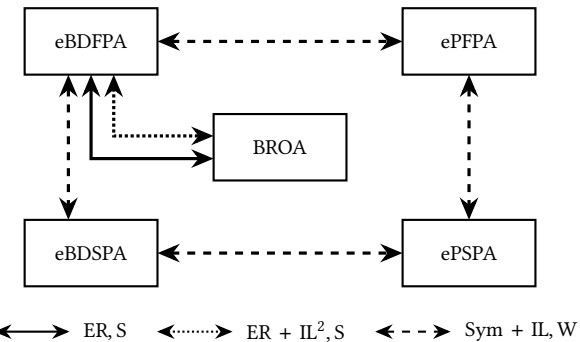

Figure 1: Summary of the results in Section 4 on the strategic equivalence among different auction types. Two auction forms are strategic-equivalent if they are connected by a bidirectional arrow. Different line types indicate the restrictions. ER: Each buyer's virtual bidding quantile function is strictly increasing and differentiable. IL: Each buyer's bidding quantile function is inverse Lipschitz continuous. $IL^2$: Each buyer's bidding quantile function and virtual bidding quantile function are both inverse Lipschitz continuous. Sym: Buyers and budget-extracting parameters are both symmetric. Strong (S) and weak (W) strategic equivalence are defined in Definition 4.1. The "e" at the front of mechanisms stands for budget-extracting, which is defined in Definition 3.1.

Within the unassured prior game, for variants of first-price/second-price auctions, we introduce the concept of *budget-extracting*, which guarantees that under the budget-extracting parameter tuple, the platform adequately consumes each advertiser's budget without violating the individual rationality (IR) constraint. This concept is similar to the notion of system equilibrium defined in Balseiro et al. [3]. We show that under minor assumptions, for bid-discount/pacing first-price auctions and symmetric pacing second-price auction, the budget-extracting condition leads to the seller's revenue maximization (Theorem 3.1). Thus, we restrict the seller's parameter choice to budget-extracting ones, which are generally dominating.

With these game-theoretic preparations, we prove that the budget-extracting bid-discount first-price auction is *strongly strategic-equivalent* to the Bayesian revenue-optimal auction (Theorem 4.2) under minor restrictions. In simpler terms, this is to say that there is a mapping from a buyer's strategy in the Bayesian revenue-optimal auction to a strategy in the budget-extracting bid-discount first-price auction, such that the outcome profile is kept when the mapping acts on each buyer's strategy. Vice versa, from the budget-extracting bid-discount first-price auction to the Bayesian revenue-optimal auction. Combined with a reduction in buyers' strategies in the Bayesian revenue-optimal auction, we show that these two auction formats yield the same set of Nash equilibrium outcomes (Theorem 4.3). This theorem can be interpreted as a simple-versus-optimal result in budget-constrained mechanisms, showing that simple mechanisms (budget-extracting bid-discount first-price auction) can be as robust as the optimal auction facing uncertain priors. Further,

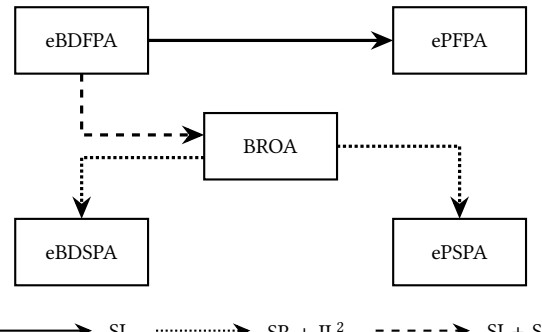

Figure 2: Summary of the results in Theorem A.4 on the dominance relationships among different auction types when buyers truthful bid. A dominates B if there is an arrow from A to B. Different line types indicate the assumptions. SI: Each buyer's bidding quantile function is strictly increasing. SR: Each buyer's virtual bidding quantile function is strictly increasing. $IL^2$: Each buyer's bidding quantile function and virtual bidding quantile function are both inverse Lipschitz continuous.

in the symmetric case, we establish a broad *weak strategic equivalence* result among first-price/second-price auctions (Theorem 4.4). In short, this result indicates that these auctions have the same set of possible symmetric outcomes when buyers are symmetric (Corollary 4.5). We summarize these results in Figure 1.

*Properties on variants of first-price auctions.* We delve into the properties of bid-discount and pacing first-price auctions. In particular, we study the sufficient and necessary conditions of budget-extracting. As revealed in Theorems A.1 and A.3, with minor restrictions, there exists a maximum budget-feasible parameter tuple for bid-discount/pacing first-price auctions, and is budget-extracting. Interestingly, for the pacing first-price auction, the budget-extracting parameter tuple is unique, while this is not necessarily the case for bid-discount first-price auction. Subsequently, we further exploit the behavior of budget-extracting tuples in the latter and derive an equivalent condition for the uniqueness of the budget-extracting tuple. Meanwhile, we show in Theorem A.2 that it is computationally efficient to derive a budget-extracting tuple for bid-discount first-price auction.

*Dominance relationships on the seller's revenue without strategic bidding.* At last, we suppose that buyers do not take strategic behaviors to exploit the intrinsic properties of auction mechanisms further. Specifically, we compare the seller's revenue in these mechanisms under the budget-extracting condition. For this part, we prove that under weak assumptions, bid-discount first-price auction dominates Bayesian revenue-optimal auction and pacing first-price auction. Meanwhile, Bayesian revenue-optimal auction outperforms two variants of second-price auctions. These results are illustrated in Figure 2.

## 1.2 Related Work

*Prior manipulation model.* In classic solutions, e.g., the seminal work of Myerson [34], a critical assumption is that the seller knows the distribution of buyers' values. In real life, however, from a buyer's view, when she takes some strategic behavior other than truthful bidding (e.g., when she wants to protect her real data), the seller can never get the true distribution. A line of work captures such inconsistency between the ideal and real worlds and focuses on how the auction market is affected when the seller wrongly estimates the buyers' value distributions. Tang and Zeng [39] studies the problem in general, and a surprising result is that Myerson auction, which is well-known for being revenue-optimal, is revenue-equivalent to first-price auction under such a model. Concurrently, Deng et al. [16], Deng and Zhu [17] consider specific distribution families from a statistical optimization view in this setting. Deng et al. [15] further studies the scenario in sponsored search auctions and shows the general equivalence of different auction types under such a setting. Our paper follows this research line by modeling the above intuition as the unassured prior game. Nevertheless, compared to these prior works, our work considers buyers' budgets and explores deep relationships among different auction forms.

*Market equilibrium with budget-constrained buyers.* In real life, it is always the case that a buyer's affordability is small compared with the massive amount of auctions happening every day. Therefore, it is reasonable for a buyer to set a budget constraint. Much research considers the market equilibrium in this scenario.

The works most related to ours are Balseiro et al. [3, 4], Feng et al. [21]. Balseiro et al. [3] surveys on various budget control methods in second-price auctions and compares these methods from the aspects of seller's revenue and social welfare in equilibrium. Nevertheless, our work is not limited to second-price auctions. We also consider the optimal auction and variants of first-price auction, and further build the connection among these three genres of auctions when buyers are budget-constrained with unassured priors. Balseiro et al. [4] focuses on the contextual scenario in standard auctions where a buyer's value is decided collaboratively by a public item type and a private buyer type. The paper shows a revenue equivalence result across all standard auctions under symmetric Bayes-Nash equilibrium, which seems similar to one of our results. However, a crucial difference is that our paper considers the setting with prior manipulation and without any contextual information. Furthermore, our result is not limited to symmetric cases, which implies that our strategic equivalence theorems differ from classical revenue equivalence results. At last, a recent work [21] focuses on the scenario when buyers can misreport the budget constraint and the maximum bid in the pacing first-price equilibrium (PFPA) [13]. In comparison, our work studies five mechanisms, including the pacing first-price auction. Meanwhile, a buyer's strategy in our work is her bid distribution, rather than her budget and the maximum bid.

The bid-discount method has been adopted to generalized second-price auctions for sponsored search in early years [1, 20, 31], with the multipliers closely related to the click-through rates. In recent years, such a method was applied to second-price auctions [25, 35], known as boosted second-price auction (BDSPA in our work). Experimental results show that such an auction form earns the seller

more revenue than the second-price and empirical Myerson auction without budget constraints. On the other hand, our work considers the scenario when buyers are budget-constrained and introduce the bid-discount method into first-price auctions.

Pacing (a.k.a. bid-shading) is perhaps the most well-studied budget control method of all. In pacing, the seller would assign a multiplier to each buyer, and the multiplier would shade the bid of any buyer before being sorted. The payment of a buyer is also correspondingly scaled to control the budget. Chen et al. [11], Conitzer et al. [13, 14] respectively considers the pacing equilibrium in first-price and second-price auctions. However, these papers focus on a discrete setting where buyers' values are known *a priori*. Instead, our work focuses on the stochastic setting where the value of each buyer is unsure *ex-ante*. Some works [5, 29] consider the behavior of many revenue-maximizing buyers with budget constraints in second-price markets. They take the mean-field equilibrium as the central solution concept and prove that pacing is an approximately optimal strategy for these buyers. In comparison, the main goal of this paper is to provide general relationships among different auction types. Besides the above, other works [8, 10, 23] consider the dynamic environment in which each budget-constrained buyer takes the pacing strategy. Our work, nevertheless, does not explicitly involve any learning behavior.

## 2 MODEL

This work considers the scenario where $n$ budget-constrained buyers participate in an auction. Each buyer $i \in [n]$ receives a value $v_i$ drawn i.i.d. from the distribution $F_i$, which captures her valuation of the item. We assume that $(F_i)_{i \in [n]}$ are independent of each other. Meanwhile, we use $b_i$ to denote buyer $i$'s bid. Both $v_i$ and $b_i$ are restricted to $[0, 1]$ for all $i$. Each buyer $i$ has a budget $\rho_i \in (0, 1]$ for the auction, which is known to the seller. The seller has a fixed opportunity cost $\lambda \in (0, 1)$ for the item. Opportunity cost reflects the seller's unwillingness to sell the item. The seller's revenue from selling an item is all buyers' total payment minus the opportunity cost.

In this work, we consider the stochastic setting, which is known to be a good approximation of the dynamic model and has been adopted by lines of research works [3, 5, 14, 29]. In this setting, we suppose that buyers take fixed bidding strategies. Meanwhile, the model requires that each buyer's *expected* payment does not exceed her budget. As a remark, we argue that such a model realistically captures the stationary behavior when buyers participate in a large number of auctions and the platform inquires for an "average budget constraint" [27]. To show how the stochastic setting works, we now dig into how the seller sets a budget-constrained auction mechanism and how buyers participate in the mechanism.

*Parameterized auction mechanisms and monotonicity.* We first formalize the parameterized auction mechanism adopted by the seller. Specifically, we use $\mathcal{M}(\theta) = (X(\theta), P(\theta))$ to denote a (direct) parameterized auction mechanism, where $\theta \geq \mathbf{0}$ is the parameter vector. Here, given $\theta$, $X(\theta, \cdot) : [0, 1]^n \to \Delta^n$ is the allocation function and $P(\theta, \cdot) : [0, 1]^n \to \mathbb{R}^n$ is the payment function. We should notice that the opportunity cost $\lambda$ could implicitly occur in

the formulae of $X(\theta, \cdot)$ and $P(\theta, \cdot)$. The utilization of the parameter vector $\theta$ guarantees each buyer's budget constraint and will be discussed in detail later.

With the above notation, given a value profile $\boldsymbol{v} = (v_i)_{i \in [n]}$ and a bid profile $\boldsymbol{b} = (b_i)_{i \in n}$, buyer $i$'s *utility* in the auction is:

$$U_i(\theta, \boldsymbol{b}, v_i) := X_i(\theta, \boldsymbol{b}) \cdot v_i - P_i(\theta, \boldsymbol{b}).$$

Correspondingly, the seller's *revenue* is

$$W(\theta, \boldsymbol{b}) := \sum_{i \in [n]} P_i(\theta, \boldsymbol{b}) - \lambda \cdot \sum_{i \in [n]} X_i(\theta, \boldsymbol{b}).$$

In this work, we concentrate on *monotone* parameterized auction mechanisms, with the following definition:

**Definition 2.1** (Monotonicity). We say a parameterized auction mechanism $\mathcal{M}(\theta) = (X(\theta), P(\theta))$ is monotone, if for any $\theta \geq \mathbf{0}$ and $i \in [n]$, buyer $i$'s allocation function $X_i(\theta, \cdot)$ is increasing of her bid $b_i$ regardless of other buyers' bids $\boldsymbol{b}_{-i}$.

*Bidding functions.* Given a monotone parameterized auction mechanism $\mathcal{M}(\theta)$, we now formally characterize buyers' bidding strategies. As mentioned earlier, under the stochastic setting, we suppose each buyer takes a *fixed bidding strategy*. In other words, with a slight abuse of notation, for each buyer $i \in [n]$, there is a bidding function $b_i : [0, 1] \to [0, 1]$, such that $b_i = b_i(v_i)$ for any $v_i$. Moreover, we prove the following lemma, which states that when facing a monotone auction mechanism, any buyer's best strategy is an increasing bidding function.

**Lemma 2.1.** *For any buyer and bidding function, there exists an increasing bidding function that yields at least the same utility for the buyer under any monotone parameterized auction mechanism and other bidders' strategies.*

The proof of Lemma 2.1 is given in Appendix B. With the result, we introduce the terminology of the quantile function (abbreviated as qf) [15–17, 39][1] to equivalently represent buyers' private information and strategies. We consider a specific buyer $i \in [n]$ and a quantile $q_i$ drawn *uniformly* from $[0, 1]$. With a slight abuse of notation, the value function $v_i(q_i) := \inf\{v_i : F_i(v_i) \geq q_i\}$ is an increasing function that follows the distribution $F_i$ and represents buyer $i$'s private information. We denote the value *function* profile as $\boldsymbol{v} := (v_i)_{i \in [n]}$. Further, according to Lemma 2.1, it is without loss of generality to assume that buyer $i$'s bid $b_i$ is also an increasing function of $v_i$. Hence, $b_i$ can be expressed as an increasing function of $q_i$, denoted as $b_i := \widetilde{v}_i(q_i)$. Consequently, buyer $i$'s strategy can be represented by an increasing bidding qf $\widetilde{v}_i$. Here, it is important to notice that since $\widetilde{v}_i$ operates on the *quantile space* and only reflects buyer $i$'s bidding distribution, this function is public to the seller. We denote the bidding qf profile as $\widetilde{\boldsymbol{v}} := (\widetilde{v}_i)_{i \in [n]}$.

With the above notations, we further define the *interim* allocation $x_i(\theta, q_i, \widetilde{\boldsymbol{v}})$ and payment $p_i(\theta, q_i, \widetilde{\boldsymbol{v}})$ of any buyer $i$ in each auction, given the strategies of other buyers. Denoting $\widetilde{\boldsymbol{v}}(\boldsymbol{q}) := (\widetilde{v}_i(q_i))_{i \in [n]}$, we define

$$x_i(\theta, q_i, \widetilde{\boldsymbol{v}}) := \mathbb{E}_{\boldsymbol{q}_{-i}} [X_i(\theta, \widetilde{\boldsymbol{v}}(\boldsymbol{q}))],$$
$$p_i(\theta, q_i, \widetilde{\boldsymbol{v}}) := \mathbb{E}_{\boldsymbol{q}_{-i}} [P_i(\theta, \widetilde{\boldsymbol{v}}(\boldsymbol{q}))].$$

---

[1]Our model is essentially equivalent to the prior manipulation model as proposed in these works. However, unlike their terminology, we employ the left quantile function in this work, which is a more accessible way to define a quantile function.

Therefore, buyer $i$'s interim utility in the auction is:

$$u_i(\boldsymbol{\theta}, q_i, \widetilde{\boldsymbol{v}}, v_i) := x_i(\boldsymbol{\theta}, q_i, \widetilde{\boldsymbol{v}}) \cdot v_i(q_i) - p_i(\boldsymbol{\theta}, q_i, \widetilde{\boldsymbol{v}}),$$

and with a slight abuse of notation, her expected utility is:

$$u_i(\boldsymbol{\theta}, \widetilde{\boldsymbol{v}}, v_i) := \int_0^1 u_i(\boldsymbol{\theta}, q_i, \widetilde{\boldsymbol{v}}, v_i) \, \mathrm{d}q_i. \quad (1)$$

Finally, the seller's expected revenue in the auction is given by

$$w(\boldsymbol{\theta}, \widetilde{\boldsymbol{v}}) := \mathbb{E}_{\boldsymbol{q}}\left[\sum_{i \in [n]} P_i(\boldsymbol{\theta}, \widetilde{\boldsymbol{v}}(\boldsymbol{q})) - \lambda \cdot \sum_{i \in [n]} X_i(\boldsymbol{\theta}, \widetilde{\boldsymbol{v}}(\boldsymbol{q}))\right]$$

$$= \sum_{i \in [n]} \left(\int_0^1 (p_i(\boldsymbol{\theta}, q_i, \widetilde{\boldsymbol{v}}) - \lambda \cdot x_i(\boldsymbol{\theta}, q_i, \widetilde{\boldsymbol{v}})) \, \mathrm{d}q_i\right). \quad (2)$$

*Budget-constrained auction mechanisms.* We now examine how the parameter vector $\boldsymbol{\theta}$ acts in a budget-constrained auction mechanism. Here, a crucial observation is that in real-life scenarios, buyers would react to a parameterized auction mechanism by devising a bidding strategy, which would subsequently influence the evolution of the parameter vector. From an information-accessing standpoint, the seller only sees buyers' bidding distributions (or, equivalently, bidding qfs) throughout the process.

Under the stochastic model [3, 5, 14], which serves as a good simplification of the complicated dynamic process, we assume that buyers report their bidding quantile functions $(\widetilde{v}_i)_{i \in [n]}$ to the seller, and the parameter vector $\boldsymbol{\theta}$ is a *pre-known public* function of $(\widetilde{v}_i)_{i \in [n]}$ and buyers' budgets $(\rho_i)_{i \in [n]}$. Under such modeling, it is essential for the parameter vector choice to adhere to the budget constraints as a fundamental requirement. For this part, we make the following definition under the stochastic setting:

**Definition 2.2** (Budget feasibility). A parameterized auction mechanism $\mathcal{M}(\boldsymbol{\theta})$ is budget-feasible, if for any bidding qf profile $(\widetilde{v}_i)_{i \in [n]}$ and budget profile $(\rho_i)_{i \in [n]}$, the mechanism and the corresponding parameter vector $\boldsymbol{\theta}$ satisfy that:

$$\int_0^1 p_i(\boldsymbol{\theta}, q_i, \widetilde{\boldsymbol{v}}) \, \mathrm{d}q_i \le \rho_i, \quad \forall 1 \le i \le n. \quad (\text{BF})$$

Meanwhile, *individual rationality* is also a crucial requirement for budget-constrained mechanisms, ensuring that any buyer's utility is non-negative as long as she truthfully bids:

**Definition 2.3** (Individual rationality). A parameterized auction mechanism $\mathcal{M}(\boldsymbol{\theta})$ is individually rational, if for any value qf profile $(v_i)_{i \in [n]}$ and budget profile $(\rho_i)_{i \in [n]}$, the mechanism and the corresponding parameter vector $\boldsymbol{\theta}$ satisfies that:

$$\int_0^1 u_i(\boldsymbol{\theta}, q_i, \boldsymbol{v}, v_i) \, \mathrm{d}q_i \ge 0, \quad \forall 1 \le i \le n. \quad (\text{IR})$$

*Unassured prior game with budget constraints among the seller and buyers.* As a conclusion of the above, we present the game-theoretic interaction between the seller and buyers for a better understanding. We call this game the *unassured prior game (with budget constraints)*:

**Step 1.** The seller commits to a parameterized auction mechanism $\mathcal{M}(\boldsymbol{\theta})$ (with a monotone allocation rule), along with a public decision rule for $\boldsymbol{\theta}$. We assume that the auction mechanism is budget-feasible (BF) and individually rational (IR).

**Step 2.** Buyers' value qf's $\{v_i\}_{i \in [n]}$ are private. Each buyer $i \in [n]$ chooses an *increasing* bidding qf $\widetilde{v}_i$ and reports it to the seller. Additionally, buyers truthfully provide the budget profile $(\rho_i)_{i \in [n]}$ to the seller.

**Step 3.** Given $\widetilde{\boldsymbol{v}} = (\widetilde{v}_i)_{i \in [n]}$ and $(\rho_i)_{i \in [n]}$, the parameter vector $\boldsymbol{\theta}$ is computed, and $\mathcal{M}(\boldsymbol{\theta})$ is run. Buyer $i \in [n]$'s utility is $u_i(\boldsymbol{\theta}, \widetilde{\boldsymbol{v}}, v_i)$ given in (1). The seller's revenue is $w(\boldsymbol{\theta}, \widetilde{\boldsymbol{v}})$ given in (2).

*Other terminologies.* With the bidding qf, we now derive an expression of the virtual bidding qf. Specifically, for a *strictly increasing* and differentiable bidding qf $\widetilde{v}$ with cumulative distribution function $\widetilde{F}$ and density $\widetilde{f}$, the virtual valuation is given by $v - (1 - \widetilde{F}(v))/\widetilde{f}(v)$. Therefore, for any quantile $q \in [0, 1]$, the virtual bidding qf is

$$\widetilde{\psi}(q) := \widetilde{v}(q) - \frac{1 - \widetilde{F}(\widetilde{v}(q))}{\widetilde{f}(\widetilde{v}(q))} = \widetilde{v}(q) - (1 - q)\widetilde{v}'(q).$$

We use $(\widetilde{\psi}_i)_{i \in [n]}$ to represent buyers' virtual bidding qfs. We say a bidding qf $\widetilde{v}$ is *(strictly) regular* if the corresponding virtual bidding qf is (strictly) increasing.

Further, an important assumption that we repeatedly make in this work is that each buyer's (virtual) bidding qf is *inverse Lipschitz continuous*, with the following definition:

**Definition 2.4** (Inverse Lipschitz continuity). We say a function $g : [0, 1] \to [0, 1]$ is inverse Lipschitz continuous, if there is a constant $L > 0$, such that for any $0 \le q_1 < q_2 \le 1$:

$$|g(q_2) - g(q_1)| \ge L(q_2 - q_1).$$

As an example, inverse Lipschitz continuity of the bidding qf implies Lipschitz continuity of the *bidding distribution CDF*. We note that since the seller typically learns the bidding distribution using parameterized continuous models, the above assumption is natural, considering that the seller will adopt a relatively simple model, e.g., truncated power-law distribution or Gaussian distribution for the density. Additionally, we need to emphasize that an inverse Lipschitz continuous function need not be continuous itself.

*Discussions on the model.* In this work, a buyer's budget is not explicitly incorporated into her utility function as long as her expected payment is within her budget. In practice, the budget for advertisement is usually set as a fixed and sunk cost within the company. On this side, the advertiser's goal is to maximize her quasi-linear utility while operating within the budget. If the budget is exceeded, we implicitly assume that the advertiser's utility becomes negative infinity. Such a model has been adopted by various works in literature [3, 4, 12–14, 21].

## 3 MECHANISMS

This work examines five budget-constrained auction forms, all of which take the opportunity cost $\lambda$ as a reserve price. We list these five mechanisms in Table 1. They include two variants of first-price auctions, two variants of second-price auctions, and the optimal auction under budget constraints. These methods effectively control a buyer's expenditure in two ways: (1) by reducing the likelihood of winning through shading the effective bid and incorporating the reserve price $\lambda > 0$, and (2) by reducing a buyer's payment

**Table 1: Definitions of parameterized mechanisms considered in this work. Buyers bid their quantiles which are uniformly drawn in $[0,1]$.**

| Mech. (Abbrev.) | Parameters | Allocation / Payment Rules |
|---|---|---|
| Bid-Discount FPA (BDFPA) | $\boldsymbol{\alpha} \in [0,1]^n$ | Buyer $i \in [n]$: wins if $\alpha_i \widetilde{v}_i(q_i) \geq \max_{i' \neq i}\{\alpha_{i'}\widetilde{v}_{i'}(q_{i'}), \lambda\}$, and pays $\widetilde{v}_i(q_i)$; pays 0 otherwise. |
| Pacing FPA (PFPA) | $\boldsymbol{\beta} \in [0,1]^n$ | Buyer $i \in [n]$: wins if $\beta_i \widetilde{v}_i(q_i) \geq \max_{i' \neq i}\{\beta_{i'}\widetilde{v}_{i'}(q_{i'}), \lambda\}$, and pays $\beta_i \widetilde{v}_i(q_i)$; pays 0 otherwise. |
| Bayesian Revenue-Optimal Auction (BROA) | $\boldsymbol{\gamma}^* \in [0,1]^n$ given by (3) | Buyer $i \in [n]$: wins if $\gamma_i^* \widetilde{\psi}_i(q_i) \geq \max_{i' \neq i}\{\gamma_{i'}^*\widetilde{\psi}_{i'}(q_{i'}), \lambda\}$, and pays $\min\{\widetilde{v}_i(z) : \gamma_i^* \widetilde{\psi}_i(z) \geq \max_{i' \neq i}\{\gamma_{i'}^*\widetilde{\psi}_{i'}(q_{i'}), \lambda\}\}$; pays 0 otherwise. |
| Bid-Discount SPA (BDSPA) | $\boldsymbol{\mu} \in [0,1]^n$ | Buyer $i \in [n]$: wins if $\mu_i \widetilde{v}_i(q_i) \geq \max_{i' \neq i}\{\mu_{i'}\widetilde{v}_{i'}(q_{i'}), \lambda\}$, and pays $\max_{i' \neq i}\{\mu_{i'}\widetilde{v}_{i'}(q_{i'}), \lambda\}/\mu_i$; pays 0 otherwise. |
| Pacing SPA (PSPA) | $\boldsymbol{\xi} \in [0,1]^n$ | Buyer $i \in [n]$: wins if $\xi_i \widetilde{v}_i(q_i) \geq \max_{i' \neq i}\{\xi_{i'}\widetilde{v}_{i'}(q_{i'}), \lambda\}$, and pays $\max_{i' \neq i}\{\xi_{i'}\widetilde{v}_{i'}(q_{i'}), \lambda\}$; pays 0 otherwise. |

FPA: First-Price Auction    SPA: Second-Price Auction

when she wins. We should notice that all these five mechanisms are monotone (Definition 2.1) and satisfy individual rationality (Definition 2.3). Meanwhile, we assume that these mechanisms break ties arbitrarily. We now discuss these mechanisms in more detail, starting with the well-studied second-price auctions, followed by the first-price auctions, and concluding with the optimal auction.

The bid-discount method [25, 35] and the pacing method [5, 8] have been widely applied to second-price auctions in literature. Under both mechanisms, the seller assigns a multiplier in $[0,1]$ to each buyer, and buyers are ranked based on the bids shaded by the multiplier. The difference between these two mechanisms is that for the pacing method, the winner pays the second-highest paced bid, while for the bid-discount method, the winner's payment is the lowest winning bid.

Pacing has also been studied in first-price auctions [13], where the winner pays her shaded bid. Meanwhile, we naturally extend the bid-discount method to first-price auctions. In this mechanism, the winner pays her original bid rather than the shaded bid. Combined with the reserve price, the bid-discount first-price auction controls a buyer's expenditure by reducing the likelihood of winning. We will delve into more structural properties of the mechanism in Section 5 and appendix A.

Finally, we introduce the Bayesian revenue-optimal auction proposed by Balseiro et al. [3]. This mechanism maximizes the seller's revenue among all budget-constrained incentive-compatible auctions when all buyers' bidding qfs are *strictly regular*. In the mechanism, buyers are ranked according to the shaded virtual bids, and the winner's payment is the lowest winning bid. Here, the optimal shading parameter $\boldsymbol{\gamma}^*$ is given by the following optimization

problem:

$$\boldsymbol{\gamma}^* := \arg\min_{\boldsymbol{\gamma} \in [0,1]^n} \left\{ \mathbb{E}_q \left[ \max_i \left\{ \gamma_i \widetilde{\psi}_i(q_i) - \lambda \right\}^+ \right] + \sum_{i=1}^n (1 - \gamma_i)\rho_i \right\}. \tag{3}$$

### 3.1 The Budget-Extracting Concept

From the seller's perspective, a crucial objective is to maximize his revenue, and one direct approach to achieving this is to fully utilize buyers' budgets. To settle this idea, we now define a budget-extracting concept that resembles the system equilibrium concept given in Balseiro et al. [3]. This concept applies to variants of first-price and second-price auctions. Specifically, the seller can carefully set the parameter vector such that either the budget feasibility constraint or the IR constraint is binding for each buyer. Formally, we give the following definition:

**Definition 3.1** (Budget-extracting). We say a parameterized auction mechanism $\mathcal{M}(\boldsymbol{\theta})$ is budget-extracting, if for any bidding qf profile $(\widetilde{v}_i)_{i \in [n]}$ and budget profile $(\rho_i)_{i \in [n]}$, the mechanism and the corresponding parameter vector $\boldsymbol{\theta}$ satisfies that:

$$\left( \int_0^1 p_i(\boldsymbol{\theta}, q_i, \widetilde{v}) \, dq_i \leq \rho_i \right) \perp \left( \theta_i \leq \bar{\theta}_i \right).$$

Here, $\bar{\theta}_i$ is the upper limit of $\theta_i$, and the $\perp$ notation means that at least one of the two constraints is binding.

We now demonstrate that the budget-extracting concept can be realized for two variants of first-price auctions and is well-defined for second-price auctions when buyers are symmetric. Further, we show that the budget-extracting mechanism is the optimal choice for the seller in the case of BDFPA, PFPA, and symmetric PSPA auctions.

**Theorem 3.1.** *We have the following:*

1. *When each buyer's bidding qf is inverse Lipschitz continuous, both BDFPA and PFPA support a budget-extracting mechanism.*

2. *When all buyers are symmetric, and their common bidding qf is inverse Lipschitz continuous, both BDSPA and PSPA support a symmetric budget-extracting mechanism.*

*Further, for BDFPA, PFPA, and symmetric PSPA, under the above conditions, respectively, committing to a budget-extracting mechanism maximizes the seller's revenue among all mechanisms.*

The proof of Theorem 3.1 is deferred to Appendix C. We should notice that the revenue-maximizing part of Theorem 3.1 works for all buyers' bidding profiles under natural conditions. Consequently, under the respective constraints, we only need to consider budget-extracting mechanisms as they represent the seller's optimal choice regardless of the buyers' strategies. For brevity, we take eBDFPA as an abbreviation of "budget-extracting BDFPA" in the rest of this work. The same abbreviation also holds for PFPA, BDSPA, and PSPA.

## 4 STRATEGIC EQUIVALENCE RESULTS

In the previous section, we have established that for two variations of first-price auctions and the symmetric pacing second-price auction, the optimal parameter choice for the seller is to satisfy the budget-extracting requirement by adequately utilizing the buyers' budgets. In this section, we focus on the buyers' perspective and explore their bidding strategies when facing different budget-extracting mechanisms or the optimal mechanism BROA. Specifically, we show broad strategic equivalence results among these five parameterized mechanisms in the unassured prior game. First, we introduce two notions of strategic equivalence with varying levels of guarantees.

**Definition 4.1** (Strategic equivalence). We say two parameterized auction mechanisms $\mathcal{M}_1(\boldsymbol{\theta})$ and $\mathcal{M}_2(\boldsymbol{\theta})$ are *weakly strategic-equivalent* (in the unassured prior game), if there are two mappings $G, H : ([0,1] \to [0,1])^n \to ([0,1] \to [0,1])^n$ such that for any strategic bidding profiles $\widetilde{\boldsymbol{v}}$, $G(\widetilde{\boldsymbol{v}})$ under $\mathcal{M}_2(\boldsymbol{\theta})$ brings the same utility-revenue profile with $\widetilde{\boldsymbol{v}}$ under $\mathcal{M}_1(\boldsymbol{\theta})$; and $H(\widetilde{\boldsymbol{v}})$ under $\mathcal{M}_1(\boldsymbol{\theta})$ brings the same revenue-utility profile with $\widetilde{\boldsymbol{v}}$ under $\mathcal{M}_2(\boldsymbol{\theta})$. Further, if $G$ and $H$ operate independently and identically as $g$ and $h$ on each bidding function, we say $\mathcal{M}_1(\boldsymbol{\theta})$ and $\mathcal{M}_2(\boldsymbol{\theta})$ are *strongly strategic-equivalent* (in the unassured prior game).

In general, weak strategic equivalence, as defined above, indicates that the sets of utility-revenue profiles under two parameterized mechanisms are identical. Additionally, strong strategic equivalence requires that each buyer's strategy profile mapping be independent and anonymous. An important observation is that under strong strategic equivalence, if the two mappings $g$ and $h$ are further inverse functions of each other, then for any bidder $i$, if $\widetilde{v}_i$ is a best-response for other bidders' strategy $\widetilde{\boldsymbol{v}}_{-i}$ under $\mathcal{M}_1(\boldsymbol{\theta})$, $g(\widetilde{v}_i)$ would also be a best-response to $g(\widetilde{\boldsymbol{v}}_{-i})$ under $\mathcal{M}_2(\boldsymbol{\theta})$ since $g$ keeps the outcome. The same applies vice versa for $h = g^{-1}$. As a result, $g$ gives a one-to-one mapping between Nash equilibria of these two parameterized mechanisms while preserving the utility-revenue

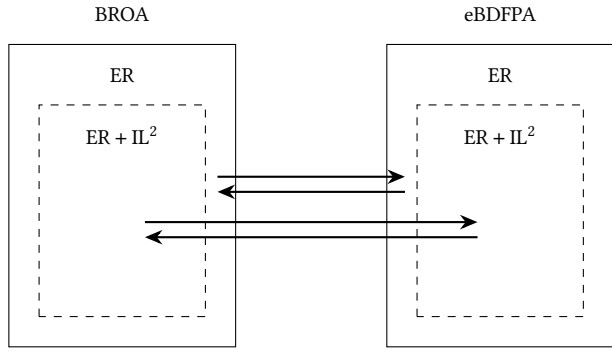

**Figure 3: An illustration of Theorem 4.2. ER: Each buyer's virtual bidding qf is strictly increasing and differentiable; IL$^2$: each buyer's bidding qf and virtual bidding qf are both inverse Lipschitz continuous.**

profile. This observation is formalized in the following lemma. All missing proofs in this section can be found in Appendix D.

**Lemma 4.1.** *If two parameterized mechanisms $\mathcal{M}_1(\boldsymbol{\theta})$ and $\mathcal{M}_2(\boldsymbol{\theta})$ are strongly strategic-equivalent and the corresponding mappings $G$ and $H$ ($g$ and $h$) are inverse of each other, then the two sets comprised of all Nash equilibrium utility-revenue profiles respectively for these two mechanisms are the same in the unassured prior game.*

The rest of this section discusses the weak/strong strategic equivalence relationships among the five mechanisms. We begin by considering the general case where buyers can be asymmetric and then proceed to the symmetric case.

### 4.1 General Case

We present our main results in the general case when buyers can be asymmetric. Specifically, we establish a strong strategic equivalence between BROA, the optimal budget-constrained mechanism, and eBDFPA, the budget-extracting bid-discount first-price auction, under minor conditions on the buyers' strategic bidding functions. The formal statement of this result is as follows:

**Theorem 4.2.** *When each buyer's virtual bidding qf is restricted to be strictly increasing and differentiable, BROA and eBDFPA are strongly strategic-equivalent in the unassured prior game. This result also holds when each buyer's bidding qf and virtual bidding qf are both inverse Lipschitz continuous.*

To provide a visual representation of Theorem 4.2, we include an illustration in Figure 3. The solid rectangles represent the subsets of BROA and eBDFPA with strictly increasing and differentiable virtual bidding qfs, which are equivalent under independent mappings. Furthermore, the dashed rectangles, which are subsets of the solid rectangles, indicate that when we additionally require the inverse Lipschitz continuity of both bidding qfs and virtual bidding qfs, these restricted subsets remain equivalent. The proof of Theorem 4.2 is based on the study of the properties of budget-extracting BDFPA. The intuition is to notice the intrinsic similarity between the programming of BROA and eBDFPA. Nevertheless, the rigorous proof is far more complex. In particular, it involves the construction of a mapping between the bidding strategies under these

two mechanisms that result in the same utility-revenue profile outcome. For this part, we adopt a technique we call "lifting" to adjust those intractable bidding functions.

It is important to notice that Lemma 4.1 cannot be directly applied to Theorem 4.2 since the mappings we construct are not inverses of each other on those "bad-behaved" strategy profiles. However, concerning buyers' utilities and the seller's revenue, we can employ the "lifting" technique to filter out these profiles by showing their equivalence with well-behaved ones. Led by the observation, we can further derive the following important theorem.

**Theorem 4.3.** *When each buyer's virtual bidding qf is restricted to be strictly increasing and differentiable, BROA and eBDFPA have the same set of Nash equilibrium utility-revenue profiles in the unassured prior game. This result also holds when each buyer's bidding qf and virtual bidding qf are both inverse Lipschitz continuous.*

At a high level, Theorems 4.2 and 4.3 extend the results in Deng et al. [15], Tang and Zeng [39] to budget-constrained stochastic auctions. These two theorems are significant results indicating that when buyers' strategic bidding behaviors affect the learning behavior of the seller and, therefore, the parameter vector, the optimal mechanism and budget-extracting BDFPA are strongly strategic-equivalent, and they yield the identical set of Nash equilibrium outcomes. It is worth noting that while BROA may be a complex auction form in practice, budget-extracting BDFPA is easier to comprehend and implement in ad platforms. Therefore, it is reasonable for platforms to favor accessible mechanisms, as they perform just as robust in the face of buyer uncertainty. Moreover, these results provide further justification for major platforms to transition to first-price auctions in the current auto-bidding environment [26], as they can behave as satisfying as the optimal auction.

## 4.2 Symmetric Case

We also consider the symmetric case, where all buyers' budgets, value qfs, and bidding qfs are correspondingly identical. In this case, we naturally examine the budget-extracting case when the parameter vector is symmetric. Under such circumstances, we provide broad weak strategic equivalence results, which bridge two variants of the first-price auction and two variants of the second-price auction.

**Theorem 4.4.** *In the symmetric case, when all buyers' identical bidding qf is inverse Lipschitz continuous, under the symmetric budget-extracting parameter vector, eBDFPA, ePFPA, eBDSPA, and ePSPA are all weakly strategic-equivalent in the unassured prior game.*

Therefore, these four mechanisms have the same outcome space in symmetry. Combining with Theorem 4.2, we further have the following two corollaries:

**Corollary 4.5.** *Under the conditions of Theorem 4.4, eBDFPA, ePFPA, eBDSPA, and ePSPA have the same set of utility-revenue profiles in the unassured prior game.*

**Corollary 4.6.** *Under the conditions of Theorems 4.2 and 4.4, BROA, eBDSPA, ePFPA, eBDSPA, and ePSPA are all weakly strategical-equivalent and have the same set of utility-revenue profiles in the unassured prior game.*

The proof of Theorem 4.4 primarily involves constructing mappings between the bidding functions under different budget-constrained auction mechanisms. A crucial point in the proof is that eBDFPA and ePFPA exhibit a symmetric parameter vector in the symmetric setting. This observation is a corollary of their properties described in Appendix A.1, and greatly aids the mapping construction.

Together, Theorem 4.4 and Corollary 4.6 demonstrate the extensive strategic equivalence of various budget-constrained mechanisms in the symmetric sense. We should mention the distinction between these results and the celebrated revenue equivalence theorem for a better understanding. First, the five mechanisms we discuss do not always lead to the same allocation with a fixed quantile profile due to the existence of the opportunity cost as a reserve price. Second, revenue equivalence results assume that the common value prior is known advance to the seller, whereas we do not make such an assumption in this work. At last, the revenue equivalence theorem focuses on the symmetric equilibrium outcome, while our result is not limited to symmetric equilibria but gives a broad strategic equivalence result regardless of the specific strategy.

## 5 STRUCTURAL PROPERTIES OF MECHANISMS

With the strategic equivalence results we have already given in Section 4, we proceed to analyze the structural properties of these mechanisms. The analysis consists of two parts. To start with, we will exploit the computational properties of BDFPA and PFPA. Further, we will reveal the revenue dominance relationships among these five mechanisms on the seller's side when buyers do not adopt strategic bidding. However, due to the space limit, we defer the results and details to Appendix A.

## 6 CONCLUDING REMARKS

This work considers the scenario where the seller lacks knowledge of the value priors of budget-constrained buyers. We investigate five mechanisms in this context: the Bayesian revenue-optimal auction, as well as the bid-discount and pacing variations of the first-price and second-price auctions. We characterize the unassured prior game between the seller and buyers under these auction forms and focus on budget-extracting mechanisms, which maximizes the seller's revenue. We give a strong strategic equivalence result between the bid-discount first-price auction and Bayesian revenue-optimal auction from the view of Nash equilibria, indicating that simple mechanisms can be as robust as optimal ones in the presence of unassured priors. This result sheds light on the valuation of first-price auctions in the auto-bidding world. We further establish vast outcome equivalence results among first-price/second-price auctions with budget constraints. In terms of structural properties, we explore the characteristics of bid-discount/pacing first-price auctions under the budget-extracting condition. Moreover, we compare the seller's revenue under these mechanisms when there is no strategic behavior. Overall, our work contributes to a comprehensive understanding of budget-constrained auction mechanisms, particularly first-price ones, from a stochastic perspective.

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

# A DETAILS FOR STRUCTURAL PROPERTIES OF MECHANISMS IN SECTION 5

This section complements the discussions in Section 5 by providing the results and details. We refer readers to Appendix E for missing proofs in this section.

## A.1 Properties of Variants of First-Price Auctions

*A.1.1 Properties of BDFPA.* We first present properties of (budget-extracting) BDFPA, which we aggregate in the following theorem.

**Theorem A.1.** *Given buyers' bidding qf profile $(\widetilde{v}_i)_{i \in [n]}$ and budget profile $(\rho_i)_{i \in [n]}$, if each buyer $1 \le i \le n$'s bidding qf $\widetilde{v}_i(\cdot)$ is inverse Lipschitz continuous, then the following statements hold:*

1. *There exists a* maximum *tuple of bid-discount multipliers $\alpha^{\max}$, i.e., for any feasible tuple of bid-discount multipliers $\alpha$, $\alpha_i^{\max} \ge \alpha_i$ for any $1 \le i \le n$.*
2. *$\alpha^{\max}$ is a budget-extracting tuple of bid-discount multipliers.*
3. *For any budget-extracting bid-discount multiplier tuple $\alpha^{e}$, the following two conditions establish:*
   (a) *There exists some $v \le 1$, such that for any $1 \le i \le n$ satisfying $p_i^{\max}$ (buyer $i$'s expected payment in $\mathcal{M}^{\mathrm{BDFPA}}(\alpha^{\max})$) is positive, $\alpha_i^{e}/\alpha_i^{\max} = v$;*
   (b) *For any $1 \le i \le n$ satisfying $p_i^{\max} = 0$, $p_i^{e} = 0$ (buyer $i$ never wins in $\mathcal{M}^{\mathrm{BDFPA}}(\alpha^{e})$) and $\alpha_i^{e} = \alpha_i^{\max} = 1$.*
4. *All budget-extracting BDFPAs bring the* same *payment for each buyer.*
5. *$\alpha^{\max}$ is the unique budget-extracting tuple of bid-discount multipliers if and only if either one of the following two conditions is satisfied:*
   (a) *$\max_{i \in \mathcal{I}_1} \alpha_i^{\max} \widetilde{v}_i(0) \le \max_{i \in \mathcal{I}_2} \{\alpha_i^{\max} \widetilde{v}_i(1), \lambda\}$, where $\mathcal{I}_1 = \{i \mid p_i^{\max} > 0\}$ and $\mathcal{I}_2 = [n] \setminus \mathcal{I}_1$, or*
   (b) *there exists $i \in \mathcal{I}_1$ such that $p_i^{\max} < \rho_i$.*

We would like to emphasize once again that the inverse Lipschitz continuity assumption on $\widetilde{v}_i(\cdot)$ is a relatively weak one and is commonly adopted in previous works [3, 22, 30]. In the proof of Theorem A.1, this assumption guarantees that a small increase in a buyer's bid-discount multiplier does not significantly change her payment. This continuity property indicates that the budget-feasible tuples form a closed set, and serves as a key lemma in proving the first two statements of the theorem. As for characterizing the set of budget-extracting multiplier tuples, we observe that the budget-extracting condition imposes a much tight restriction. Specifically, for any buyer, her payment should be the same across all budget-extracting BDFPA mechanisms. This essential observation helps with the remaining three statements.

We highlight that Theorem A.1 provides a comprehensive depiction of the behavior of budget-extracting BDFPA(s) under minor restrictions. As revealed, all budget-extracting BDFPAs share similar structures. We further demonstrate that a budget-extracting BDFPA can be efficiently computed with convex optimization techniques.

**Theorem A.2.** *A budget-extracting BDFPA can be computed by solving the global minimum of a convex function.*

For interested readers, we mention that the underlying convex function mentioned in Theorem A.2 is given by $\chi^{\mathrm{BDF}}(\tau)$ as given in (6) for $\tau \in [0,1]^n$, and a budget-extracting tuple is given by $\alpha = 1^n - \tau^*$ with $\tau^*$ the global minimum of $\chi^{\mathrm{BDF}}(\tau)$.

*A.1.2 Properties of PFPA.* For PFPA, we also derive a counterpart of Theorem A.1, as in the following theorem.

**Theorem A.3.** *Given buyers' bidding qf profile $(\widetilde{v}_i)_{i \in [n]}$ and budget profile $(\rho_i)_{i \in [n]}$, if each buyer $1 \le i \le n$'s bidding qf $\widetilde{v}_i(\cdot)$ is inverse Lipschitz continuous, then:*

1. *There exists a* maximum *tuple of pacing multipliers $\beta^{\max}$, i.e., for any feasible tuple of pacing multipliers $\beta$, $\beta_i^{\max} \ge \beta_i$ for any $1 \le i \le n$.*
2. *$\beta^{\max}$ is the* unique *budget-extracting tuple of pacing multipliers.*
3. *$\beta^{\max}$ maximizes seller's revenue among all feasible tuples of pacing multipliers, i.e., $\beta^{\max}$ is the optimal solution of the following programming:*

$$\max_{\beta \in [0,1]^n} \int_{q} \max_{i} \{\beta_i \widetilde{v}_i(q_i) - \lambda\}^{+} \, \mathrm{d}q,$$

$$\text{s.t.} \quad \int_0^1 \beta_i \widetilde{v}_i(q_i) \cdot \left( \int_{q_{-i}} I\left[ \beta_i \widetilde{v}_i(q_i) \ge \max_{i' \ne i} \{\beta_{i'} \widetilde{v}_{i'}(q_{i'}), \lambda\} \right] \mathrm{d}q_{-i} \right) \mathrm{d}q_i \le \rho_i, \tag{4}$$

$$\forall 1 \le i \le n.$$

An interesting point here is that, unlike in the case of BDFPA, there is only one budget-extracting PFPA under minor restrictions. The main reason here is that in PFPA, a buyer's payment is correlated with her multiplier, while this is not the case for BDFPA as long as she wins.

## A.2 Dominance Relationships on the Seller's Revenue

Now let us compare these five mechanisms in terms of the seller's revenue when all buyers bid truthfully. In other words, we do not consider buyers' strategic behaviors and write the value/bidding qf profile as $(\widetilde{v}_i)_{i \in [n]}$. This assumption allows for a better understanding of the intrinsic allocation/payment properties of budget-constrained mechanisms. Here, we should notice that such comparisons are not

**Table 2: Summary of Example A.1.**

|  | eBDFPA | ePFPA | BROA | eBDSPA | ePSPA |
|---|---|---|---|---|---|
| Each buyer's payment | 0.312 | 0.312 | 0.207 | 0.171 | 0.171 |
| Seller's revenue | 0.54 | 0.525 | 0.344 | 0.243 | 0.243 |
| Budget exhausted? | Yes | Yes | No | No | No |

straightforward since the mechanism should ensure each buyer's budget constraint is satisfied, leading to potentially different parameter tuples for different mechanisms. This complicates the analysis of the seller's revenue.

In particular, we consider these mechanisms under the budget-extracting condition. The dominance relationships are presented in the following theorem. Here, we say $A \succeq B$ if the seller's revenue in A is higher than his revenue in B when all buyers bid truthfully.

**Theorem A.4.** *The following dominance relationships hold with respect to the seller's revenue:*

1. *When each buyer's bidding qf is strictly increasing and strictly regular, eBDFPA $\succeq$ BROA.*
2. *When each buyer's bidding qf is strictly increasing, eBDFPA $\succeq$ ePSPA.*
3. *When each buyer's bidding qf is strictly regular, BROA $\succeq$ eBDSPA, and BROA $\succeq$ ePSPA.*

We derive this theorem through two key observations. For the first and second results, we identify the inherent relationships among the programming of eBDFPA, BROA, and ePFPA. For the third part, we employed the budget-constrained incentive compatibility methodology introduced in Balseiro et al. [3]. A corollary of Theorem A.4 is that under mild assumptions, eBDFPA dominates the other four mechanisms when buyers truthfully bid.

**Corollary A.5.** *When each buyer's bidding qf is strictly increasing and strictly regular, eBDFPA $\succeq$ {BROA, ePSPA, eBDSPA, ePSPA}.*

We now use an example further to illustrate Theorem A.4 and corollary A.5.

**Example A.1.** Now consider a symmetric scenario with $n = 2$ buyers. Either buyer's value/bidding pdf is a uniform distribution on $[0, 1]$, and either buyer's budget is $\rho_0 = 39/125 = 0.312$. Let the opportunity cost of the seller be $\lambda = 0.1$. Then, the value/bidding qf of each buyer is $\widetilde{v}_0(\cdot)$ with $\widetilde{v}_0(x) = x$ on $[0, 1]$, and the virtual value/bidding qf $\widetilde{\psi}_0(\cdot)$ satisfies $\widetilde{\psi}_0(x) = 2x - 1$ on $[0, 1]$. Consequently, we have the following for eBDFPA, ePFPA, BROA, eBDSPA, and ePSPA, respectively:

- For eBDFPA, the maximum budget-extracting multiplier tuple $\boldsymbol{\alpha}^{\max} = (1/4, 1/4)$. Both buyers exhaust their budgets in expectation, and the seller's expected revenue equals 0.54.
- For ePFPA, the maximum budget-extracting multiplier $\boldsymbol{\beta}^{\max} = (\beta_0, \beta_0)$, where $\beta_0 \approx 0.937$ is the solution to $1000\beta^3 - 936\beta^2 - 1 = 0$. Both buyers also exhaust their budget in expectation, and the seller's expected revenue is approximately 0.525.
- For BROA, the solution to programming (3) is $\boldsymbol{\gamma}^* = (0, 0)$. Either buyer's expected payment is 0.207, and the seller's expected revenue equals $1377/4000 \approx 0.344$.
- For eBDSPA, the budget-extracting multiplier tuple is $\boldsymbol{\mu}^* = (1, 1)$. Either buyer's payment is 0.171, and the seller's expected revenue equals 0.243.
- For ePSPA, the budget-extracting multiplier tuple is also $\boldsymbol{\xi}^* = (1, 1)$, and therefore, either buyer's payment is 0.171, and the seller's expected revenue equals 0.243 as well.

For a better view, we list the above numerical results in Table 2.

## B  PROOF OF LEMMA 2.1

In accordance with the notation, we let $b_i = \widetilde{v}_i(q_i)$, and it suffices for us to prove that the best choice of $\widetilde{v}_i$ is an increasing function since $v_i$ is also increasing of $q_i$. With the other buyers' bidding strategies fixed, write out the buyer $i$'s expected utility as

$$\int_0^1 (x_i(\boldsymbol{\theta}, q_i, \widetilde{\boldsymbol{v}}) \cdot v_i(q_i) - p_i(\boldsymbol{\theta}, q_i, \widetilde{\boldsymbol{v}})) \, \mathrm{d}q_i.$$

Now, for a non-negative function $f$, as given in Bennett and Sharpley [9], we define its distribution function as

$$\mu_f(s) := \mu\{x : f(x) \geq s\},$$

and therefore, the decreasing rearrangement of $f$ is given as

$$f^*(t) := \inf\{s : \mu_f(s) \leq t\}.$$

For the function on $[0, 1]$, we further define its increasing rearrangement as

$$(f)^+(t) := (f)^*(1 - t).$$

Apparently, $(\widetilde{v}_i)^+$ is an increasing function. Notice that $p_i(\boldsymbol{\theta}, q_i, \widetilde{\boldsymbol{v}})$ is in fact a function of $\widetilde{v}_i(q_i)$. Now, since the increasing rearrangement $\widetilde{v}_i \to (\widetilde{v}_i)^+$ is a uniform function preserving function, by Porubský et al. [36], we have

$$\int_0^1 p_i(\boldsymbol{\theta}, q_i, \widetilde{\boldsymbol{v}}) \, \mathrm{d}q_i = \int_0^1 p_i(\boldsymbol{\theta}, q_i, ((\widetilde{v}_i)^+, \widetilde{\boldsymbol{v}}_{-i})) \, \mathrm{d}q_i.$$

Therefore, we only need to consider $x_i(\boldsymbol{\theta}, q_i, \widetilde{\boldsymbol{v}}) \cdot v_i(q_i)$. Since the allocation function is monotone, $x_i(\boldsymbol{\theta}, q_i, \widetilde{\boldsymbol{v}})$ is an increasing function of $\widetilde{v}_i(q_i)$. Meanwhile, notice that $v_i$ is increasing as well. By Lieb and Loss [32][2], we have

$$(x_i)^+(\boldsymbol{\theta}, q_i, \widetilde{\boldsymbol{v}}) = x_i(\boldsymbol{\theta}, q_i, ((\widetilde{v}_i)^+, \widetilde{\boldsymbol{v}}_{-i})), \quad (v_i)^+(q_i) = v_i(q_i).$$

Therefore, by the Hardy–Littlewood inequality with the version given in Bennett and Sharpley [9], we have

$$\int_0^1 x_i(\boldsymbol{\theta}, q_i, \widetilde{\boldsymbol{v}}) \cdot v_i(q_i) \, \mathrm{d}q_i \le \int_0^1 (x_i)^*(\boldsymbol{\theta}, q_i, \widetilde{\boldsymbol{v}}) \cdot (v_i)^*(q_i) \, \mathrm{d}q_i$$

$$= \int_0^1 (x_i)^+(\boldsymbol{\theta}, q_i, \widetilde{\boldsymbol{v}}) \cdot (v_i)^+(q_i) \, \mathrm{d}q_i$$

$$= \int_0^1 x_i(\boldsymbol{\theta}, q_i, ((\widetilde{v}_i)^+, \widetilde{\boldsymbol{v}}_{-i})) \cdot v_i(q_i) \, \mathrm{d}q_i.$$

Consequently, concerning her expected utility, a buyer's bidding qf $\widetilde{v}_i$ is dominated by its increasing rearrangement $(\widetilde{v}_i)^+$. This finishes the proof of the lemma.

## C   PROOF OF THEOREM 3.1

In the theorem, the first result for BDFPA is presented in Theorem A.1, the results for PFPA are given in Theorem A.3, and the results for symmetric BDSPA and symmetric PSPA are deferred to Lemma D.5, in the context when we concentrate on symmetric cases. We now prove the second result for BDFPA, which states that the seller's revenue is maximized under the budget-extracting condition. In fact, we can relax the inverse Lipschitz continuity condition to strict monotonicity. Specifically, we formally prove the following theorem.

**Theorem C.1.** *When each buyer's bidding qf is strictly increasing, the budget-extracting condition implies the seller's revenue maximization for BDFPA, if it is feasible.*

We devote the remaining to prove Theorem C.1, which follows three steps. First, we give an upper bound on the Lagrangian dual of the revenue-maximizing problem, which happens to be similar to programming (3). Next, we characterize the optimality of the dual problem via a pair of equivalent conditions. With these conditions, we show that strong duality holds. Finally, we relate the above optimality conditions to the budget-extracting condition of BDFPA to prove the theorem.

For briefness, we write $\Phi_i(\boldsymbol{\alpha}, \widetilde{\boldsymbol{v}}, \boldsymbol{q}) \coloneqq I\left[\alpha_i \widetilde{v}_i(q_i) \ge \max_{i' \ne i}\{\alpha_i \widetilde{v}_i(q_{i'}), \lambda\}\right]$ for any $1 \le i \le n$ through the whole appendix, which describes whether $i$ has the largest value $\alpha_i \widetilde{v}_i(q_i)$ no less than $\lambda$ among all buyers. This function is seen as a choice function and fits all five mechanisms' allocation functions with the suitable parameter notation and quantile function (bidding qf or virtual bidding qf).

We now formalize the seller's revenue-maximizing problem in BDFPA as follows and denote the optimal objective as $\mathrm{OPT}^{\mathrm{BDF}}$:

$$\max_{\boldsymbol{\alpha} \in [0,1]^n} \sum_{i=1}^n \int_0^1 (\widetilde{v}_i(q_i) - \lambda) \cdot \left(\int_{\boldsymbol{q}_{-i}} \Phi_i(\boldsymbol{\alpha}, \widetilde{\boldsymbol{v}}, \boldsymbol{q}) \, \mathrm{d}\boldsymbol{q}_{-i}\right) \mathrm{d}q_i,$$

$$\text{s.t.} \quad \int_0^1 \widetilde{v}_i(q_i) \cdot \left(\int_{\boldsymbol{q}_{-i}} \Phi_i(\boldsymbol{\alpha}, \widetilde{\boldsymbol{v}}, \boldsymbol{q}) \, \mathrm{d}\boldsymbol{q}_{-i}\right) \mathrm{d}q_i \le \rho_i, \quad \forall 1 \le i \le n. \tag{5}$$

Now we consider the Lagrangian dual problem of (5), which is as follows:

$$\chi^{\mathrm{BDF}}(\boldsymbol{\tau}) \coloneqq \max_{\boldsymbol{\alpha} \in [0,1]^n} \sum_{i=1}^n \int_0^1 ((1 - \tau_i)\widetilde{v}_i(q_i) - \lambda) \cdot \left(\int_{\boldsymbol{q}_{-i}} \Phi_i(\boldsymbol{\alpha}, \widetilde{\boldsymbol{v}}, \boldsymbol{q}) \, \mathrm{d}\boldsymbol{q}_{-i}\right) \mathrm{d}q_i + \sum_{i=1}^n \tau_i \rho_i. \tag{6}$$

Here, $\tau_i \ge 0$ is the dual variable for the restriction on buyer $i$'s payment. For brevity, we reorganize the integral part of $\chi^{\mathrm{BDF}}(\boldsymbol{\tau})$ as the following:

$$\sum_{i=1}^n \int_0^1 ((1 - \tau_i)\widetilde{v}_i(q_i) - \lambda) \cdot \left(\int_{\boldsymbol{q}_{-i}} \Phi_i(\boldsymbol{\alpha}, \widetilde{\boldsymbol{v}}, \boldsymbol{q}) \, \mathrm{d}\boldsymbol{q}_{-i}\right) \mathrm{d}q_i = \mathbb{E}_{\boldsymbol{q}}\left[(1 - \tau_{i^*})\widetilde{v}_{i^*}(q_{i^*}) - \lambda\right],$$

where for any quantile profile $\boldsymbol{q}$, $i^*$ is defined as the buyer $0 \le i \le n$ with the highest value $\alpha_i \widetilde{v}_i(q_i)$. Here, we involve a phantom buyer 0 with $\alpha_0 = 1$, $\tau_0 = 0$ and $\widetilde{v}_0(q_0) = \lambda$ for all $q_0 \in [0, 1]$.[3] Intuitively, the equation is based on the fact that when $\boldsymbol{q}$ is fixed, functions $\{\Phi_{1 \le i \le n}\}$ collaboratively act as a choice function to pick a buyer $0 \le i^* \le n$ with the highest discounted bid. Rigorously, the above equation establishes

---

[2]Although Lieb and Loss [32] considers the symmetric decreasing rearrangement, these results can be naturally extended to our scenario.

[3]Involving the phantom buyer is only for the succinctness of writing and does not matter with all the restrictions on buyers 1 to $n$.

due to the strict monotonicity condition, as the Lebesgue measure that at least two buyers share the same highest discounted bid is zero. With the equation, we can rewrite $\chi^{\mathrm{BDF}}(\boldsymbol{\tau})$ as:

$$\chi^{\mathrm{BDF}}(\boldsymbol{\tau}) = \max_{\alpha \in [0,1]^n} \mathbb{E}_{\boldsymbol{q}} \left[ (1 - \tau_{i^*}) \widetilde{v}_{i^*}(q_{i^*}) - \lambda \right] + \sum_{i=1}^{n} \tau_i \rho_i. \tag{7}$$

Now, for fixed $\boldsymbol{\tau} \in [0,1]^n$, since $i^*$ represents a specific buyer from 0 to $n$, we have

$$\chi^{\mathrm{BDF}}(\boldsymbol{\tau}) \leq \mathbb{E}_{\boldsymbol{q}} \left[ \max_{1 \leq i \leq n} \{(1 - \tau_i) \widetilde{v}_i(q_i) - \lambda\}^+ \right] + \sum_{i=1}^{n} \tau_i \rho_i,$$

and further, we can see that the equality holds if $\boldsymbol{\tau} \in [0,1]^n$ and we take $\boldsymbol{\alpha} = 1^n - \boldsymbol{\tau}$ in (7), where $i^*$ is subsequently $\arg\max_{0 \leq i \leq n} \{(1 - \tau_i) \widetilde{v}_i(q_i) - \lambda\}$. Now by weak duality, we have

$$\mathrm{OPT}^{\mathrm{BDF}} \leq \min_{\boldsymbol{\tau} \geq 0} \chi^{\mathrm{BDF}}(\boldsymbol{\tau}) \leq \min_{\boldsymbol{\tau} \in [0,1]^n} \chi^{\mathrm{BDF}}(\boldsymbol{\tau}) = \min_{\boldsymbol{\tau} \in [0,1]^n} \mathbb{E}_{\boldsymbol{q}} \left[ \max_{1 \leq i \leq n} \{(1 - \tau_i) \widetilde{v}_i(q_i) - \lambda\}^+ \right] + \sum_{i=1}^{n} \tau_i \rho_i. \tag{8}$$

We now characterize the optimal solution of the program via the following lemma.

**Lemma C.2.** *When each buyer $1 \leq i \leq n$'s bidding qf $\widetilde{v}_i$ is strictly increasing, $\boldsymbol{\tau}$ is a solution of $\min_{\boldsymbol{\tau} \in [0,1]^n} \chi^{\mathrm{BDF}}(\boldsymbol{\tau})$ if and only if for each $1 \leq i \leq n$:*

1. *$\int_0^1 \widetilde{v}_i(q_i) \cdot \left( \int_{\boldsymbol{q}_{-i}} \Phi_i(1^n - \boldsymbol{\tau}, \widetilde{\boldsymbol{v}}, \boldsymbol{q}) \, \mathrm{d}\boldsymbol{q}_{-i} \right) \mathrm{d}q_i \leq \rho_i$, and*
2. *$\tau_i \cdot \left( \int_0^1 \widetilde{v}_i(q_i) \cdot \left( \int_{\boldsymbol{q}_{-i}} \Phi_i(1^n - \boldsymbol{\tau}, \widetilde{\boldsymbol{v}}, \boldsymbol{q}) \, \mathrm{d}\boldsymbol{q}_{-i} \right) \mathrm{d}q_i - \rho_i \right) = 0$.*

PROOF OF LEMMA C.2. We first give some temporary notations to ease the description. We let $p_i := \int_0^1 \widetilde{v}_i(q_i) \cdot \left( \int_{\boldsymbol{q}_{-i}} \Phi_i(1^n - \boldsymbol{\tau}, \widetilde{\boldsymbol{v}}, \boldsymbol{q}) \, \mathrm{d}\boldsymbol{q}_{-i} \right) \mathrm{d}q_i$ be buyer $i$'s expected payment. We further write $y_0(\boldsymbol{\tau}, \boldsymbol{q}) := 0$ and $y_i(\boldsymbol{\tau}, \boldsymbol{q}) := (1 - \tau_i) \widetilde{v}_i(q_i) - \lambda$ for $1 \leq i \leq n$. At last, we let $y(\boldsymbol{\tau}, \boldsymbol{q}) = \max_{0 \leq i \leq n} y_i(\boldsymbol{\tau}, \boldsymbol{q})$. We now have

$$\chi^{\mathrm{BDF}}(\boldsymbol{\tau}) = \mathbb{E}_{\boldsymbol{q}} \left[ y(\boldsymbol{\tau}, \boldsymbol{q}) \right] + \sum_{i=1}^{n} \tau_i \rho_i.$$

Note that for any $0 \leq i \leq n$, $y_i(\boldsymbol{\tau}, \boldsymbol{q})$ is convex on $\boldsymbol{\tau}$. As a result, $y(\boldsymbol{\tau}, \boldsymbol{q})$ is also convex on $\boldsymbol{\tau}$. Meanwhile, let $\mathcal{J}(\boldsymbol{\tau}, \boldsymbol{q}) = \arg\max_{0 \leq i \leq n} y_i(\boldsymbol{\tau}, \boldsymbol{q})$, then with probability 1, $|\mathcal{J}(\boldsymbol{\tau}, \boldsymbol{q})| = 1$ when $\boldsymbol{q}$ is chosen uniformly from $[0,1]^n$, and therefore, $y(\boldsymbol{\tau}, \boldsymbol{q})$ is differentiable with probability 1. By Theorem 7.46 from Shapiro et al. [37], $\chi^{\mathrm{BDF}}(\boldsymbol{\tau})$ is convex and differentiable. Further by Theorem 7.44 from Shapiro et al. [37],

$$\begin{aligned}
\frac{\partial}{\partial \tau_i} \chi^{\mathrm{BDF}}(\boldsymbol{\tau}) &= \mathbb{E}_{\boldsymbol{q}} \left[ \frac{\partial}{\partial \tau_i} y(\boldsymbol{\tau}, \boldsymbol{q}) \right] + \rho_i \\
&= \mathbb{E}_{\boldsymbol{q}} \left[ -\widetilde{v}_i(q_i) I \left[ i \in \mathcal{J}(\boldsymbol{\tau}, \boldsymbol{q}) \right] \right] + \rho_i \\
&= -p_i + \rho_i.
\end{aligned}$$

Therefore, $\nabla \chi^{\mathrm{BDF}}(\boldsymbol{\tau}) = (-p_i + \rho_i)_{1 \leq i \leq n}$. As $\chi^{\mathrm{BDF}}(\boldsymbol{\tau})$ is convex, $\boldsymbol{\tau}$ is optimal if and only if for any $\tau' \in [0,1]^n$,

$$\nabla \chi^{\mathrm{BDF}}(\boldsymbol{\tau}) \cdot (\boldsymbol{\tau}' - \boldsymbol{\tau}) \geq 0. \tag{9}$$

We now finish the proof of the lemma.

*"If" side.* Suppose the given two conditions are satisfied. Let $\mathcal{K}(\boldsymbol{\tau}) = \{i \mid \tau_i = 0\}$. For any $\boldsymbol{\tau}' \in [0,1]^n$, if $i \notin \mathcal{K}(\boldsymbol{\tau})$, then $-p_i + \rho_i = 0$, and $\nabla_i \chi^{\mathrm{BDF}}(\boldsymbol{\tau}) \cdot (\tau'_i - \tau_i) = 0$ holds. Otherwise, $-p_i + \rho_i \leq 0$ and $\tau'_i \geq \tau_i$, which leads to $\nabla_i \chi^{\mathrm{BDF}}(\boldsymbol{\tau}) \cdot (\tau'_i - \tau_i) \geq 0$. As a result, $\nabla \chi^{\mathrm{BDF}}(\boldsymbol{\tau}) \cdot (\boldsymbol{\tau}' - \boldsymbol{\tau}) \geq 0$ and $\boldsymbol{\tau}$ is optimal.

*"Only if" side.* To start with, we claim that all entries of any optimal solution $\boldsymbol{\tau}^*$ of the programming are strictly smaller than 1. To see this, counterfactually suppose $\tau_i^* = 1$, then $(1 - \tau_i^*) \widetilde{v}_i(q_i) - \lambda < 0$ holds for any $q_i$ as $\lambda > 0$. Therefore, since $\widetilde{v}_i(\cdot)$ is bounded (by 1), subtracting $\tau_i^*$ by a small amount does not affect the expectation part of $\chi^{\mathrm{BDF}}(\boldsymbol{\tau}^*)$, but will strictly lessen the latter sum part $\sum_{i=1}^{n} \tau_i \rho_i$, contradicting the optimality.

Now, for some optimal $\boldsymbol{\tau}^*$, let $\mathcal{K}(\boldsymbol{\tau}^*) = \{i \mid \tau_i^* = 0\}$. For $i \in \mathcal{K}(\boldsymbol{\tau}^*)$, let $\boldsymbol{\tau}' = \boldsymbol{\tau}^* + \delta \boldsymbol{e}_i$ for some $\delta \in (0,1]$, where $\boldsymbol{e}_i$ is the vector with the $i$-th entry one and all other entries zero. Plugging in (9), we derive that $p_i^* \leq \rho_i$. For $i \notin \mathcal{K}(\boldsymbol{\tau}^*)$, we take $\boldsymbol{\tau}' = \boldsymbol{\tau}^* \pm \delta \boldsymbol{e}_i$ in order for some small $\delta > 0$ satisfying $\tau_i^* \pm \delta \in [0,1]$. Such $\delta$ exists since $0 < \tau_i^* < 1$. Taking into (9) respectively, we derive that $p_i^* - \rho_i = 0$. Lemma C.2 is proved. □

We show that strong duality holds with Lemma C.2. For the solution $\boldsymbol{\tau}^*$ which minimizes $\chi^{\text{BDF}}(\boldsymbol{\tau})$ in $[0,1]^n$, we have

$$
\begin{aligned}
\chi^{\text{BDF}}(\boldsymbol{\tau}^*) &= \mathbb{E}_{\boldsymbol{q}}\left[\max_{1 \le i \le n}\left\{(1-\tau_i^*)\widetilde{v}_i(q_i) - \lambda\right\}^+\right] + \sum_{i=1}^n \tau_i^* \rho_i \\
&= \int_0^1 \left((1-\tau_i^*)\widetilde{v}_i(q_i) - \lambda\right) \cdot \left(\int_{\boldsymbol{q}_{-i}} \Phi_i(1^n - \boldsymbol{\tau}^*, \widetilde{\boldsymbol{v}}, \boldsymbol{q})\,\mathrm{d}\boldsymbol{q}_{-i}\right)\mathrm{d}q_i + \sum_{i=1}^n \tau_i^* \rho_i \\
&= \int_0^1 (\widetilde{v}_i(q_i) - \lambda) \cdot \left(\int_{\boldsymbol{q}_{-i}} \Phi_i(1^n - \boldsymbol{\tau}^*, \widetilde{\boldsymbol{v}}, \boldsymbol{q})\,\mathrm{d}\boldsymbol{q}_{-i}\right)\mathrm{d}q_i.
\end{aligned}
$$

Here, the first equation is by definition and $\boldsymbol{\tau}^* \in [0,1]^n$, and the last equation is by the first condition in Lemma C.2. Let $\boldsymbol{\alpha}^* = 1^n - \boldsymbol{\tau}^*$. Now by the second condition in Lemma C.2, $\boldsymbol{\alpha}^*$ satisfies all budget constraints in (5), with the objective value $\chi^{\text{BDF}}(\boldsymbol{\tau}^*)$. As a result, strong duality holds, and $\boldsymbol{\alpha}^*$ is the revenue-maximizing tuple of bid-discount multipliers.

Finally, we come to prove the theorem. For a budget-extracting bid-discount multiplier tuple $\boldsymbol{\alpha}^{\text{e}}$, by definition, the following two groups of constraints hold:

$$
\int_0^1 \widetilde{v}_i(q_i) \cdot \left(\int_{\boldsymbol{q}_{-i}} \Phi_i(\boldsymbol{\alpha}^{\text{e}}, \widetilde{\boldsymbol{v}}, \boldsymbol{q})\,\mathrm{d}\boldsymbol{q}_{-i}\right)\mathrm{d}q_i \le \rho_i, \quad \forall 1 \le i \le n.
$$

$$
\alpha_i^{\text{e}} \cdot \left(\rho_i - \int_0^1 \widetilde{v}_i(q_i) \cdot \left(\int_{\boldsymbol{q}_{-i}} \Phi_i(\boldsymbol{\alpha}^{\text{e}}, \widetilde{\boldsymbol{v}}, \boldsymbol{q})\,\mathrm{d}\boldsymbol{q}_{-i}\right)\mathrm{d}q_i\right) = 0, \quad \forall 1 \le i \le n. \tag{10}
$$

By Lemma C.2, $\boldsymbol{\tau}^* = 1^n - \boldsymbol{\alpha}^{\text{e}}$ is an optimal solution of $\min_{\boldsymbol{\tau} \in [0,1]^n} \chi^{\text{BDF}}(\boldsymbol{\tau})$, and by strong duality, $\boldsymbol{\alpha}^{\text{e}} = 1^n - \boldsymbol{\tau}^*$ is revenue-maximizing, which finishes the proof of Theorem C.1.

# D  PROOFS IN SECTION 4

## D.1  Proof of Lemma 4.1

If $\widetilde{\boldsymbol{v}} = (\widetilde{v}_i)_{i \in [n]}$ is a Nash equilibrium strategy profile for $\mathcal{M}_1(\boldsymbol{\theta})$, then we argue that $(g(\widetilde{v}_i))_{i \in [n]}$ forms a Nash equilibrium for $\mathcal{M}_2(\boldsymbol{\theta})$. Or else, if a buyer $i$ can strictly increase her utility from $u_i$ to $u_i'$ by switching $g(\widetilde{v}_i)$ to $\widetilde{v}_i'$, then since $h = g^{-1}$ and the property of the mapping, we derive that her utility with $(g^{-1}(\widetilde{v}_i'), \widetilde{v}_{-i})$ under $\mathcal{M}_1(\boldsymbol{\theta})$ is $u_i'$. Meanwhile, buyer $i$'s utility with $(\widetilde{v}_i', \widetilde{v}_{-i})$ under $\mathcal{M}_1(\boldsymbol{\theta})$ is $u_i < u_i'$, contradicting that $(\widetilde{v}_i)_{i \in [n]}$ implies a Nash equilibrium. The reverse direction from $\mathcal{M}_2(\boldsymbol{\theta})$ to $\mathcal{M}_1(\boldsymbol{\theta})$ is similar. This finishes the proof of the lemma.

## D.2  Proof of Theorem 4.2

We will make some preparations before we come to the main part of the proof.

We first discuss on the optimal tuple $\boldsymbol{\gamma}^*$ in BROA. By Lemma C.2, since the virtual bidding qf is strictly increasing, programming (3) is equivalent to the following conditions:

$$
\int_0^1 \widetilde{\psi}_i(q_i) \cdot \left(\int_{\boldsymbol{q}_{-i}} \Phi_i(\boldsymbol{\gamma}, \widetilde{\boldsymbol{\psi}}, \boldsymbol{q})\,\mathrm{d}\boldsymbol{q}_{-i}\right)\mathrm{d}q_i \le \rho_i, \quad \forall 1 \le i \le n.
$$

$$
(1 - \gamma_i) \cdot \left(\rho_i - \int_0^1 \widetilde{\psi}_i(q_i) \cdot \left(\int_{\boldsymbol{q}_{-i}} \Phi_i(\boldsymbol{\gamma}, \widetilde{\boldsymbol{\psi}}, \boldsymbol{q})\,\mathrm{d}\boldsymbol{q}_{-i}\right)\mathrm{d}q_i\right) = 0, \quad \forall 1 \le i \le n. \tag{11}
$$

Note that the second multiplying term in the second set of constraints represents the expected remaining budget of each buyer. Therefore, Lemma C.2 shows that as long as buyer $i$'s budget is not binding, $\gamma_i^* = 0$ establishes.

Meanwhile, for a better look, we restate here the equivalence conditions of the multiplier tuple of the budget-extracting BDFPA according to Theorem C.1:

$$
\int_0^1 \widetilde{v}_i(q_i) \cdot \left(\int_{\boldsymbol{q}_{-i}} \Phi_i(\boldsymbol{\alpha}, \widetilde{\boldsymbol{v}}, \boldsymbol{q})\,\mathrm{d}\boldsymbol{q}_{-i}\right)\mathrm{d}q_i \le \rho_i, \quad \forall 1 \le i \le n.
$$

$$
(1 - \alpha_i) \cdot \left(\rho_i - \int_0^1 \widetilde{v}_i(q_i) \cdot \left(\int_{\boldsymbol{q}_{-i}} \Phi_i(\boldsymbol{\alpha}, \widetilde{\boldsymbol{v}}, \boldsymbol{q})\,\mathrm{d}\boldsymbol{q}_{-i}\right)\mathrm{d}q_i\right) = 0, \quad \forall 1 \le i \le n. \tag{12}
$$

We are now ready to prove our main theorem. We suppose that the bidding qf profile in BROA is $\widetilde{\boldsymbol{v}}^{(1)}$, while the counterpart in eBDFPA is $\widetilde{\boldsymbol{v}}^{(2)}$. Balseiro et al. [3] characterizes any buyer's expected payment in BROA in the following proposition:

**Proposition D.1** (in Balseiro et al. [3]). *For BROA, any buyer $i$'s expected payment satisfies:*

$$
\mathbb{E}_{q_i}\left[p_i(\boldsymbol{\gamma}^*, q_i, \widetilde{\boldsymbol{v}})\right] = \mathbb{E}_{q_i}\left[\widetilde{\psi}_i(q_i) \cdot x_i(\boldsymbol{\gamma}^*, q_i, \widetilde{\boldsymbol{v}})\right].
$$

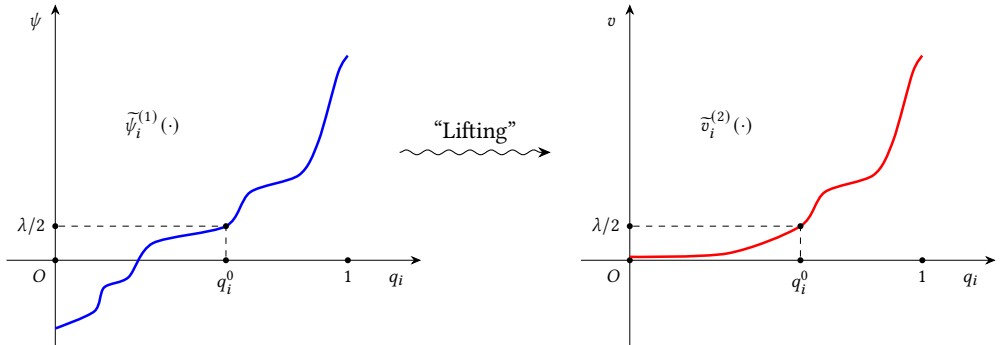

**Figure 4: The "lifting" process, which is adopted to construct $\widetilde{v}_i^{(2)}$ when $\widetilde{\psi}_i^{(1)}$ has negative parts.**

With the help of Proposition D.1, we give the expected utility of any buyer $1 \le i \le n$ under both auction mechanisms in the following:

$$u_i^{\mathrm{BRO}}(\boldsymbol{\gamma}^*, \widetilde{\boldsymbol{v}}^{(1)}, v_i) = \int_0^1 \left( v_i(q_i) - \widetilde{\psi}_i^{(1)}(q_i) \right) \cdot \left( \int_{\boldsymbol{q}_{-i}} \Phi_i(\boldsymbol{\gamma}^*, \widetilde{\boldsymbol{\psi}}^{(1)}, \boldsymbol{q}) \, \mathrm{d}\boldsymbol{q}_{-i} \right) \mathrm{d}q_i. \tag{13}$$

$$u_i^{\mathrm{eBDF}}(\boldsymbol{\alpha}^{\mathrm{e}}, \widetilde{\boldsymbol{v}}^{(2)}, v_i) = \int_0^1 \left( v_i(q_i) - \widetilde{v}_i^{(2)}(q_i) \right) \cdot \left( \int_{\boldsymbol{q}_{-i}} \Phi_i(\boldsymbol{\alpha}^{\mathrm{e}}, \widetilde{\boldsymbol{v}}^{(2)}, \boldsymbol{q}) \, \mathrm{d}\boldsymbol{q}_{-i} \right) \mathrm{d}q_i. \tag{14}$$

Further, the seller's expected revenue in these two mechanisms are correspondingly the following:

$$w^{\mathrm{BRO}}(\boldsymbol{\gamma}^*, \widetilde{\boldsymbol{v}}^{(1)}) = \sum_{i=1}^n \left( \int_0^1 \left( \widetilde{\psi}_i^{(1)}(q_i) - \lambda \right) \cdot \left( \int_{\boldsymbol{q}_{-i}} \Phi_i(\boldsymbol{\gamma}^*, \widetilde{\boldsymbol{\psi}}^{(1)}, \boldsymbol{q}) \, \mathrm{d}\boldsymbol{q}_{-i} \right) \mathrm{d}q_i \right). \tag{15}$$

$$w^{\mathrm{eBDF}}(\boldsymbol{\alpha}^{\mathrm{e}}, \widetilde{\boldsymbol{v}}^{(2)}) = \sum_{i=1}^n \left( \int_0^1 \left( \widetilde{v}_i^{(2)}(q_i) - \lambda \right) \cdot \left( \int_{\boldsymbol{q}_{-i}} \Phi_i(\boldsymbol{\alpha}^{\mathrm{e}}, \widetilde{\boldsymbol{v}}^{(2)}, \boldsymbol{q}) \, \mathrm{d}\boldsymbol{q}_{-i} \right) \mathrm{d}q_i \right). \tag{16}$$

We now start our main part of the proof.

*From BROA to eBDFPA.* In this part, recall that the virtual bidding qf $\widetilde{\psi}_i^{(1)}$ is strictly increasing, differentiable, and inverse Lipschitz continuous for any $1 \le i \le n$. Without loss of generality, we suppose that $\widetilde{\psi}_i^{(1)}(1) \ge \lambda$ holds for any $1 \le i \le n$, or that the corresponding buyer has no chance to win any item at all. For the injective mapping, we let $\widetilde{v}_i^{(2)}$ to be a modification of $\widetilde{\psi}_i^{(1)}$ for any $1 \le i \le n$. Specifically, if $\widetilde{\psi}_i^{(1)}(0) \ge 0$, we easily let $\widetilde{v}_i^{(2)} = \widetilde{\psi}_i^{(1)}$. Otherwise, we will "lift" the negative part $\widetilde{\psi}_i^{(1)}$ so that the resulting function is non-negative, with no loss on other required properties.

In particular, we construct $\widetilde{v}_i^{(2)}$ by replacing the head part of $\widetilde{\psi}_i^{(1)}$ with an exponential function or a trigonometric function, depending on whether $\widetilde{\psi}_i^{(1)}$ is flat at the joint point. More specifically, let $q_i^0 \in (0, 1)$ be the point satisfying $\widetilde{\psi}_i^{(1)}(q_i^0) = \lambda/2$ (we will argue later that such $q_i^0$ exists in the proof of Lemma D.2), and $k := \left( \widetilde{\psi}_i^{(1)} \right)'(q_i^0) \ge 0$. If $k > 0$, let $\widetilde{v}_i^{(2)}$ be as follows:

$$\widetilde{v}_i^{(2)}(q_i) := \begin{cases} a_1 \cdot \exp\{a_2 q_i\} & 0 \le q_i < q_i^0 \\ \widetilde{\psi}_i^{(1)}(q_i) & q_i^0 \le q_i \le 1, \end{cases}$$

where $a_1 = \lambda \cdot \exp\{-2kq_i^0/\lambda\}/2$ and $a_2 = 2k/\lambda$. Such "lifting" process is plotted in Figure 4. Otherwise when $k = 0$, we use a trigonometric function instead to "lift" the negative part and define $\widetilde{v}_i^{(2)}$ as:

$$\widetilde{v}_i^{(2)}(q_i) := \begin{cases} (\lambda/4) \cdot \sin(a_3 q_i + \pi/4) + \lambda/4 & 0 \le q_i < q_i^0 \\ \widetilde{\psi}_i^{(1)}(q_i) & q_i^0 \le q_i \le 1, \end{cases}$$

where $a_3 = \pi/(4q_i^0)$.

Apparently, the vale range of $\widetilde{v}_i^{(2)}$ as constructed above lies within $[0, 1]$. In the following lemma, we show that the above injective mapping guarantees that $\widetilde{v}_i^{(2)}$ is non-negative, strictly increasing, and differentiable. Inverse Lipschitz continuity also remains after the mapping for any $1 \le i \le n$.

**Lemma D.2.** *Suppose that $\widetilde{\psi}_i^{(1)}$ is strictly increasing and differentiable for some $i$, then under the mapping given above, $\widetilde{v}_i^{(2)}$ also satisfies these properties, and is further non-negative. Meanwhile, inverse Lipschitz continuity is also kept under the mapping.*

PROOF OF LEMMA D.2. In the first case when $\widetilde{\psi}_i^{(1)}(x)$ is non-negative on $[0, 1]$, the lemma trivially holds. For the second case, by continuity and that $\widetilde{\psi}_i^{(1)}(0) < 0, \widetilde{\psi}_i^{(1)}(1) \geq \lambda$, the point $q_i^0 \in (0, 1)$ such that $\widetilde{\psi}_i^{(1)}(q_i^0) = \lambda/2$ exists. By construction, $\widetilde{v}_i^{(2)}(x)$ is certainly non-negative and continuous. We further show that $\widetilde{v}_i^{(2)}(x)$ is differentiable, which clearly, reduces to demonstrate the function is differentiable at $q_i^0$. We can verify this by direct computation that $\left(\widetilde{v}_i^{(2)}\right)'(q_i^0) = k$ whether $k > 0$ or $k = 0$. Strictly increasing monotonicity follows from that $\widetilde{\psi}_i^{(1)}(x)$ is strictly increasing, and $a_1 \exp\{a_2 x\}$ (when $k > 0$) and $(\lambda/4)\sin(a_3 x + \pi/4) + \lambda/4$ (when $k = 0$) are both strictly increasing on $[0, q_i^0]$.

Furthermore, when inverse Lipschitz continuity holds for $\widetilde{\psi}_i^{(1)}$, then $\left(\widetilde{\psi}_i^{(1)}\right)'$ has an upper bound strictly higher than 0 by definition, which leads to $k > 0$. In this case, $a_1 \exp\{a_2 x\}$ is also inverse Lipschitz continuous on $[0, q_i^0]$ as $a_1, a_2$ are both constants. As a result, $\widetilde{v}_i^{(2)}(x)$ is inverse Lipschitz continuous as well. □

By Lemma D.2, $\widetilde{v}^{(2)}$ is a valid qf profile. We now let $\boldsymbol{\alpha} = \boldsymbol{\gamma}^*$, and argue that $\boldsymbol{\alpha}$ is a budget-extracting tuple for BDFPA.

In fact, noticing that under the injective mapping we give above, $\widetilde{v}_i^{(2)}(q_i) = \widetilde{\psi}_i^{(1)}(q_i)$ always holds when the value is above $\lambda/2$. At the same time, due to the threshold effect brought by the opportunity cost $\lambda$ and that $\boldsymbol{\alpha} \leq 1^n$, we derive that for any quantile profile $\boldsymbol{q}$, we have

$$\int_{\boldsymbol{q}_{-i}} \Phi_i(\boldsymbol{\gamma}^*, \widetilde{\boldsymbol{\psi}}^{(1)}, \boldsymbol{q}) \, d\boldsymbol{q}_{-i} = \int_{\boldsymbol{q}_{-i}} \Phi_i(\boldsymbol{\alpha}, \widetilde{\boldsymbol{v}}^{(2)}, \boldsymbol{q}) \, d\boldsymbol{q}_{-i}$$

holds for any $1 \leq i \leq n$ and $q_i \in [0, 1]$. Meanwhile, the above equals 0 when $q_i \leq q_i^0$. Hence, $\boldsymbol{\alpha} = \boldsymbol{\gamma}^*$ leads to a budget-extracting BDFPA. Note that buyers' utility and the seller's revenue in BROA and budget-extracting BDFPA are given by (13), (14), (15), and (16) respectively. Therefore, in any BROA with $\widetilde{v}^{(1)}$, there is some budget-extracting BDFPA with $\widetilde{v}^{(2)}$, such that for the any buyer's utilities and the seller's revenues are the same in the two auctions.

*From eBDFPA to BROA.* In this part, we show that for any budget-extracting BDFPA with $\widetilde{v}^{(2)}$, there is some BROA with $\widetilde{v}^{(1)}$, such that every buyer's revenue is identical in the two auctions.

For the mapping from $\widetilde{v}^{(2)}$ to $\widetilde{v}^{(1)}$, we hope to have $\widetilde{\psi}_i^{(1)} = \widetilde{v}_i^{(2)}$ for any $1 \leq i \leq n$. In other words, we carefully pick $\widetilde{v}_i^{(1)}$ such that for any $q_i \in [0, 1]$, $\widetilde{v}_i^{(1)}(q_i) - (1 - q_i)\left(\widetilde{v}_i^{(1)}\right)'(q_i) = \widetilde{v}_i^{(2)}(q_i)$. The following lemma shows that such a function $\widetilde{v}_i^{(1)}$ exists.

**Lemma D.3.** *For each strictly increasing and differentiable function $r(x)$ on $[0, 1]$ that satisfies $0 < r(1) \leq 1$ and $\int_0^1 r(x) \, dx \geq 0$, there exists a non-negative, strictly increasing and differentiable function $s(x)$ on $[0, 1]$ such that $r(x) = s(x) - (1 - x)s'(x)$ and $s(x) \leq 1$ hold for any $x \in [0, 1]$. Meanwhile, inverse Lipschitz continuity is also kept under the mapping.*

PROOF OF LEMMA D.3. We define the function $s(x)$ as follows: $s(1) = r(1)$, and when $x < 1$, $s(x) = (\int_x^1 r(z) \, dz)/(1 - x)$. Since $r(z) \leq 1$ for $t \in [0, 1]$, $s(x) \leq 1$ for each $x$. Further, differentiability of $s$ holds naturally. We now show that $s$ satisfies that $r(x) = s(x) - (1 - x)s'(x)$ for any $x \in [0, 1]$. In fact, the equality obviously holds when $x = 1$, and when $x < 1$, we have

$$s(x) - (1 - x)s'(x) = \frac{\int_x^1 r(z) \, dz}{1 - x} - (1 - x)\left(\frac{\int_x^1 r(z) \, dz}{1 - x}\right)'$$

$$= \frac{\int_x^1 r(z) \, dz}{1 - x} - (1 - x)\left(\frac{-r(x)(1 - x) + \int_x^1 r(z) \, dz}{(1 - x)^2}\right) = r(x).$$

Further, $s(x) = (\int_x^1 r(z) \, dz)/(1 - x)$ is non-negative and strictly increasing on $[0, 1]$, which establishes as $r(z)$ is strictly increasing and $\int_0^1 r(x) \, dx \geq 0$.

It remains to show that $s$ is inverse Lipschitz continuous when $r$ is. To see this, we do a derivation on $s$,

$$s'(x) = \left(\frac{\int_x^1 r(z) \, dz}{1 - x}\right)' = \frac{-r(x)(1 - x) + \int_x^1 r(z) \, dz}{(1 - x)^2} = \frac{\int_x^1 (r(z) - r(x)) \, dz}{(1 - x)^2}.$$

By strict monotonicity and inverse Lipschitz continuity of $r$, there exists a positive constant $L$, such that $r(z) - r(x) \geq L(t - x)$ for any $0 \leq z < x \leq 1$. Therefore,

$$s'(x) = \frac{\int_x^1 (r(z) - r(x)) \, dz}{(1 - x)^2} \geq L \cdot \frac{\int_x^1 (z - x) \, dz}{(1 - x)^2} = L/2 > 0,$$

which indicates that $s$ is inverse Lipschitz continuous as well. □

Lemma D.3 demonstrates the feasibility of $\widetilde{v}_i^{(1)}$. Now we let $\gamma = \alpha^{\mathrm{e}}$ with the latter a budget-extracting tuple for BDFPA. By the mapping, we have

$$\int_{q_{-i}} \Phi_i(\gamma, \widetilde{\psi}^{(1)}, q)\, \mathrm{d}q_{-i} = \int_{q_{-i}} \Phi_i(\alpha^{\mathrm{e}}, \widetilde{v}^{(2)}, q)\, \mathrm{d}q_{-i}.$$

Further, since $\widetilde{\psi}_i^{(1)} = \widetilde{v}_i^{(2)}$, $\gamma$ is feasible for (11). Therefore, by Lemma C.2, $\gamma$ is optimal for programming (3). As a result, by (13), (14), (15), and (16), the proof of this direction is finished.

Synthesizing the above two parts, we finish the proof of the essential theorem.

## D.3 Proof of Theorem 4.3

To prove the theorem, we first show that for BROA, any buyer $i$'s bidding strategy is equivalent to one with non-negative virtual values on $[0, 1]$. Hence, we can reduce our discussion to "well-behaved" BROA instances and apply Lemma 4.1 to finish the proof. On this side, we have the following lemma.

**Lemma D.4.** *In BROA, given that each buyer's virtual bidding function is strictly increasing and differentiable, then each buyer $i$'s bidding strategy is equivalent to a bidding strategy with non-negative virtual values on $[0, 1]$ regardless of other buyers' strategies. The same result also holds when the virtual bidding function is further constrained to be inverse Lipschitz continuous.*

PROOF OF LEMMA D.4. We only need to consider the case when a buyer's virtual bidding strategy is negative at some interval. In this case, we use the "lifting" technique as considered in Lemma D.2 to map her virtual bidding function to another non-negative function, keeping the properties of strict monotonicity, differentiability, and inverse Lipschitz continuity. As shown in Lemma D.3, the constructed function is a valid virtual bidding function. Clearly, these two functions are equivalent to the buyer in BROA since they coincide when the virtual value is above $\lambda/2$. □

With Lemma D.4, we can only consider BROA instances with positive virtual valuations. Consequently, we can construct the mappings for strong strategic equivalence as follows:

- For a bidding strategy $\widetilde{v}$ in BROA, let $g(\widetilde{v})(x) = \widetilde{v}(x) - (1-x)\widetilde{v}'(x)$.
- For a bidding strategy $\widetilde{v}$ in eBDFPA, let $h(\widetilde{v})(x) = (\int_x^1 \widetilde{v}(z)\, \mathrm{d}z)/(1-x)$.

By the proof of Theorem 4.2, we can derive that $g$ and $h$ construct the strong strategic equivalence mappings under the assumptions. In addition, by Lemma D.3 and Newton-Leibniz theorem, we obtain that $g$ and $h$ are inverse functions of each other. Therefore, applying Lemma 4.1, we finish the proof of the theorem.

## D.4 Proof of Theorem 4.4

To start the proof, we first show that in the symmetric case, each of the four auction mechanisms we consider admits a symmetric budget-extracting tuple. This result, given in Lemma D.5, sets a preliminary for the theorem. We also notice the cases for BDSPA and PSPA complete the last part of Theorem 3.1.

**Lemma D.5.** *In the symmetric case, and when each buyer's bidding qf is inverse Lipschitz continuous, there is a corresponding symmetric budget-extracting multiplier tuple for BDFPA, PFPA, BDSPA, and PSPA. Further, for PSPA, committing to a budget-extracting mechanism maximizes the seller's revenue among all symmetric PSPA mechanisms.*

PROOF OF LEMMA D.5. We prove the results for these four mechanisms one by one. For simplicity, we suppose that all buyers' common bidding qf is $\widetilde{v}_0$, and their common budget is $\rho_0$.

*For BDFPA and PFPA.* We prove this by contradiction. By Theorem A.1, there exists a maximum budget-extracting tuple $\alpha^{\max}$ for BDFPA. Suppose otherwise $\alpha^{\max}$ is not symmetric. Then again by Theorem A.1 and symmetry, $(\max(\alpha^{\max}))^n$ is also budget-extracting, which contradicts the optimality of $\alpha^{\max}$. The case for PFPA is similar concerning Theorem A.3.

*For BDSPA.* Under the given conditions, when the multiplier tuple is symmetric (suppose, to be $(\mu_0)^n$), the payment of each buyer is the following:

$$p^{\mathrm{BDS}}((\mu_0)^n, \widetilde{v}_0) = \int_{[0,1]^n} \max_{i' \neq i} \left\{ \widetilde{v}_0(q_{i'}), \frac{\lambda}{\mu_0} \right\} \cdot \Phi_i((\mu_0)^n, \widetilde{v}_0, q)\, \mathrm{d}q,$$

which certainly, is an increasing Lipschitz continuous function of $\mu_0$ when $\mu_0 > \lambda/\widetilde{v}_0(1) > 0$. Therefore, if $\sup\{p^{\mathrm{BDS}}((\mu_0)^n, \widetilde{v}_0) \mid \mu_0 \in [0,1]\} < \rho_0$, then by definition, $1^n$ is the unique symmetric budget-extracting tuple. Otherwise, there exists some $0 < \mu_0^* < 1$ such that $p^{\mathrm{BDS}}((\mu_0)^n, \widetilde{v}_0) = \rho_0$ by continuity, from which we derive that $(\mu_0^*)^n$ is a symmetric budget-extracting tuple.

*For PSPA.* At last, for PSPA, we notice that with symmetric multipliers $(\xi_0)^n$, the payment of each buyer is

$$p^{\text{PS}}((\xi_0)^n, \widetilde{\boldsymbol{v}}_0) = \int_{[0,1]^n} \max_{i' \neq i} \{\xi_0 \widetilde{v}_0(q_{i'}), \lambda\} \cdot \Phi_i((\xi_0)^n, \widetilde{\boldsymbol{v}}_0, \boldsymbol{q}) \, \mathrm{d}\boldsymbol{q},$$

and is increasingly Lipschitz continuous on $\xi_0$. Therefore, if $p^{\text{PS}}(1) < \rho_0$, then by definition, $1^n$ is the unique symmetric budget-extracting tuple. Otherwise, there exists some $0 < \xi_0^* < 1$ such that $p^{\text{PS}}(\xi_0^*, \widetilde{v}_0) = \rho_0$ by continuity, which indicates that $(\xi_0^*)^n$ is a budget-extracting tuple.

Further, the seller's expected revenue under PSPA is:

$$w^{\text{PS}}((\xi_0)^n, \widetilde{\boldsymbol{v}}_0) = \sum_{i=1}^{n} \left( \int_0^1 \left( \max_{i' \neq i} \{\xi_0 \widetilde{v}_0(q_{i'}), \lambda\} - \lambda \right) \cdot \left( \int_{\boldsymbol{q}_{-i}} \Phi_i((\xi_0)^n, \widetilde{\boldsymbol{v}}_0, \boldsymbol{q}) \, \mathrm{d}\boldsymbol{q}_{-i} \right) \mathrm{d}q_i \right). \tag{17}$$

Clearly, this is an increasing function of $\xi_0$. Combining that the payment function is also increasing of $\xi_0$, we derive that a budget-extracting symmetric PSPA maximizes the seller's revenue among all symmetric PSPAs.

Here, we should mention that the results for PFPA are also given in Balseiro et al. [3]. □

For the rest of the proof, we mainly focus on two parts, respectively, on the strategic equivalence between eBDFPA and ePFPA, and between eBDFPA and eBDSPA. The case between ePFPA and ePSPA is similar to the latter, which we omit. Therefore, we are reduced to the following two theorems.

**Theorem D.6.** *In the symmetric case, when all buyers' identical bidding qf is inverse Lipschitz continuous, under the symmetric budget-extracting parameter vector, eBDFPA and ePFPA are strategically equivalent.*

**Theorem D.7.** *In the symmetric case, when all buyers' identical bidding qf is inverse Lipschitz continuous, under the symmetric budget-extracting parameter vector, eBDFPA and eBDSPA are strategically equivalent.*

We prove the above two theorems in sequence.

PROOF OF THEOREM D.6. We let the common bidding qf in eBDFPA be $\widetilde{v}_0^{(2)}$, and the counterpart of PFPA be $\widetilde{v}_0^{(3)}$. Meanwhile, we let some budget-extracting tuple of BDFPA and PFPA be respectively $\boldsymbol{\alpha}^{\text{e}} = (\alpha_0)^n$ and $\boldsymbol{\beta}^{\text{e}} = (\beta_0)^n$. Further, let the common value qf be $v_0$. Similar to the proof of Theorem 4.2, we give the expected utility of any buyer under both auction mechanisms in the following:

$$u^{\text{eBDF}}((\alpha_0)^n, \widetilde{v}_0^{(2)}, v_0) = \int_0^1 \left( v_0(q_i) - \widetilde{v}_0^{(2)}(q_i) \right) \cdot \left( \int_{\boldsymbol{q}_{-i}} \Phi_i((\alpha_0)^n, \widetilde{v}_0^{(2)}, \boldsymbol{q}) \, \mathrm{d}\boldsymbol{q}_{-i} \right) \mathrm{d}q_i. \tag{18}$$

$$u^{\text{ePF}}((\beta_0)^n, \widetilde{v}_0^{(3)}, v_0) = \int_0^1 \left( v_0(q_i) - \beta_0 \widetilde{v}_0^{(3)}(q_i) \right) \cdot \left( \int_{\boldsymbol{q}_{-i}} \Phi_i((\beta_0)^n, \widetilde{v}_0^{(3)}, \boldsymbol{q}) \, \mathrm{d}\boldsymbol{q}_{-i} \right) \mathrm{d}q_i. \tag{19}$$

Further, the seller's expected revenue in these two mechanisms are correspondingly the following:

$$w^{\text{eBDF}}((\alpha_0)^n, \widetilde{\boldsymbol{v}}_0^{(2)}) = \sum_{i=1}^{n} \left( \int_0^1 \left( \widetilde{v}_0^{(2)}(q_i) - \lambda \right) \cdot \left( \int_{\boldsymbol{q}_{-i}} \Phi_i((\alpha_0)^n, \widetilde{v}_0^{(2)}, \boldsymbol{q}) \, \mathrm{d}\boldsymbol{q}_{-i} \right) \mathrm{d}q_i \right). \tag{20}$$

$$w^{\text{ePF}}((\beta_0)^n, \widetilde{\boldsymbol{v}}_0^{(3)}) = \sum_{i=1}^{n} \left( \int_0^1 \left( \beta_0 \widetilde{v}_0^{(3)}(q_i) - \lambda \right) \cdot \left( \int_{\boldsymbol{q}_{-i}} \Phi_i((\beta_0)^n, \widetilde{v}_0^{(3)}, \boldsymbol{q}) \, \mathrm{d}\boldsymbol{q}_{-i} \right) \mathrm{d}q_i \right). \tag{21}$$

*From eBDFPA to ePFPA.* For this part, if $\alpha_0 = 1$, then by the budget constraints of BDFPA and PFPA, $\boldsymbol{\beta} = 1^n$ is also feasible for PFPA with $\widetilde{v}_0^{(3)} = \widetilde{v}_0^{(2)}$. As a result, under the identical mapping from $\widetilde{v}_0^{(2)}$ to $\widetilde{v}_0^{(3)}$, we have $\beta_0 = 1$, and the two mechanisms are essentially the same. Buyers' expected utility and the seller's expected revenue face no change naturally under the mapping.

We now consider the more general case with $\alpha_0 < 1$. Our intuition is to "raise" the right part of $\widetilde{v}_0^{(2)}$ to construct $\widetilde{v}_0^{(3)}$. However, we have to maintain that $\widetilde{v}_0^{(3)}(1) \leq 1$, therefore, it could be impossible to keep $\beta_0 = \alpha_0$, but rather have $\beta_0$ close to 1.

By definition of budget-extracting, each buyer exhausts her budget under $(\alpha_0)^n$. Therefore, we can define $q_0 \in [0, 1]$ as $q_0 := \inf\{x \in [0, 1] \mid \widetilde{v}_0^{(2)}(x) \geq \lambda/\alpha_0\}$.

Thus, the expected payment of any buyer $i$ satisfies that

$$p^{\text{eBDF}}((\alpha_0)^n, \widetilde{v}_0^{(2)}) = \int_0^1 \widetilde{v}_0^{(2)}(q_i) \cdot \left( \int_{\boldsymbol{q}_{-i}} \Phi_i((\alpha_0)^n, \widetilde{v}_0^{(2)}, \boldsymbol{q}) \, \mathrm{d}\boldsymbol{q}_{-i} \right) \mathrm{d}q_i$$

$$= \int_{q_0}^1 \widetilde{v}_0^{(2)}(x) \cdot x^{n-1} \, \mathrm{d}x = \rho_0. \tag{22}$$

Since $\lambda < \lambda/\alpha_0 \leq \widetilde{v}_0(x) \leq 1$ on $[q_0, 1]$, we directly derive by strict monotonicity and inverse Lipschitz continuity that

$$\lambda \cdot \frac{1 - q_0^n}{n} < \rho_0 < \frac{1 - q_0^n}{n}.$$

Now, we take $\lambda < \beta_0 < 1$ such that $\rho_0/\beta_0 < (1 - q_0^n)/n$, and we use different ways to construct the function basing on the value of $\rho_0$. Concretely, we let

$$t_0(\beta_0, q_0) := \beta_0 \cdot \int_{q_0}^1 \left( \frac{1 - \lambda/\beta_0}{1 - q_0} \cdot x + \frac{\lambda/\beta_0 - q_0}{1 - q_0} \right) x^{n-1} \, \mathrm{d}x.$$

*Case 1:* $t_0(\beta_0, q_0) < \rho_0 < \beta_0(1 - q_0^n)/n$. For this case, for $a \in [0, 1]$, we let

$$y_0(a) := \beta_0 \cdot \int_{q_0}^1 \left( \left(1 - \frac{\lambda}{\beta_0}\right) \left(\frac{x - q_0}{1 - q_0}\right)^a + \frac{\lambda}{\beta_0} \right) x^{n-1} \, \mathrm{d}x,$$

which is a continuous and strictly decreasing function on $[0, 1]$. Notice that $y_0(0) = \beta_0 \cdot (1 - q_0^n)/n$ and $y_0(1) = t_0(\beta_0, q_0)$, then by intermediate value theorem, there exists $a^* \in (0, 1)$ such that $y_0(a^*) = \rho_0$. And we define

$$\widetilde{v}_0^{(3)}(x) = \begin{cases} a_1 \cdot \exp\{a_2 x\} & 0 \le x < q_0, \\ (1 - \lambda/\beta_0) \cdot ((x - q_0)/(1 - q_0))^{a^*} + \lambda/\beta_0 & q_0 \le x \le 1, \end{cases}$$

where $a_1 = \lambda/\beta_0 \cdot \exp\{-k^* \beta_0 q_0/\lambda\}$, $a_2 = k^* \beta_0/\lambda$, and $k^* > 0$ is the right derivative of $(1 - \lambda/\beta_0) \cdot ((x - q_0)/(1 - q_0))^{a^*} + \lambda/\beta_0$ on $x = q_0$. Feasibility, strict monotonicity, differentiability, and inverse Lipschitz continuity naturally follow. Further, we note that $\widetilde{v}_0^{(3)}(q_0) = \lambda/\beta_0$, and

$$\int_{q_0}^1 \beta_0 \widetilde{v}_0^{(3)}(x) x^{n-1} \, \mathrm{d}x = y_0(a^*) = \rho_0.$$

*Case 2:* $\lambda(1 - q_0^n)/n < \rho_0 \le t_0(\beta_0, q_0)$. For this case, for $k \in [0, (1 - \lambda/\beta_0)/(1 - q_0)]$, we let

$$z_0(k) := \beta_0 \int_{q_0}^1 \left( k(x - q_0) + \frac{\lambda}{\beta_0} \right) x^{n-1} \, \mathrm{d}x,$$

which is continuous and strictly increasing on $[0, (1 - \lambda/\beta_0)/(1 - q_0)]$. Since $z_0(0) = \lambda(1 - q_0^n)$ and $z_0(1 - \lambda/\beta_0)/(1 - q_0) = t_0(\beta_0, q_0)$, there exists $k^* \in (0, (1 - \lambda/\beta_0)/(1 - q_0)]$ such that $z_0(k^*) = \rho_0$. We therefore let

$$\widetilde{v}_0^{(3)}(x) = \begin{cases} a_1 \cdot \exp\{a_2 x\} & 0 \le x < q_0, \\ k^*(x - q_0) + \lambda/\beta_0 & q_0 \le x \le 1, \end{cases}$$

where $a_1 = \lambda/\beta_0 \cdot \exp\{-k^* \beta_0 q_0/\lambda\}$, $a_2 = k^* \beta_0/\lambda$. Similarly, feasibility, strict monotonicity, differentiability, and inverse Lipschitz continuity hold. Further, we still have $\widetilde{v}_0^{(3)}(q_0) = \lambda/\beta_0$, and

$$\int_{q_0}^1 \beta_0 \widetilde{v}_0^{(3)}(x) x^{n-1} \, \mathrm{d}x = y_0(a^*) = \rho_0.$$

For both two cases, we derive that $\beta_0$ is the budget-extracting multiplier for PFPA under $\widetilde{v}_0^{(3)}$, and each buyer exhausts her budget. Further, notice that for each $1 \le i \le n$,

$$\Phi_i((\alpha_0)^n, \widetilde{v}_0^{(2)}, \mathbf{q}) = \Phi_i((\beta_0)^n, \widetilde{v}_0^{(3)}, \mathbf{q}),$$

since either of them equals 1 if and only if $q_i = \max \mathbf{q}$ and $q_i \ge q_0$. Combining with (18), (19), (20), and (21), the proof of this side is finished.

*From ePFPA to eBDFPA.* The proof of this side is similar. To start with, the case of $\beta_0 = 1$ is almost the same to the other side we have already discussed, and we now suppose $\beta_0 < 1$. Let $q_0 := \inf\{x \in [0, 1] \mid \widetilde{v}_0^{(3)}(x) \ge \lambda/\beta_0\} \in [0, 1]$, which should not be confused with the $q_0$ defined in the previous part. Since $\beta_0 < 1$ and every buyer's budget is binding, the expected payment of each buyer satisfies

$$p^{\mathrm{ePF}}((\beta_0)^n, \widetilde{v}_0^{(3)}) = \int_0^1 \beta_0 \widetilde{v}_0^{(3)}(q_i) \cdot \left( \int_{\mathbf{q}_{-i}} \Phi_i(\beta^{\mathrm{e}}, \widetilde{v}_0^{(3)}, q) \, \mathrm{d}\mathbf{q}_{-i} \right) \mathrm{d}q_i$$

$$= \int_{q_0}^1 \beta_0 \widetilde{v}_0^{(3)}(x) \cdot x^{n-1} \, \mathrm{d}x = \rho_0. \tag{23}$$

By strict monotonicity and inverse Lipschitz continuity, we have

$$\lambda \cdot \frac{1 - q_0^n}{n} = \lambda \int_{q_0}^1 x^{n-1} \, \mathrm{d}x < \rho_0 < \int_{q_0}^1 x^{n-1} \, \mathrm{d}x = \frac{1 - q_0^n}{n}.$$

We now let

$$t_1(\beta_0, q_0) := \int_{q_0}^1 \left( \frac{1 - \lambda/\beta_0}{1 - q_0} \cdot x + \frac{\lambda/\beta_0 - q_0}{1 - q_0} \right) x^{n-1} \, \mathrm{d}x,$$

which sets the threshold for $\rho_0$, and we similarly construct function $\widetilde{v}_0^{(2)}$ for two different cases like the previous part, with

$$y_1(a) := \int_{q_0}^1 \left( \left(1 - \frac{\lambda}{\beta_0}\right) \left(\frac{x - q_0}{1 - q_0}\right)^a + \frac{\lambda}{\beta_0}\right) x^{n-1} \, \mathrm{d}x,$$

and

$$z_1(k) := \int_{q_0}^1 \left(k(x - q_0) + \frac{\lambda}{\beta_0}\right) x^{n-1} \, \mathrm{d}x.$$

Under a similar reasoning, we can derive that $\beta_0$ makes an eBDFPA, i.e., $\alpha_0 = \beta_0$, and the interim allocation function is the same for these two auctions. Again by (18), (19), (20), and (21), the proof of this part is also done.

By combining the two directions, we finish the proof of the theorem. □

PROOF OF THEOREM D.7. Following previous notations, we let the identical bidding qf of all buyers in BDFPA be $\widetilde{v}_0^{(2)}$, and the counterpart for BDSPA be $\widetilde{v}_0^{(4)}$. Further, for BDFPA, let the maximum symmetric budget-extracting parameter tuple be $\boldsymbol{\alpha}^{\mathrm{e}} = (\alpha_0)^n$; while for BDSPA, let the symmetric budget-extracting multiplier vector be $\boldsymbol{\mu}^{\mathrm{e}} = (\mu_0)^n$. Further, let the common value qf be $v_0$. We now present any buyer's expected utility and the seller's expected revenue under eBDSPA in the following. Note that (18) and (20) already gave these two values for eBDFPA.

$$u^{\mathrm{eBDS}}((\mu_0)^n, \widetilde{v}_0^{(4)}, v_0) = \int_0^1 \left( v_0(q_i) - \max_{i' \neq i} \left\{ \widetilde{v}_0^{(4)}(q_{i'}), \frac{\lambda}{\mu_0} \right\} \right) \cdot \left( \int_{\boldsymbol{q}_{-i}} \Phi_i((\mu_0)^n, \widetilde{v}_0^{(4)}, \boldsymbol{q}) \, \mathrm{d}\boldsymbol{q}_{-i} \right) \mathrm{d}q_i. \tag{24}$$

$$w^{\mathrm{eBDS}}((\mu_0)^n, \widetilde{v}_0^{(4)}) = \sum_{i=1}^n \left( \int_0^1 \left( \max_{i' \neq i} \left\{ \widetilde{v}_0^{(4)}(q_{i'}), \frac{\lambda}{\mu_0} \right\} - \lambda \right) \cdot \left( \int_{\boldsymbol{q}_{-i}} \Phi_i((\mu_0)^n, \widetilde{v}_0^{(4)}, \boldsymbol{q}) \, \mathrm{d}\boldsymbol{q}_{-i} \right) \mathrm{d}q_i \right). \tag{25}$$

The rest of the proof, i.e., the construction part, largely simulates the proof of Theorem D.6.

*From eBDSPA to eBDFPA.* For this part, we will give a mapping from $\widetilde{v}_0^{(4)}$ to $\widetilde{v}_0^{(2)}$ meanwhile guaranteeing that $\alpha_0 = \mu_0$. Nevertheless, we first deal with extreme cases when $\mu_0 = 1$ and $\widetilde{v}_0^{(4)}(1) \leq \lambda$, which means that the item is never allocated. In this scenario, let $\widetilde{v}_0^{(2)} = \widetilde{v}_0^{(4)}$ suffices, as any buyer's utility and the seller's revenue in both auctions are always zero.

Now we let $q_0 := \inf\{x \in [0, 1] \mid \widetilde{v}_0^{(4)}(x) \geq \lambda/\mu_0\} \in [0, 1]$. Such $q_0$ exists since the payment of each buyer is non-zero by the definition of budget-extracting. We first write the payment of buyers in BDSPA, which is

$$\begin{aligned}
p^{\mathrm{eBDS}}((\mu_0)^n, \widetilde{v}_0^{(4)}) &= \int_{[0,1]^n} \max_{i' \neq i} \left\{ \widetilde{v}_0^{(4)}(q_{i'}), \frac{\lambda}{\mu_0} \right\} \cdot \Phi_i((\mu_0)^n, \widetilde{v}^{(4)}, \boldsymbol{q}) \, \mathrm{d}\boldsymbol{q} \\
&= \int_{q_0}^1 \left( \frac{\lambda}{\mu_0} q_0^{n-1} + \int_{q_0}^x \widetilde{v}_0^{(4)}(z)(n-1)z^{n-2} \, \mathrm{d}z \right) \mathrm{d}x \\
&> \int_{q_0}^1 \left( \frac{\lambda}{\mu_0} q_0^{n-1} + \int_{q_0}^x \frac{\lambda}{\mu_0}(n-1)z^{n-2} \, \mathrm{d}z \right) \mathrm{d}x \\
&= \int_{q_0}^1 \frac{\lambda}{\mu_0} x^{n-1} \, \mathrm{d}x = \frac{\lambda}{\mu_0} \cdot \frac{1 - q_0^n}{n} > \lambda \cdot \frac{1 - q_0^n}{n}.
\end{aligned} \tag{26}$$

Here, the second equality follows by considering the second-max discounted value when the max value is fixed. The inequality holds since $\widetilde{v}_0^{(4)}$ is strictly increasing and inverse Lipschitz continuous. Further, we have

$$\begin{aligned}
p^{\mathrm{eBDS}}((\mu_0)^n, \widetilde{v}_0^{(4)}) &= \int_{q_0}^1 \left( \frac{\lambda}{\mu_0} q_0^{n-1} + \int_{q_0}^x \widetilde{v}_0^{(4)}(z)(n-1)z^{n-2} \, \mathrm{d}z \right) \mathrm{d}x \\
&< \int_{q_0}^1 \left( q_0^{n-1} + \int_{q_0}^x (n-1)z^{n-2} \, \mathrm{d}z \right) \mathrm{d}x \\
&= \frac{1 - q_0^n}{n}.
\end{aligned}$$

The inequality is due to $\mu_0 \geq \lambda$. Therefore,

$$\lambda \cdot \frac{1 - q_0^n}{n} < p^{\mathrm{eBDS}}((\mu_0)^n, \widetilde{v}_0^{(4)}) < \frac{1 - q_0^n}{n}.$$

With the above inequality, we can now construct $\widetilde{v}_0^{(2)}$ as the construction from ePFPA to eBDFPA in the proof of Theorem D.6 by replacing $\beta_0$ there with $\mu_0$ and $\rho_0$ with $p^{\mathrm{eBDS}}((\mu_0)^n, \widetilde{v}_0^{(4)})$. We should notice here that $p^{\mathrm{eBDS}}((\mu_0)^n, \widetilde{v}_0^{(4)}) = \rho_0$ may not establish as it is possible that each buyer does not exhaust her budget even with $\mu_0 = 1$. The reasoning part also inherits from the previous proof by showing $\alpha_0 = \mu_0$ and considering (18), (24), (20), and (25).

*From eBDFPA to eBDSPA.* For this part, we also first deal with the special case that $\alpha_0 = 1$ and $\widetilde{v}_0^{(2)}(1) \leq \lambda$. Under this case, we take $\widetilde{v}_0^{(4)} = \widetilde{v}_0^{(2)}$, therefore, the expected payment of any buyer in either BDSPA or BDFPA is zero. As a result, the revenue of the seller stays at zero as well.

In the more general case, we let $q_0 := \inf\{x \in [0,1] \mid \widetilde{v}_0^{(2)}(x) \geq \lambda/\alpha_0\} \in [0,1]$, which exists by the definition of budget-extracting. Therefore, we have

$$p^{\text{eBDF}}((\alpha_0)^n, \widetilde{v}_0^{(2)}) = \int_{q_0}^1 \widetilde{v}_0^{(2)}(x) \cdot x^{n-1} \, dx,$$

and

$$\lambda \cdot \frac{1 - q_0^n}{n} < p^{\text{eBDF}}((\alpha_0)^n, \widetilde{v}_0^{(2)}) < \frac{1 - q_0^n}{n}.$$

Thus, we let the threshold be

$$t_2(\alpha_0, q_0) := \int_{q_0}^1 \left( \frac{\lambda}{\alpha_0} q_0^{n-1} + \int_{q_0}^x \left( \frac{1 - \lambda/\alpha_0}{1 - q_0} \cdot z + \frac{\lambda/\alpha_0 - q_0}{1 - q_0} \right)(n-1)z^{n-1} \, dz \right) dx,$$

and for two cases, the functions used for construction become

$$y_2(a) := \int_{q_0}^1 \left( \frac{\lambda}{\alpha_0} q_0^{n-1} + \int_{q_0}^x \left( \left( 1 - \frac{\lambda}{\alpha_0} \right) \left( \frac{z - q_0}{1 - q_0} \right)^a + \frac{\lambda}{\alpha_0} \right)(n-1)z^{n-1} \, dz \right) dx,$$

and

$$z_2(k) := \int_{q_0}^1 \left( \frac{\lambda}{\alpha_0} q_0^{n-1} + \int_{q_0}^x \left( k(x - q_0) + \frac{\lambda}{\alpha_0} \right)(n-1)z^{n-1} \, dz \right) dx.$$

Under similar constructions, we see that $(\alpha_0)^n$ is a budget-extracting multiplier for BDSPA with $\widetilde{v}_0^{(4)}$. In fact, when $\alpha_0 = 1$, the budget-extracting condition naturally holds. When $\alpha_0 < 1$, $(\alpha_0)^n$ exhausts each buyer's budget. Therefore, the proof for this side is done, and we finish the proof of the theorem. $\qquad\square$

*Remark* D.1. By comparing the expected payment of a buyer in BDFPA and BDSPA, an appealing approach to prove the theorem is to take $\alpha_0 = \mu_0$, the effective quantile of both auctions start at an identical $q_0$, and to have when $x \geq q_0$,

$$\widetilde{v}_0^{(2)}(x) \cdot x^{n-1} = \frac{\lambda}{\mu_0} q_0^{n-1} + \int_{q_0}^x \widetilde{v}_0^{(4)}(z)(n-1)z^{n-2} \, dz.$$

This seems to be an elegant solution, with $\widetilde{v}_0^{(2)}$ being a continuous weighted average of $\widetilde{v}_0^{(4)}$. However, this idea does not work. The reason is that the above mapping from $\widetilde{v}_0^{(4)}$ to $\widetilde{v}_0^{(2)}$ would lose the inverse Lipschitz continuity, as the derivative of $\widetilde{v}_0^{(2)}$ at $q_0$ would be zero. On the other side, the mapping from $\widetilde{v}_0^{(2)}$ to $\widetilde{v}_0^{(4)}$ would even lose the strict monotonicity. As a result, we have to adopt the methodology we use in the proof of Theorem D.7.

# E PROOFS IN SECTION A.1

## E.1 Proof of Theorem A.1

The theorem is proved in steps. First, we characterize some essential properties of bid-discount in the first-price auction. Then we prove the five statements in the theorem in order.

We come to some basic features of the bid-discount method in first-price auctions. To start with, obviously, given buyers' bidding qf profile $\widetilde{v} = (\widetilde{v}_i)_{1 \leq i \leq n}$ and budget profile $(\rho_i)_{1 \leq i \leq n}$, notice that $\mathcal{M}^{\text{BDF}}(0)$ must be a feasible bid-discount mechanism, in which the item is never assigned and each buyer's payment is zero. Thus there exists a tuple of bid-discount multipliers $\boldsymbol{\alpha}$ such that $\mathcal{M}^{\text{BDF}}(\boldsymbol{\alpha})$ is a feasible bid-discount mechanism.

We now show that when a buyer's bid-discount multiplier slightly increases, her expected payment does not increase too much.

**Lemma E.1.** *There exists a constant $C$ which satisfies the following: For any $\boldsymbol{\alpha} = (\alpha_1, \ldots, \alpha_n)$ and $1 \leq i \leq n$ such that $\alpha_i < 1$, let $\boldsymbol{\alpha}' = \boldsymbol{\alpha} + \delta \boldsymbol{e}_i$ where $0 < \delta \leq 1 - \alpha_i$ and $\boldsymbol{e}_i$ is the vector with the $i$-th entry one and all other entries zero. Then the expected payment of buyer $i$ in $\mathcal{M}^{\text{BDF}}(\boldsymbol{\alpha}')$ is at most the expected payment of buyer $i$ in $\mathcal{M}^{\text{BDF}}(\boldsymbol{\alpha})$ plus $C\delta$.*

Before we prove the lemma, some preparations are required. We define $G_{\boldsymbol{\alpha}, i}$ be the cumulative distribution function of $\max_{i' \neq i}\{\alpha_{i'}\widetilde{v}_{i'}(q_{i'}), \lambda\}$ when $\boldsymbol{q}_{-i}$ is chosen uniformly in $[0,1]^{n-1}$. Then,

**Lemma E.2.** *For any $\boldsymbol{\alpha} \in [0,1]^n$, we have:*

- $G_{\boldsymbol{\alpha}, i}(\cdot)$ *is zero on $[0, \lambda)$.*
- $\lambda$ *is the only possible discontinuous point of $G_{\boldsymbol{\alpha}, i}(\cdot)$.*
- *If $G_{\boldsymbol{\alpha}, i}(\lambda) < 1$, then $G_{\boldsymbol{\alpha}, i}(\cdot)$ is Lipschitz continuous on $[\lambda, +\infty)$.*

PROOF OF LEMMA E.2. Let $\widehat{v} := \max_{i' \neq i} \{\alpha_{i'}\widetilde{v}_{i'}(q_{i'}), \lambda\}$ be a random variable when $\boldsymbol{q}_{-i}$ is uniformly drawn from $[0, 1]^{n-1}$. The only non-trivial part is the third part, which is to show the Lipschitz continuity when $\widehat{v} \geq \lambda$. For any buyer $i' \neq i$, since her bidding cdf $\widetilde{F}_{i'}(\cdot)$ is Lipschitz continuous, there exists a constant $C_{i'}$ such that for any $\widetilde{v}_{i'}^{(1)}$ and $\widetilde{v}_{i'}^{(2)}$, we have

$$\left|\widetilde{F}_{i'}(\widetilde{v}_{i'}^{(1)}) - \widetilde{F}_{i'}(\widetilde{v}_{i'}^{(2)})\right| \leq C_{i'}\left|\widetilde{v}_{i'}^{(1)} - \widetilde{v}_{i'}^{(2)}\right|.$$

Since $\widetilde{v}_{i'}(q_{i'})$ is upper bounded (say, by $\bar{v}_{i'}$) for any $1 \leq i' \leq n$, there exists a constant $\delta_{i'} > 0$ for each $i'$ such that $\delta_{i'} \cdot \bar{v}_{i'} < \lambda$. Now let $\lambda \leq \widehat{v}^{(1)} < \widehat{v}^{(2)}$. Clearly, since $G_{\boldsymbol{\alpha},i}(\lambda) < 1$, there exists at least one $i' \neq i$ such that $\alpha_{i'} > \delta_{i'}$. Then we have

$$\left|G_{\boldsymbol{\alpha},i}(\widehat{v}^{(1)}) - G_{\boldsymbol{\alpha},i}(\widehat{v}^{(2)})\right| = \left|\prod_{i' \neq i, \alpha_{i'} > \delta_{i'}} \widetilde{F}_{i'}(\widehat{v}^{(1)}/\alpha_{i'}) - \prod_{i' \neq i, \alpha_{i'} > \delta_{i'}} \widetilde{F}_{i'}(\widehat{v}^{(2)}/\alpha_{i'})\right|$$

$$\leq \sum_{i' \neq i, \alpha_{i'} > \delta_{i'}} \left|\widetilde{F}_{i'}(\widehat{v}^{(1)}/\alpha_{i'}) - \widetilde{F}_{i'}(\widehat{v}^{(2)}/\alpha_{i'})\right|$$

$$\leq \left(\sum_{i' \neq i, \alpha_{i'} > \delta_{i'}} C_{i'}/\alpha_{i'}\right) \cdot \left|\widehat{v}^{(1)} - \widehat{v}^{(2)}\right|$$

$$\leq \left(\sum_{i' \neq i} C_{i'}/\delta_{i'}\right) \cdot \left|\widehat{v}^{(1)} - \widehat{v}^{(2)}\right|$$

Here, the first inequality is because of $\widetilde{F}_{i'}$ is no greater than 1 for any $i' \neq i$. This shows that $G_{\boldsymbol{\alpha},i}$ is continuous with Lipschitz constant $\widehat{C}_i := \sum_{i' \neq i} C_{i'}/\delta_{i'}$ on the right side of $\lambda$, and the proof is finished. $\square$

Now, we come back to prove Lemma E.1.

PROOF OF LEMMA E.1. Recall that by definition, the expected payment of buyer $i$ in $\mathcal{M}^{\mathrm{BDF}}(\boldsymbol{\alpha})$ is

$$\int_0^1 \widetilde{v}_i(q_i) \cdot \left(\int_{\boldsymbol{q}_{-i}} I\left[\alpha_i\widetilde{v}_i(q_i) \geq \max_{i' \neq i}\{\alpha_{i'}\widetilde{v}_{i'}(q_{i'}), \lambda\}\right] d\boldsymbol{q}_{-i}\right) dq_i.$$

Similarly, the expected payment of buyer $i$ in $\mathcal{M}^{\mathrm{BDF}}(\boldsymbol{\alpha}')$ is

$$\int_0^1 \widetilde{v}_i(q_i) \cdot \left(\int_{\boldsymbol{q}_{-i}} I\left[(\alpha_i + \delta)\widetilde{v}_i(q_i) \geq \max_{i' \neq i}\{\alpha_{i'}\widetilde{v}_{i'}(q_{i'}), \lambda\}\right] d\boldsymbol{q}_{-i}\right) dq_i.$$

Note that

$$I\left[(\alpha_i + \delta)\widetilde{v}_i(q_i) \geq \max_{i' \neq i}\{\alpha_{i'}\widetilde{v}_{i'}(q_{i'}), \lambda\}\right] = I\left[\alpha_i\widetilde{v}_i(q_i) \geq \max_{i' \neq i}\{\alpha_{i'}\widetilde{v}_{i'}(q_{i'}), \lambda\}\right]$$

$$+ I\left[\alpha_i\widetilde{v}_i(q_i) < \max_{i' \neq i}\{\alpha_{i'}\widetilde{v}_{i'}(q_{i'}), \lambda\} \leq (\alpha_i + \delta)\widetilde{v}_i(q_i)\right]. \tag{27}$$

Thus the increment of buyer $i$'s expected payment after replacing $\boldsymbol{\alpha}$ with $\boldsymbol{\alpha}'$ is

$$\int_0^1 \widetilde{v}_i(q_i) \cdot \left(\int_{\boldsymbol{q}_{-i}} I\left[\alpha_i\widetilde{v}_i(q_i) < \max_{i' \neq i}\{\alpha_{i'}\widetilde{v}_{i'}(q_{i'}), \lambda\} \leq (\alpha_i + \delta)\widetilde{v}_i(q_i)\right] d\boldsymbol{q}_{-i}\right) dq_i.$$

$$= \int_0^1 \widetilde{v}_i(q_i) \cdot \left(G_{\boldsymbol{\alpha},i}\left((\alpha_i + \delta)\widetilde{v}_i(q_i)\right) - G_{\boldsymbol{\alpha},i}\left(\alpha_i\widetilde{v}_i(q_i)\right)\right) dq_i. \tag{28}$$

By Lemma E.2, $G_{\boldsymbol{\alpha},i}(\delta_i\widetilde{v}_i(q_i)) = 0$ always holds for any $q_i \in [0, 1]$, where we recall that $\delta_i$ is defined as a constant such that $\delta_i \cdot \bar{v}_i < \lambda$. We use $\delta_i$ as a threshold to analyze the formula (28).

When $\alpha_i \geq \delta_i$, there are three parts which we analyze correspondingly, depending on whether $\lambda$ lies in $(\alpha_i\widetilde{v}_i(q_i), (\alpha_i + \delta)\widetilde{v}_i(q_i))$.

- $(\alpha_i + \delta)\widetilde{v}_i(q_i) \leq \lambda$. Let $\bar{q}$ be the minimum $q_i \leq 1$ such that $(\alpha_i + \delta)\widetilde{v}_i(q_i) \leq \lambda$, if there exists, and $\bar{q} := 1$ otherwise. By monotonicity, $(\alpha_i + \delta)\widetilde{v}_i(q_i) \leq \lambda$ holds for any $\bar{q} < q_i \leq 1$. By Lemma E.2, we have $\int_{\bar{q}}^1 \widetilde{v}_i(q_i) \cdot \left(G_{\boldsymbol{\alpha},i}\left((\alpha_i + \delta)\widetilde{v}_i(q_i)\right) - G_{\boldsymbol{\alpha},i}\left(\alpha_i\widetilde{v}_i(q_i)\right)\right) dq_i = 0$.
- $\alpha_i\widetilde{v}_i(q_i) < \lambda < (\alpha_i + \delta)\widetilde{v}_i(q_i)$. Let $\underline{q}$ be the maximum $q_i \geq 0$ such that $\alpha_i\widetilde{v}_i(q_i) \geq \lambda$, if there exists, and $\underline{q} := 0$ otherwise. By monotonicity, $\alpha_i\widetilde{v}_i(q_i) \leq \lambda$ holds for any $\underline{q} \leq q_i < 1$. Now, since $\widetilde{v}_i$ is upper bounded by $\bar{v}_i$ and that $\widetilde{F}_i(\cdot)$ is continuous with

Lipschitz constant $C_i$, we derive that

$$\int_{\underline{q}}^{\bar{q}} \widetilde{v}_i(q_i) \cdot \left( G_{\boldsymbol{\alpha},i} \left( (\alpha_i + \delta)\widetilde{v}_i(q_i) \right) - G_{\boldsymbol{\alpha},i} \left( \alpha_i \widetilde{v}_i(q_i) \right) \right) \mathrm{d}q_i \leq \bar{v}_i \cdot \left( \bar{q} - \underline{q} \right)$$

$$\leq \bar{v}_i \cdot \left( \widetilde{F}_i(\lambda/\alpha_i) - \widetilde{F}_i(\lambda/(\alpha_i + \delta)) \right) \leq \bar{v}_i \cdot C_i \cdot (\lambda/\alpha_i - \lambda/(\alpha_i + \delta))$$

$$\leq \bar{v}_i \cdot C_i \cdot \frac{\lambda}{\alpha_i^2} \cdot \delta.$$

Here, the first inequality holds since $G_{\boldsymbol{\alpha},i}$ is bounded by 1.

- $\alpha_i \widetilde{v}_i(q_i) \geq \lambda$. By monotonicity, $\alpha_i \widetilde{v}_i(q_i) \geq \lambda$ holds for any $0 \leq q_i < \underline{q}$. By Lemma E.2, we have

$$\int_0^{\underline{q}} \widetilde{v}_i(q_i) \cdot \left( G_{\boldsymbol{\alpha},i} \left( (\alpha_i + \delta)\widetilde{v}_i(q_i) \right) - G_{\boldsymbol{\alpha},i} \left( \alpha_i \widetilde{v}_i(q_i) \right) \right) \mathrm{d}q_i$$

$$\leq \int_0^{\underline{q}} \widetilde{v}_i^2(q_i) \cdot \widehat{C}_i \cdot \delta \, \mathrm{d}q_i \leq \bar{v}_i^2 \cdot \widehat{C}_i \cdot \delta.$$

As a result, in this scenario, we have

$$\int_0^1 \widetilde{v}_i(q_i) \cdot \left( G_{\boldsymbol{\alpha},i} \left( (\alpha_i + \delta)\widetilde{v}_i(q_i) \right) - G_{\boldsymbol{\alpha},i} \left( \alpha_i \widetilde{v}_i(q_i) \right) \right) \mathrm{d}q_i$$

$$= \left( \int_0^{\underline{q}} + \int_{\underline{q}}^{\bar{q}} + \int_{\bar{q}}^1 \right) \left( \widetilde{v}_i(q_i) \cdot \left( G_{\boldsymbol{\alpha},i} \left( (\alpha_i + \delta)\widetilde{v}_i(q_i) \right) - G_{\boldsymbol{\alpha},i} \left( \alpha_i \widetilde{v}_i(q_i) \right) \right) \right) \mathrm{d}q_i$$

$$\leq \max \left\{ \bar{v}_i \cdot C_i \cdot \frac{\lambda}{\alpha_i^2}, \bar{v}_i^2 \cdot \widehat{C}_i \right\} \cdot \delta \leq \max \left\{ \bar{v}_i \cdot C_i \cdot \frac{\lambda}{\delta_i^2}, \bar{v}_i^2 \cdot \widehat{C}_i \right\} \cdot \delta.$$

In the case that $\alpha_i < \delta_i$, since $\delta_i \cdot \bar{v}_i < \lambda$, by Lemma E.2, $G_{\boldsymbol{\alpha},i}(\delta_i \widetilde{v}_i(q_i)) = G_{\boldsymbol{\alpha},i}(\alpha_i \widetilde{v}_i(q_i))$ always holds for any $q_i \in [0,1]$. Therefore, when $\delta \leq \delta_i$, then the proof is finished, otherwise, notice that

$$\int_0^1 \widetilde{v}_i(q_i) \cdot \left( G_{\boldsymbol{\alpha},i} \left( (\alpha_i + \delta)\widetilde{v}_i(q_i) \right) - G_{\boldsymbol{\alpha},i} \left( \alpha_i \widetilde{v}_i(q_i) \right) \right) \mathrm{d}q_i$$

$$= \int_0^1 \widetilde{v}_i(q_i) \cdot \left( G_{\boldsymbol{\alpha},i} \left( (\alpha_i + \delta)\widetilde{v}_i(q_i) \right) - G_{\boldsymbol{\alpha},i} \left( \delta_i \widetilde{v}_i(q_i) \right) \right) \mathrm{d}q_i$$

$$\leq \max \left\{ \bar{v}_i \cdot C_i \cdot \frac{\lambda}{\delta_i^2}, \bar{v}_i^2 \cdot \widehat{C}_i \right\} \cdot (\alpha_i + \delta - \delta_i) \leq \max \left\{ \bar{v}_i \cdot C_i \cdot \frac{\lambda}{\delta_i^2}, \bar{v}_i^2 \cdot \widehat{C}_i \right\} \cdot \delta.$$

Now, we conclude the proof of Lemma E.1 by having $C := \max_{1 \leq i \leq n} \{ \bar{v}_i \cdot C_i \cdot \lambda/\delta_i^2, \bar{v}_i^2 \cdot \widehat{C}_i \}$. □

From Lemma E.1, we derive an essential property that the set of all feasible tuples of bid-discount multipliers is compact, which is given in the following lemma.

**Lemma E.3.** Let $\mathcal{A}$ be the set of all $\boldsymbol{\alpha}$ such that $\mathcal{M}^{\mathrm{BDF}}(\boldsymbol{\alpha})$ is a feasible bid-discount mechanism. Then $\mathcal{A}$ is compact.

PROOF OF LEMMA E.3. It suffices to show that all $\boldsymbol{\alpha}$s satisfying the budget-feasible constraints form a closed set.

Define $\varphi : [0,1]^n \to \mathbb{R}^n$ to be the mapping from the tuple of multipliers $\boldsymbol{\alpha}$ to the expected payment vector of all buyers when the quantile profile is uniformly distributed in $[0,1]^n$. By Lemma E.1 we know that $\varphi$ is Lipschitz continuous, which implies that the pre-image of every closed set under $\varphi$ is also closed. Since $\mathcal{A} = \{ \boldsymbol{\alpha} : \varphi(\boldsymbol{\alpha}) \in \prod_i [0, \rho_i] \}$ is the pre-image of $\prod_i [0, \rho_i]$, which is apparently a closed set, $\mathcal{A}$ is closed as well. Moreover, $\mathcal{A} \subseteq [0,1]^n$ is apparently bounded. Therefore, $\mathcal{A}$ is a compact set. □

Now, we are ready to show that the maximum tuple of bid-discount multipliers exists by reasoning that a buyer's payment decreases when other buyers' discount multiplier increases and then applying Lemma E.3.

**Lemma E.4.** There exists a maximum tuple of bid-discount multipliers $\boldsymbol{\alpha}^{\max}$, i.e., for any feasible tuple of bid-discount multipliers $\boldsymbol{\alpha}$, $\alpha_i^{\max} \geq \alpha_i$ for any $1 \leq i \leq n$.

PROOF OF LEMMA E.4. First, for any given $\mathcal{M}^{\mathrm{BDF}}(\boldsymbol{\alpha}^{(1)})$ and $\mathcal{M}^{\mathrm{BDF}}(\boldsymbol{\alpha}^{(2)})$, define $\boldsymbol{\alpha}^{\mathrm{h}}$ be the entry-wise maximum of $\boldsymbol{\alpha}^{(1)}$ and $\boldsymbol{\alpha}^{(2)}$. We will show that $\boldsymbol{\alpha}^{\mathrm{h}}$ is also a feasible tuple of bid-discount multipliers.

We only need to verify that the budget-feasible constraint is met for any buyer, and we prove this by showing that a buyer's expected payment in $\mathcal{M}^{\mathrm{BDF}}(\boldsymbol{\alpha}^{\mathrm{h}})$ is no more than her expected payment in the higher of $\mathcal{M}^{\mathrm{BDF}}(\boldsymbol{\alpha}^{(1)})$ and $\mathcal{M}^{\mathrm{BDF}}(\boldsymbol{\alpha}^{(2)})$. For some buyer $i$, assume that $\alpha_i^{\mathrm{h}} = \alpha_i^{(1)}$ without loss of generality. Note that the payment is irrelevant with the discount multipliers given the allocation. Thus it suffices to show that when $q_1, \ldots, q_n$ are fixed, if buyer $i$ does not win in $\mathcal{M}^{\mathrm{BDF}}(\boldsymbol{\alpha}^{(1)})$, she does not win in $\mathcal{M}^{\mathrm{BDF}}(\boldsymbol{\alpha}^{\mathrm{h}})$ as well. Now that

buyer $i$ does not win in $\mathcal{M}^{\mathrm{BDF}}(\boldsymbol{\alpha}^{(1)})$, the highest discounted bid in $\mathcal{M}^{\mathrm{BDF}}(\boldsymbol{\alpha}^{(1)})$ must be higher than the discounted bid of buyer $i$. Since buyer $i$'s discounted bid in $\mathcal{M}^{\mathrm{BDF}}(\boldsymbol{\alpha}^{\mathrm{h}})$ is the same as in $\mathcal{M}^{\mathrm{BDF}}(\boldsymbol{\alpha}^{(1)})$, and the highest discounted bid in $\mathcal{M}^{\mathrm{BDF}}(\boldsymbol{\alpha}^{\mathrm{h}})$ is no less than the highest discounted bid in $\mathcal{M}^{\mathrm{BDF}}(\boldsymbol{\alpha}^{(1)})$, we conclude that buyer $i$ does not win in $\mathcal{M}^{\mathrm{BDF}}(\boldsymbol{\alpha}^{\mathrm{h}})$.

We now complete the proof of the lemma. Let $\alpha_i^{\max} = \sup\{\alpha_i \mid \boldsymbol{\alpha} \text{ is budget-feasible}\}$. We will show that $\boldsymbol{\alpha}^{\max} = (\alpha_i^{\max})_{1 \le i \le n}$ is also a feasible tuple of multipliers. In fact, for any $\epsilon > 0$ and any $1 \le i \le n$, there exists a feasible $\boldsymbol{\alpha}$ such that $\alpha_i > \alpha_i^{\max} - \epsilon$. By repeatedly taking the component-wise maximum for all $i$, there is a feasible $\boldsymbol{\alpha}^\epsilon$ such that for every $i$, $\alpha_i^\epsilon > \alpha_i^{\max} - \epsilon$. Thus the sequence $\boldsymbol{\alpha}^\epsilon$ (as $\epsilon \to 0$) has a limit point $\boldsymbol{\alpha}^{\max}$. By Lemma E.3, this limit point is also a feasible tuple of multipliers. □

Next, We demonstrate that the bid-discount mechanism induced by the maximum tuple of multipliers is budget-extracting.

**Lemma E.5.** $\mathcal{M}^{\mathrm{BDF}}(\boldsymbol{\alpha}^{\max})$ *is budget-extracting.*

PROOF OF LEMMA E.5. Prove by contradiction. Now suppose $\mathcal{M}^{\mathrm{BDF}}(\boldsymbol{\alpha}^{\max})$ is not budget-extracting, which means there is a buyer $i$ such that $\alpha_i^{\max} < 1$ and her budget is not binding, i.e., her expected payment is strictly less than her budget. By Lemma E.1, we can slightly increase $\alpha_i^{\max}$ while buyer $i$'s budget is still not binding. Note that when buyer $i$'s multiplier increases, the payment of any other buyer does not increase. Hence the budget-feasible constraint is still met for all buyers, and we obtain a feasible tuple with a strictly larger component, which contradicts the assumption that $\boldsymbol{\alpha}^{\max}$ is the maximum feasible tuple of multipliers. □

We have now already proved the first two statements, which claim that the maximum tuple of bid-discount multipliers $\boldsymbol{\alpha}^{\max}$ exists as well as its budget-extracting. Before proving the remaining statements, we present a critical observation in Lemma E.6. That is, given a budget-extracting bid-discount multiplier tuple $\boldsymbol{\alpha}^{\mathrm{e}}$ and another feasible tuple $\boldsymbol{\alpha}$, if there is some buyer $l$ with minimum $\alpha_l/\alpha_l^{\mathrm{e}}$ that satisfies $\alpha_l < 1$, and another buyer $k$ with a larger $\alpha_k/\alpha_k^{\mathrm{e}}$ that has a positive payment in $\mathcal{M}^{\mathrm{BDF}}(\boldsymbol{\alpha}^{\mathrm{e}})$, then $\mathcal{M}^{\mathrm{BDF}}(\boldsymbol{\alpha})$ is not budget-extracting. The insight of this observation is that for any quantile profile $\boldsymbol{q}$, buyer $l$ does not win in $\mathcal{M}^{\mathrm{BDF}}(\boldsymbol{\alpha})$ as long as she does not get allocated in $\mathcal{M}^{\mathrm{BDFPA}}(\boldsymbol{\alpha}^{\mathrm{e}})$. Moreover, buyer $k$ overbids buyer $l$ in $\mathcal{M}^{\mathrm{BDF}}(\boldsymbol{\alpha})$ on some quantile profiles with positive measure on which buyer $l$ wins in $\mathcal{M}^{\mathrm{BDF}}(\boldsymbol{\alpha}^{\mathrm{e}})$. Therefore, buyer $l$ strictly pays less in $\mathcal{M}^{\mathrm{BDF}}(\boldsymbol{\alpha})$ than in $\mathcal{M}^{\mathrm{BDF}}(\boldsymbol{\alpha}^{\mathrm{e}})$ in expectation, rendering that $\mathcal{M}^{\mathrm{BDF}}(\boldsymbol{\alpha})$ is not budget-extracting.

**Lemma E.6.** *Let $\boldsymbol{\alpha}^{\mathrm{e}}$ be a budget-extracting feasible tuple of bid-discount multipliers, and $\boldsymbol{\alpha}' \le \boldsymbol{\alpha}^{\mathrm{e}}$ be another feasible one. Let $\mathcal{I} = \arg\min_i \alpha_i'/\alpha_i^{\mathrm{e}}$. If there is some $l \in \mathcal{I}$ such that $\alpha_l' < 1$, and $k \notin \mathcal{I}$ such that the payment of buyer $k$ in $\mathcal{M}^{\mathrm{BDF}}(\boldsymbol{\alpha}^{\mathrm{e}})$ is positive, then $\boldsymbol{\alpha}'$ is not budget-extracting.*

PROOF OF LEMMA E.6. Prove by contradiction. Suppose $\boldsymbol{\alpha}'$ is budget-extracting instead. Since buyer $l$'s bid-discount multiplier is cut the most fraction from $\boldsymbol{\alpha}^{\mathrm{e}}$ to $\boldsymbol{\alpha}'$, when she does not win in $\mathcal{M}^{\mathrm{BDF}}(\boldsymbol{\alpha}^{\mathrm{e}})$ with quantile profile $\boldsymbol{q}$, she does not win the item in $\mathcal{M}^{\mathrm{BDF}}(\boldsymbol{\alpha}')$ as well. Thereby $p_l^{\mathrm{e}} \ge p_l'$, where $p_l'$ and $p_l^{\mathrm{e}}$ denote buyer $l$'s expected payment in $\mathcal{M}^{\mathrm{BDF}}(\boldsymbol{\alpha}')$ and $\mathcal{M}^{\mathrm{BDF}}(\boldsymbol{\alpha}^{\mathrm{e}})$ respectively. Now it suffices to show that $p_l' < p_l^{\mathrm{e}}$, which, combining $\alpha_l < 1$, is inconsistent with the fact that $\boldsymbol{\alpha}'$ is budget-extracting.

By definition, we have

$$p_l^{\mathrm{e}} - p_l' = \int_0^1 \widetilde{v}_l(q_l) \left( \int_{q_{-l}} \left( \Phi_l(\boldsymbol{\alpha}^{\mathrm{e}}, \widetilde{\boldsymbol{v}}, \boldsymbol{q}) - \Phi_l(\boldsymbol{\alpha}', \widetilde{\boldsymbol{v}}, \boldsymbol{q}) \right) \, \mathrm{d}q_{-l} \right) \mathrm{d}q_l$$

$$= \int_0^1 \widetilde{v}_l(q_l) \left( \int_0^1 \left( \Phi_{l>k}(\boldsymbol{\alpha}^{\mathrm{e}}, q_l, q_k) - \Phi_{l>k}(\boldsymbol{\alpha}', q_l, q_k) \right) \, \mathrm{d}q_k \right) \mathrm{d}q_l.$$

Here, $\Phi_{l>k}(\boldsymbol{\alpha}, q_l, q_k)$ is defined as $\int_{\boldsymbol{q}_{-\{l,k\}}} \Phi_l(\boldsymbol{\alpha}, \widetilde{\boldsymbol{v}}, \boldsymbol{q}) \, \mathrm{d}q_{-\{l,k\}}$, which represents the probability that buyer $l$ wins the item given $q_l$ and $q_k$. We implicitly take $\widetilde{\boldsymbol{v}}$ as fixed. Further, define

$$H(\boldsymbol{\alpha}, \underline{\eta}, \bar{\eta}, \underline{\theta}, \bar{\theta}) := \int_{\underline{\eta}}^{\bar{\eta}} \widetilde{v}_l(q_l) \left( \int_{\underline{\theta}}^{\bar{\theta}} \Phi_{l>k}(\boldsymbol{\alpha}, q_l, q_k) \, \mathrm{d}q_k \right) \mathrm{d}q_l$$

as the expected payment of buyer $l$ in $\mathcal{M}^{\mathrm{BDF}}(\boldsymbol{\alpha})$ when $q_l$ and $q_k$ range from $[\underline{\eta}, \bar{\eta}]$ and $[\underline{\theta}, \bar{\theta}]$ respectively. Notice that for any $\boldsymbol{\alpha}$, $0 \le \eta_1 \le \eta_3 \le \eta_2 \le 1$ and $0 \le \theta_1 \le \theta_3 \le \theta_2 \le 1$,

$$H(\boldsymbol{\alpha}, \eta_1, \eta_2, \theta_1, \theta_2) = H(\boldsymbol{\alpha}, \eta_1, \eta_3, \theta_1, \theta_2) + H(\boldsymbol{\alpha}, \eta_3, \eta_2, \theta_1, \theta_2)$$
$$= H(\boldsymbol{\alpha}, \eta_1, \eta_2, \theta_1, \theta_3) + H(\boldsymbol{\alpha}, \eta_1, \eta_2, \theta_3, \theta_2).$$

The remaining proof of Lemma E.6 is divided into two parts. We first demonstrate that for any $0 \le \eta_1 \le \eta_2 \le 1$ and $0 \le \theta_1 \le \theta_2 \le 1$, we have $H(\boldsymbol{\alpha}^{\mathrm{e}}, \eta_1, \eta_2, \theta_1, \theta_2) - H(\boldsymbol{\alpha}', \eta_1, \eta_2, \theta_1, \theta_2) \ge 0$. Then we find $\eta_1^0, \eta_2^0, \theta_1^0, \theta_2^0$ such that $H(\boldsymbol{\alpha}^{\mathrm{e}}, \eta_1^0, \eta_2^0, \theta_1^0, \theta_2^0) - H(\boldsymbol{\alpha}', \eta_1^0, \eta_2^0, \theta_1^0, \theta_2^0) > 0$. The above collaboratively implies that

$$p_l^{\mathrm{e}} - p_l' = H(\boldsymbol{\alpha}^{\mathrm{e}}, 0, 1, 0, 1) - H(\boldsymbol{\alpha}', 0, 1, 0, 1) \ge H(\boldsymbol{\alpha}^{\mathrm{e}}, \eta_1^0, \eta_2^0, \theta_1^0, \theta_2^0) - H(\boldsymbol{\alpha}', \eta_1^0, \eta_2^0, \theta_1^0, \theta_2^0) > 0,$$

which concludes the proof of Lemma E.6.

For the first part, with the observation that

$$H(\boldsymbol{\alpha}^{\mathrm{e}}, \eta_1, \eta_2, \theta_1, \theta_2) - H(\boldsymbol{\alpha}', \eta_1, \eta_2, \theta_1, \theta_2)$$

$$= \int_{\eta_1}^{\eta_2} \widetilde{v}_l(q_l) \left( \int_{\theta_1}^{\theta_2} \left( \Phi_{l>k}(\boldsymbol{\alpha}^{\mathrm{e}}, q_l, q_k) - \Phi_{l>k}(\boldsymbol{\alpha}', q_l, q_k) \right) \, \mathrm{d}q_k \right) \mathrm{d}q_l,$$

it suffices to prove for any $q_l \in [0,1], q_k \in [0,1]$,

$$\Phi_{l>k}(\boldsymbol{\alpha}^{\mathrm{e}}, q_l, q_k) - \Phi_{l>k}(\boldsymbol{\alpha}', q_l, q_k) \geq 0. \tag{29}$$

Recall that the two terms in (29) are the probability that buyer $l$ wins the item in $\mathcal{M}^{\mathrm{BDF}}(\boldsymbol{\alpha}^{\mathrm{e}})$ and $\mathcal{M}^{\mathrm{BDF}}(\boldsymbol{\alpha})$ when $q_k$ and $q_l$ are fixed, respectively. Given quantile profile $\boldsymbol{q}$, since $l \in \mathcal{I} = \arg\min_i \alpha_i'/\alpha_i^{\mathrm{e}}$ and $\boldsymbol{\alpha}' \leq \boldsymbol{\alpha}^{\mathrm{e}}$, then by the allocation rule, if $l$ wins in $\mathcal{M}^{\mathrm{BDF}}(\boldsymbol{\alpha})$, she wins in $\mathcal{M}^{\mathrm{BDF}}(\boldsymbol{\alpha}^{\mathrm{e}})$ as well. As a result, when $q_k$ and $q_l$ are fixed, buyer $l$ certainly does not have less probability to win in $\mathcal{M}^{\mathrm{BDF}}(\boldsymbol{\alpha}^{\mathrm{e}})$ than in $\mathcal{M}^{\mathrm{BDF}}(\boldsymbol{\alpha})$.

Now we establish the existence of $\eta_1^0 < \eta_2^0$ and $\theta_1^0 < \theta_2^0$ such that $H(\boldsymbol{\alpha}^{\mathrm{e}}, \eta_1^0, \eta_2^0, \theta_1^0, \theta_2^0) > H(\boldsymbol{\alpha}', \eta_1^0, \eta_2^0, \theta_1^0, \theta_2^0)$.

The budget-extracting property of $\boldsymbol{\alpha}'$ and that $\alpha_l' < 1$ in together imply $p_l' = B_l > 0$. Since $p_l^{\mathrm{e}} \geq p_l'$, we have $p_l^{\mathrm{e}} = B_l > 0$. As a result, there are $q_l^{(1)} < 1$ and $q_k^{(1)} > 0$ such that $\Phi_{l>k}(\boldsymbol{\alpha}^{\mathrm{e}}, q_l^{(1)}, q_k^{(1)}) > 0$.[4] Symmetrically, since $p_k^{\mathrm{e}}$ is positive as well, there are $q_k^{(2)} < 1$ and $q_l^{(2)} > 0$ such that $\Phi_{k>l}(\boldsymbol{\alpha}^{\mathrm{e}}, q_k^{(2)}, q_l^{(2)}) > 0$. We can further assume that $0 < q_l^{(2)} \leq q_l^{(1)} < 1$ and $0 < q_k^{(1)} \leq q_k^{(2)} < 1$, or else, we can swap $q_l^{(1)}$ and $q_l^{(2)}$ or $q_k^{(1)}$ and $q_k^{(2)}$ without breaking the above statements. We want to find $q_k^{(3)}$ and $q_l^{(3)}$ such that $\alpha_l^{\mathrm{e}}\widetilde{v}_l(q_l^{(3)}) = \alpha_k^{\mathrm{e}}\widetilde{v}_k(q_k^{(3)})$, and the probability that $l$ wins with $q_l^{(3)}$ under $\boldsymbol{\alpha}^{\mathrm{e}}$ is positive. We construct as follows:

- If $\alpha_k^{\mathrm{e}}\widetilde{v}_k(q_k^{(2)}) \geq \alpha_l^{\mathrm{e}}\widetilde{v}_l(q_l^{(1)})$, let $q_l^{(3)} = q_l^{(1)}$, and there exists $q_k^{(3)} \in [q_k^{(1)}, q_k^{(2)}]$ such that $\alpha_k^{\mathrm{e}}\widetilde{v}_k(q_k^{(3)}) = \alpha_l^{\mathrm{e}}\widetilde{v}_l(q_l^{(3)})$ due to the continuity of $\widetilde{v}_k(q_k)$ and that $\alpha_l^{\mathrm{e}}\widetilde{v}_l(q_l^{(1)}) \geq \alpha_k^{\mathrm{e}}\widetilde{v}_k(q_k^{(1)})$. $l$ wins with positive probability with $q_l^{(3)}$ since $q_l^{(3)} = q_l^{(1)}$.
- If $\alpha_k^{\mathrm{e}}\widetilde{v}_k(q_k^{(2)}) < \alpha_l^{\mathrm{e}}\widetilde{v}_l(q_l^{(1)})$, let $q_k^{(3)} = q_k^{(2)}$, and there exists $q_l^{(3)} \in [q_l^{(2)}, q_l^{(1)}]$ such that $\alpha_l^{\mathrm{e}}\widetilde{v}_l(q_l^{(3)}) = \alpha_k^{\mathrm{e}}\widetilde{v}_k(q_k^{(3)})$ due to the continuity of $\widetilde{v}_l(q_l)$ and that $\alpha_k^{\mathrm{e}}\widetilde{v}_k(q_k^{(2)}) \geq \alpha_l^{\mathrm{e}}\widetilde{v}_l(q_l^{(2)})$. $l$ wins with positive probability with $q_l^{(3)}$ since $k$ wins with positive probability with $q_k^{(3)} = q_k^{(2)}$.

Moreover, as $\alpha_l'/\alpha_l^{\mathrm{e}} < \alpha_k'/\alpha_k^{\mathrm{e}}$, there exists a sufficiently small $\delta > 0$ such that for any $q_k \in [q_k^{(3)} - \delta, q_k^{(3)}]$ and $q_l \in [q_l^{(3)}, q_l^{(3)} + \delta]$ (note that $q_k^{(3)} < 1$ and $q_l^{(3)} > 0$), the probability that $l$ wins with $q_l$ under $\boldsymbol{\alpha}^{\mathrm{e}}$ is no less than a positive constant, and

$$\alpha_k^{\mathrm{e}}\widetilde{v}_k(q_k) \cdot \frac{\alpha_k'}{\alpha_k^{\mathrm{e}}} > \alpha_l^{\mathrm{e}}\widetilde{v}_l(q_l) \cdot \frac{\alpha_l'}{\alpha_l^{\mathrm{e}}},$$

that is, $\alpha_k'\widetilde{v}_k(q_k) > \alpha_l'\widetilde{v}_l(q_l)$. Moreover, for any $q_k \in [q_k^{(3)} - \delta, q_k^{(3)}], q_l \in [q_l^{(3)}, q_l^{(3)} + \delta]$, we have $\alpha_l^{\mathrm{e}}\widetilde{v}_l(q_l) \geq \alpha_l^{\mathrm{e}}\widetilde{v}_l(q_l^{(3)}) = \alpha_k^{\mathrm{e}}\widetilde{v}_k(q_k^{(3)}) \geq \alpha_k^{\mathrm{e}}\widetilde{v}_k(q_k)$. Therefore,

$$H(\boldsymbol{\alpha}^{\mathrm{e}}, q_l^{(3)}, q_l^{(3)} + \delta, q_k^{(3)} - \delta, q_k^{(3)}) = \int_{q_l^{(3)}}^{q_l^{(3)}+\delta} \widetilde{v}_l(q_l) \left( \int_{q_k^{(3)}-\delta}^{q_k^{(3)}} \Phi_{l>k}(\boldsymbol{\alpha}^{\mathrm{e}}, q_l, q_k) \, \mathrm{d}q_k \right) \mathrm{d}q_l > 0,$$

$$H(\boldsymbol{\alpha}', q_l^{(3)}, q_l^{(3)} + \delta, q_k^{(3)} - \delta, q_k^{(3)}) = \int_{q_l^{(3)}}^{q_l^{(3)}+\delta} \widetilde{v}_l(q_l) \left( \int_{q_k^{(3)}-\delta}^{q_k^{(3)}} \Phi_{l>k}(\boldsymbol{\alpha}', q_l, q_k) \, \mathrm{d}q_k \right) \mathrm{d}q_l = 0.$$

Taking $\eta_1^0 = q_l^{(3)}, \eta_2^0 = q_l^{(3)} + \delta, \theta_1^0 = q_k^{(3)} - \delta, \theta_2^0 = q_k^{(3)}$ finishes this part, and the above collaboratively concludes the proof of Lemma E.6. □

With the help of Lemma E.6, we can characterize a budget-extracting bid-discount multiplier tuple by comparing it with the maximum tuple $\boldsymbol{\alpha}^{\max}$.

**Lemma E.7.** *For any budget-extracting bid-discount multiplier tuple $\boldsymbol{\alpha}^{\mathrm{e}}$, the following two conditions are satisfied:*

- *There exists some $v \leq 1$, such that for any $1 \leq i \leq n$ satisfying $p_i^{\max}$ (buyer $i$'s expected payment in $\mathcal{M}^{\mathrm{BDF}}(\boldsymbol{\alpha}^{\max})$) is positive, $\alpha_i^{\mathrm{e}}/\alpha_i^{\max} = v$;*
- *For any $1 \leq i \leq n$ satisfying $p_i^{\max} = 0$, $p_i^{\mathrm{e}} = 0$ (buyer $i$ never wins in $\mathcal{M}^{\mathrm{BDF}}(\boldsymbol{\alpha}^{\mathrm{e}})$) and $\alpha_i^{\mathrm{e}} = \alpha_i^{\max} = 1$.*

PROOF OF LEMMA E.7. Let $\mathcal{I}_1 = \{1 \leq i \leq n \mid p_i^{\max} > 0\}$ be the set of buyers whose payment in $\mathcal{M}^{\mathrm{BDF}}(\boldsymbol{\alpha}^{\max})$ are positive, and $\mathcal{I}_2 = [n] \setminus \mathcal{I}_1$ be the set of buyers whose payment in $\mathcal{M}^{\mathrm{BDF}}(\boldsymbol{\alpha}^{\max})$ are 0. For a budget-extracting tuple of bid-discount multipliers $\boldsymbol{\alpha}^{\mathrm{e}}$ different from $\boldsymbol{\alpha}^{\max}$, we have $\alpha_i^{\mathrm{e}} \leq \alpha_i^{\max}$ for all $i$, with the inequality holds for at least one buyer. Define $\mathcal{I} := \arg\min_i \alpha_i^{\mathrm{e}}/\alpha_i^{\max}$ as the set of buyers whose

---

[4]Otherwise, the Lebesgue measure of quantile profiles that $l$ wins is zero, contradicting that the expected payment of $l$ is positive.

bid-discount multipliers are cut the most from $\boldsymbol{\alpha}^{\max}$ to $\boldsymbol{\alpha}^{\mathrm{e}}$. Note that $\min_i \alpha_i^{\mathrm{e}}/\alpha_i^{\max} < 1$. Then we have $\mathcal{I}_2 \cap \mathcal{I} = \emptyset$, since otherwise every buyer in $\mathcal{I}_2 \cap \mathcal{I}$ has smaller bid-discount multiplier in $\mathcal{M}^{\mathrm{BDF}}(\boldsymbol{\alpha}^{\mathrm{e}})$ than in $\mathcal{M}^{\mathrm{BDF}}(\boldsymbol{\alpha}^{\max})$, whereas her payment remains 0 in $\mathcal{M}^{\mathrm{BDF}}(\boldsymbol{\alpha}^{\mathrm{e}})$, contradicting that $\boldsymbol{\alpha}^{\mathrm{e}}$ is budget-extracting.

If $\mathcal{I}_1 \neq \mathcal{I}$, let $l \in \mathcal{I}$ and $k \in \mathcal{I}_1 \setminus \mathcal{I}$. Since $\alpha_l^{\mathrm{e}} < \alpha_l^{\max} \leq 1$, $p_k^{\max} > 0$ and $\alpha_l^{\mathrm{e}}/\alpha_l^{\max} < \alpha_k^{\mathrm{e}}/\alpha_k^{\max}$, by Lemma E.6 we derive a contradiction that $\boldsymbol{\alpha}^{\mathrm{e}}$ is not budget-extracting. Thus $\mathcal{I}_1 = \mathcal{I}$ must hold, which gives the first statement.

Moreover, if there exists $k \in \mathcal{I}_2$ such that $p_k^{\mathrm{e}} > 0$, let $l$ be an arbitrary buyer in $\mathcal{I}_1$. Applying Lemma E.6, we conclude that $\boldsymbol{\alpha}^{\max}$ is not budget-extracting, which contradicts the assumption. Hence $p_i^{\mathrm{e}} = 0$ for every $i \in \mathcal{I}_2$. This implies the second statement. □

The properties of budget-extracting BDFPA presented in Lemma E.7 are sufficient to show that all budget-extracting BDFPAs bring the same payment for each buyer.

**Lemma E.8.** *All budget-extracting BDFPAs bring the same payment for each buyer.*

Proof of Lemma E.8. Suppose $\boldsymbol{\alpha}^{\mathrm{e}}$ is a budget-extracting bid-discount tuple different from $\boldsymbol{\alpha}^{\max}$. Now by Lemma E.7, we know that the buyers with payment 0 in $\mathcal{M}^{\mathrm{BDF}}(\boldsymbol{\alpha}^{\max})$ have payment 0 in $\mathcal{M}^{\mathrm{BDF}}(\boldsymbol{\alpha}^{\mathrm{e}})$ as well. As for those buyers with positive payment in $\mathcal{M}^{\mathrm{BDF}}(\boldsymbol{\alpha}^{\max})$ (i.e., in $\mathcal{I}_1$), the corresponding ratios $\alpha_i^{\mathrm{e}}/\alpha_i^{\max}$ are identical, which are strictly less than 1. This indicates that these buyers' budgets are all binding in $\mathcal{M}^{\mathrm{BDF}}(\boldsymbol{\alpha}^{\mathrm{e}})$ since $\boldsymbol{\alpha}^{\mathrm{e}}$ is budget-extracting. Meanwhile, we claim that for any buyer in $\mathcal{I}_1$, her payment in $\mathcal{M}^{\mathrm{BDF}}(\boldsymbol{\alpha}^{\mathrm{e}})$ is no more than her payment in $\mathcal{M}^{\mathrm{BDF}}(\boldsymbol{\alpha}^{\max})$. In fact, any buyer in $\mathcal{I}_1$ cannot win on more quantile profiles in $\mathcal{M}^{\mathrm{BDF}}(\boldsymbol{\alpha}^{\mathrm{e}})$ than in $\mathcal{M}^{\mathrm{BDF}}(\boldsymbol{\alpha}^{\max})$. Therefore, buyers in $\mathcal{I}_1$ also exhaust their budgets in $\mathcal{M}^{\mathrm{BDF}}(\boldsymbol{\alpha}^{\max})$. □

Finally, we present the proof of the last statement, which gives the necessary and sufficient conditions for the uniqueness of a budget-extracting bid-discount multiplier tuple.

**Lemma E.9.** $\boldsymbol{\alpha}^{\max}$ *is the unique budget-extracting tuple of bid-discount multipliers if and only if either one of the following two conditions is satisfied:*

1. $\max_{i \in \mathcal{I}_1} \alpha_i^{\max} \widetilde{v}_i(0) \leq \max_{i \in \mathcal{I}_2} \{\alpha_i^{\max} \widetilde{v}_i(1), \lambda\}$, *where* $\mathcal{I}_1 = \{i \mid p_i^{\max} > 0\}$ *and* $\mathcal{I}_2 = [n] \setminus \mathcal{I}_1$, *or*
2. *there exists* $i \in \mathcal{I}_1$ *such that* $p_i^{\max} < \rho_i$.

Proof of Lemma E.9. We prove the two sides respectively.

*"If" side.* The proof of Lemma E.8 implies that if there is a budget-extracting tuple other than $\boldsymbol{\alpha}^{\max}$, then the budgets of the buyers with positive payments in $\mathcal{M}^{\mathrm{BDF}}(\boldsymbol{\alpha}^{\max})$ are binding. In other words, if there exists $i \in \mathcal{I}_1$ such that $p_i^{\max} < \rho_i$ (which is the second condition), then $\boldsymbol{\alpha}^{\max}$ must be the unique budget-extracting tuple.

Furthermore, if $\max_{i \in \mathcal{I}_1} \alpha_i^{\max} \widetilde{v}_i(0) \leq \max_{i \in \mathcal{I}_2} \{\alpha_i^{\max} \widetilde{v}_i(1), \lambda\}$, suppose there is another budget-extracting tuple $\boldsymbol{\alpha}^{\mathrm{e}}$ different from $\boldsymbol{\alpha}^{\max}$. By Lemma E.7, for any $i \in \mathcal{I}_2$, we have $p_i^{\max} = p_i^{\mathrm{e}} = 0$ and $\alpha_i^{\max} = \alpha_i^{\mathrm{e}} = 1$. Also, there exists $0 < \nu < 1$ such that for any $i \in \mathcal{I}_1$, we have $\alpha_i^{\mathrm{e}} = \nu \alpha_i^{\max}$. Note that since the payment of each buyer in $\mathcal{I}_1$ is non-zero, we have $\max_{i \in \mathcal{I}_1} \alpha_i^{\max} \widetilde{v}_i(1) > \max_{i \in \mathcal{I}_2} \{\alpha_i^{\max} \widetilde{v}_i(1), \lambda\}$. Therefore, by the continuity and strict monotonicity of quantile functions, as well as noticing that $\max_{i \in \mathcal{I}_2} \{\alpha_i^{\max} \widetilde{v}_i(1), \lambda\} \geq \lambda > 0$, we derive that for some $r \in (0, 1]$,

$$\max_{i \in \mathcal{I}_1} \alpha_i^{\mathrm{e}} \widetilde{v}_i(r) < \max_{i \in \mathcal{I}_2} \{\alpha_i^{\mathrm{e}} \widetilde{v}_i(1), \lambda\} = \max_{i \in \mathcal{I}_2} \{\alpha_i^{\max} \widetilde{v}_i(1), \lambda\} < \max_{i \in \mathcal{I}_1} \alpha_i^{\max} \widetilde{v}_i(r).$$

Therefore, we state that under some quantile profiles with positive measure, no buyer in $\mathcal{I}_1$ wins when the multiplier tuple is $\boldsymbol{\alpha}^{\mathrm{e}}$ but some buyer in $\mathcal{I}_1$ wins with $\boldsymbol{\alpha}^{\max}$. Meanwhile, the reverse case never happens since $\alpha_i^{\mathrm{e}} = \alpha_i^{\max}$ when $i \in \mathcal{I}_2$ while $\alpha_i^{\mathrm{e}} < \alpha_i^{\max}$ when $i \in \mathcal{I}_1$. This indicates that the total payment of buyers in $\mathcal{I}_1$ is strictly cut from $\mathcal{M}^{\mathrm{BDF}}(\boldsymbol{\alpha}^{\max})$ to $\mathcal{M}^{\mathrm{BDF}}(\boldsymbol{\alpha}^{\mathrm{e}})$, contradicting that $\boldsymbol{\alpha}^{\mathrm{e}}$ is budget-extracting, as the budget of at least one buyer in $\mathcal{I}_1$ is not binding.

*"Only if" side.* We prove by contradiction for this part. We suppose that $\max_{i \in \mathcal{I}_1} \alpha_i^{\max} \widetilde{v}_i(0) > \max_{i \in \mathcal{I}_2} \{\alpha_i^{\max} \widetilde{v}_i(1), \lambda\}$, $p_i^{\max} = \rho_i$ for any $i \in \mathcal{I}_1$, and $p_i^{\max} = 0$ for any $i \in \mathcal{I}_2$ reversely. As a result, there exists some $0 < \nu < 1$ such that $\nu \cdot \max_{i \in \mathcal{I}_1} \alpha_i^{\max} \widetilde{v}_i(0) > \max_{i \in \mathcal{I}_2} \{\alpha_i^{\max} \widetilde{v}_i(1), \lambda\}$. Define $\boldsymbol{\alpha} = (\alpha_i)_{1 \leq i \leq n}$ as

$$\alpha_i = \begin{cases} \nu \alpha_i^{\max} & i \in \mathcal{I}_1 \\ 0 & i \in \mathcal{I}_2. \end{cases}$$

Now we show that $\boldsymbol{\alpha}$ is budget-extracting. On the one hand, for $\mathcal{M}^{\mathrm{BDF}}(\boldsymbol{\alpha})$, the maximum discounted bid of buyers in $\mathcal{I}_2$ is less than the minimum discounted bid of buyers in $\mathcal{I}_1$, therefore, any buyer in $\mathcal{I}_2$ does not win at all in all quantile profiles. Meanwhile, the item is always allocated as $\nu \cdot \max_{i \in \mathcal{I}_1} \alpha_i^{\max} \widetilde{v}_i(0) > \lambda$. On the other hand, the bid-discount multipliers of buyers in $\mathcal{I}_1$ are scaled by the same constant from $\boldsymbol{\alpha}^{\max}$ to $\boldsymbol{\alpha}$, thus the ordering of buyers in $\mathcal{I}_1$ remains unchanged in all quantile profiles from $\mathcal{M}^{\mathrm{BDF}}(\boldsymbol{\alpha}^{\max})$ to $\mathcal{M}^{\mathrm{BDF}}(\boldsymbol{\alpha})$. This reasoning implies that payments of all buyers stay the same, and budget-extracting still holds for $\boldsymbol{\alpha}$. Therefore, $\boldsymbol{\alpha}^{\max}$ is not the unique budget-extracting tuple. □

The proof of Theorem A.1 is finished by putting Lemmas E.4, E.5, E.7, E.8 and E.9 together.

## E.2 Proof of Theorem A.2

Notice that $\chi^{\text{BDF}}(\tau)$, which defined in the proof of Theorem C.1 is a convex function by Theorem 7.46 from Shapiro et al. [37]. Now consider the optimal solution of $\min_{\tau \in [0,1]^n} \chi^{\text{BD}}(\tau)$, $\tau^*$. By Lemma C.2, $\alpha = 1^n - \tau^*$ is a budget-extracting tuple of multipliers for BDFPA. This concludes the proof.

## E.3 Proof of Theorem A.3

We will adopt a similar methodology as in the proof of Theorem A.1. Let $\beta = (\beta_1, \ldots, \beta_n)$ be pacing multipliers and define $G_{\beta,i}$ as the cumulative distribution function of $\max_{i' \neq i}\{\beta_{i'}\widetilde{v}_{i'}(q'_i), \lambda\}$ where $q_{-i}$ is chosen uniformly in $[0,1]^{n-1}$. Then $G_{\beta,i}(\cdot)$ has the properties stated in Lemma E.2. Next, we prove an analog of Lemma E.1.

**Lemma E.10.** *There exists a constant $C$ which satisfies the following: For any $\beta = (\beta_1, \ldots, \beta_n)$ and $1 \leq i \leq n$ such that $\beta_i < 1$, let $\beta' = \beta + \delta e_i$ where $0 < \delta \leq 1 - \beta_i$ and $e_i$ is the vector with the $i$-th entry one and all other entries zero. Then the expected payment of buyer $i$ in $\mathcal{M}^{\text{PF}}(\beta')$ is at most the expected payment of buyer $i$ in $\mathcal{M}^{\text{PF}}(\beta)$ plus $C\delta$.*

PROOF OF LEMMA E.10. By definition, the expected payment of buyer $i$ in $\mathcal{M}^{\text{PF}}(\beta)$ is

$$\int_0^1 \beta_i \widetilde{v}_i(q_i) \cdot \left( \int_{q_{-i}} I\left[ \beta_i \widetilde{v}_i(q_i) \geq \max_{i' \neq i}\{\beta_{i'}\widetilde{v}_{i'}(q_{i'}), \lambda\} \right] \, dq_{-i} \right) \, dq_i.$$

Using the technique in (27) and (28), we upper bound the increment of buyer $i$'s expected payment after replacing $\beta$ with $\beta' = \beta + \delta e_i$ as

$$\int_0^1 (\beta_i + \delta)\widetilde{v}_i(q_i) \cdot \left( G_{\beta,i}\left((\beta_i + \delta)\widetilde{v}_i(q_i)\right) - G_{\beta,i}\left(\beta_i \widetilde{v}_i(q_i)\right) \right) \, dq_i$$

$$\leq \int_0^1 \widetilde{v}_i(q_i) \cdot \left( G_{\beta,i}\left((\beta_i + \delta)\widetilde{v}_i(q_i)\right) - G_{\beta,i}\left(\beta_i \widetilde{v}_i(q_i)\right) \right) \, dq_i.$$

The conclusion now follows directly from the same case analysis as in the proof of Lemma E.1. □

We then show that the set of budget feasible pacing multipliers is compact.

**Lemma E.11.** *Let $\mathcal{B}$ be the set of all $\beta$ such that $\mathcal{M}^{\text{PF}}(\beta)$ is a feasible pacing mechanism. Then $\mathcal{B}$ is compact.*

PROOF OF LEMMA E.11. Let $\varphi : [0,1]^n \to \mathbb{R}^n$ be defined as the map from the tuple of multipliers $\beta$ to the expected payment vector of all buyers. Then by Lemma E.10, $\varphi$ is Lipschitz continuous. Since $\mathcal{B} = \{\beta : \varphi(\beta) \in \prod_i [0, \rho_i]\}$ is the pre-image of $\prod_i [0, \rho_i]$ (which is certainly closed) under $\varphi$, $\mathcal{B}$ is closed. $\mathcal{B}$ is also bounded for $\mathcal{B} \subseteq [0,1]^n$. This concludes the proof of the Lemma. □

We are now able to establish the existence of a maximum tuple of pacing multipliers $\beta^{\text{max}}$.

**Lemma E.12.** *There exists a maximum tuple of pacing multipliers $\beta^{\text{max}}$, i.e., for any feasible tuple of pacing multipliers $\beta$, $\beta_i^{\text{max}} \geq \beta_i$, for any $1 \leq i \leq n$.*

PROOF OF LEMMA E.12. We first show that given any feasible multipliers $\beta^{(1)}, \beta^{(2)}$, the element-wise maximum $\beta^{\text{h}} = \max(\beta^{(1)}, \beta^{(2)})$ is still feasible.

We need to show that budget feasibility is met for each buyer. For any buyer $1 \leq i \leq n$, without loss generality, assume $\beta_i^{\text{h}} = \beta_i^{(1)}$, and we claim that buyer $i$'s payment in $\mathcal{M}^{\text{PF}}(\beta^{\text{h}})$ is no more than her payment in $\mathcal{M}^{\text{PF}}(\beta^{(1)})$. Note that for any quantile profile $q$, if buyer $i$ wins in $\mathcal{M}^{\text{PF}}(\beta^{\text{h}})$, she definitely wins in $\mathcal{M}^{\text{PF}}(\beta^{(1)})$ since $\beta^{\text{h}} \geq \beta^{(1)}$, and her payment would be identical in these two auctions as her multipliers are the same in these two tuples. Therefore the claim is shown.

Now let $\beta_i^{\text{max}} = \sup\{\beta_i \mid \beta \text{ is budget-feasible}\}$ for each $1 \leq i \leq n$. Resembling the argument in the proof of Lemma E.5, $\beta^{\text{max}} = (\beta_i^{\text{max}})_{1 \leq i \leq n}$ is also a feasible tuple of multipliers. This concludes the proof of the lemma. □

Now that we have established the existence of maximum multipliers, we show that it is the unique budget-extracting tuple. We first show it is budget-extracting in the proceeding lemma.

**Lemma E.13.** *$\mathcal{M}^{\text{PF}}(\beta^{\text{max}})$ is budget-extracting.*

PROOF OF LEMMA E.13. We prove the lemma by contradiction. Suppose $\mathcal{M}^{\text{PF}}(\beta^{\text{max}})$ is not budget-extracting, then there is some buyer $i$ such that $\beta_i^{\text{max}} < 1$ and her budget is not binding. By Lemma E.10, we can increase $\beta_i^{\text{max}}$ slightly so that buyer $i$'s budget is still not binding. Note that other buyers' payments will not increase when only buyer $i$'s multiplier increases. Therefore, this new tuple of multipliers is still feasible, which contradicts our definition of $\beta^{\text{max}}$ that it is the entry-wise supremum over all feasible $\beta$. □

It remains to show that $\beta^{\text{max}}$ is the unique budget-extracting tuple of multipliers.

**Lemma E.14.** *$\mathcal{M}^{\text{PF}}(\beta^{\text{max}})$ is the unique budget-extracting tuple of multipliers.*

PROOF OF LEMMA E.14. Let $\boldsymbol{\beta} \neq \boldsymbol{\beta}^{\max}$ be a feasible tuple of multipliers. We show that $\boldsymbol{\beta}$ is not budget-extracting by contradiction.

Suppose $\boldsymbol{\beta}$ is budget-extracting otherwise. Let $\mathcal{I}$ be the set of buyers such that for any $i \in \mathcal{I}$, $\beta_i < \beta_i^{\max}$, i.e., any buyer in $\mathcal{I}$ has a strictly smaller pacing multiplier in $\boldsymbol{\beta}$ than in $\boldsymbol{\beta}^{\max}$. Since for $i \in \mathcal{I}$, $\beta_i < \beta_i^{\max} \leq 1$, the expected payment of buyer $i$ equals to her budget in $\mathcal{M}^{\mathrm{PF}}(\boldsymbol{\beta})$ by the definition of budget-extracting. Hence, the Lebesgue measure of quantile profiles won by buyers in $\mathcal{I}$ is positive. Now consider buyers in $\mathcal{I}$ in $\mathcal{M}^{\mathrm{PF}}(\boldsymbol{\beta}^{\max})$. For any quantile profile won by some buyer in $\mathcal{I}$ in $\mathcal{M}^{\mathrm{PF}}(\boldsymbol{\beta})$, the quantile profile is also won by $\mathcal{I}$ in $\mathcal{M}^{\mathrm{PF}}(\boldsymbol{\beta}^{\max})$, as buyers outside $\mathcal{I}$ see no change in the paced bid from $\mathcal{M}^{\mathrm{PF}}(\boldsymbol{\beta})$ to $\mathcal{M}^{\mathrm{PF}}(\boldsymbol{\beta}^{\max})$. However, on these quantile profiles, buyers in $\mathcal{I}$ pay more in $\mathcal{M}^{\mathrm{PF}}(\boldsymbol{\beta}^{\max})$ than in $\mathcal{M}^{\mathrm{PF}}(\boldsymbol{\beta})$ with strictly higher pacing multipliers. Therefore the total payment of buyers in $\mathcal{I}$ strictly increases from $\mathcal{M}^{\mathrm{PF}}(\boldsymbol{\beta})$ to $\mathcal{M}^{\mathrm{PF}}(\boldsymbol{\beta}^{\max})$, and as a result, $\mathcal{M}^{\mathrm{PF}}(\boldsymbol{\beta}^{\max})$ is not budget-feasible. A contradiction. Hence $\boldsymbol{\beta}$ is not budget-extracting, and $\boldsymbol{\beta}^{\max}$ is the unique budget-extracting pacing multiplier tuple. □

Finally, we establish that $\boldsymbol{\beta}^{\max}$ maximizes the seller's revenue among all feasible tuples of pacing multipliers.

**Lemma E.15.** $\boldsymbol{\beta}^{\max}$ *maximizes the seller's revenue among all feasible tuples of pacing multipliers.*

PROOF OF LEMMA E.15. Note that for PFPA, the seller's revenue in $\mathcal{M}^{\mathrm{PF}}(\boldsymbol{\beta})$ equals to

$$\int_q \max_i \{\beta_i \widetilde{v}_i(q_i) - \lambda\}^+ \, \mathrm{d}q,$$

by definition, which, increases with any entry of $\boldsymbol{\beta}$. Now that $\boldsymbol{\beta}^{\max}$ is defined as the supremum over all feasible tuples, it extracts no lower revenue for the seller than any other feasible tuple. □

Synthesizing Lemmas E.12, E.13, E.14 and E.15, we conclude the proof of Theorem A.3.

# F PROOF OF THEOREM A.4

We prove the three statements in the theorem in order.

*eBDPFA $\succeq$ BROA.* Note that when $(\widetilde{v}_i)_{1 \leq i \leq n}$ is strictly regular, by Balseiro et al. [3], the seller's expected revenue in BROA is the value of programming (3), or

$$\min_{\boldsymbol{\gamma} \in [0,1]^n} \chi^{\mathrm{BRO}}(\boldsymbol{\gamma}) := \left\{ \mathbb{E}_q \left[ \max_i \left\{ \gamma_i \widetilde{\psi}_i(q_i) - \lambda \right\}^+ \right] + \sum_{i=1}^n (1 - \gamma_i) \rho_i \right\}.$$

On the other side, by the strong duality result that we give in the proof of Theorem C.1, when $(\widetilde{v}_i)_{1 \leq i \leq n}$ is strictly increasing, the seller's revenue in eBDFPA equals to

$$\min_{\boldsymbol{\tau} \in [0,1]^n} \chi^{\mathrm{BDF}}(\boldsymbol{\tau}) = \left\{ \mathbb{E}_q \left[ \max_i \{(1 - \tau_i)\widetilde{v}_i(q_i) - \lambda\}^+ \right] + \sum_{i=1}^n \tau_i \rho_i \right\}.$$

Notice that $\widetilde{v}_i(q_i) \geq \widetilde{\psi}_i(q_i)$ for any $1 \leq i \leq n$ and $q_i \in [0,1]$ by definition. Then for any $\boldsymbol{\tau} \in [0,1]^n$, $\chi^{\mathrm{BDF}}(\boldsymbol{\tau}) \geq \chi^{\mathrm{BRO}}(1^n - \boldsymbol{\tau})$. As a result, $\min_{\boldsymbol{\tau} \in [0,1]^n} \chi^{\mathrm{BDF}}(\boldsymbol{\tau}) \geq \min_{\boldsymbol{\gamma} \in [0,1]^n} \chi^{\mathrm{BRO}}(\boldsymbol{\gamma})$, which finishes the proof.

*eBDPFA $\succeq$ ePFPA.* The theorem follows a duality argument. To start with, as we provide in the proof of Theorem C.1, we have a strong duality result for BDFPA when each buyer's bidding qf is strictly increasing. In other words, given $(\widetilde{v}_i)_{1 \leq i \leq n}$ and $(\rho_i)_{1 \leq i \leq n}$, the seller's revenue in eBDFPA, or $\mathrm{OPT}^{\mathrm{BDF}}$, equals to $\min_{\boldsymbol{\tau} \in [0,1]^n} \chi^{\mathrm{BDF}}(\boldsymbol{\tau})$, with $\chi^{\mathrm{BDF}}(\boldsymbol{\tau})$ defined as:

$$\chi^{\mathrm{BDF}}(\boldsymbol{\tau}) := \mathbb{E}_q \left[ \max_{1 \leq i \leq n} \{(1 - \tau_i)\widetilde{v}_i(q_i) - \lambda\}^+ \right] + \sum_{i=1}^n \tau_i \rho_i.$$

Now, the revenue of any budget-extracting pacing mechanism is no larger than the value of the following programming, which represents the optimal revenue of any feasible (no requirement for budget-extracting) pacing first-price auction. Note that such reasoning does not depend on Theorem A.3, which demands that the bidding qfs are inverse Lipschitz continuous.

$$\max_{\boldsymbol{\beta} \in [0,1]^n} \int_q \max_i \{\beta_i \widetilde{v}_i(q_i) - \lambda\}^+ \, \mathrm{d}q,$$

$$\text{s.t.} \quad \int_0^1 \beta_i \widetilde{v}_i(q_i) \cdot \left( \int_{q_{-i}} \Phi_i(\boldsymbol{\beta}, \widetilde{v}, q) \, \mathrm{d}q_{-i} \right) \mathrm{d}q_i \leq \rho_i, \quad \forall 1 \leq i \leq n.$$

Denote the optimal value of the above programming by $\mathrm{OPT}^{\mathrm{PF}}$. We consider the Lagrangian dual $\chi^{\mathrm{PF}}$ of the above programming, with dual variables $\{\kappa_i\}_{1 \leq i \leq n}$:

$$\chi^{\mathrm{PF}}(\boldsymbol{\kappa}) := \max_{\boldsymbol{\beta} \in [0,1]^n} \sum_{i=1}^n \int_0^1 ((1 - \kappa_i)\beta_i \widetilde{v}_i(q_i) - \lambda) \cdot \left( \int_{q_{-i}} \Phi_i(\boldsymbol{\beta}, \widetilde{v}, q) \, \mathrm{d}q_{-i} \right) \mathrm{d}q_i + \sum_{i=1}^n \kappa_i \rho_i. \tag{30}$$

By weak duality, we have

$$\text{OPT}^{\text{PF}} \leq \min_{\boldsymbol{\kappa} \geq 0} \chi^{\text{PF}}(\boldsymbol{\kappa}) \leq \min_{\boldsymbol{\kappa} \in [0,1]^n} \chi^{\text{PF}}(\boldsymbol{\kappa})$$

$$= \min_{\boldsymbol{\kappa} \in [0,1]^n} \max_{\boldsymbol{\beta} \in [0,1]^n} \sum_{i=1}^{n} \int_0^1 \left( (1 - \kappa_i) \beta_i \widetilde{v}_i(q_i) - \lambda \right) \cdot \left( \int_{\boldsymbol{q}_{-i}} \Phi_i(\boldsymbol{\beta}, \widetilde{\boldsymbol{v}}, \boldsymbol{q}) \, \mathrm{d}\boldsymbol{q}_{-i} \right) \mathrm{d}q_i + \sum_{i=1}^{n} \kappa_i \rho_i$$

$$\leq \min_{\boldsymbol{\kappa} \in [0,1]^n} \max_{\boldsymbol{\beta} \in [0,1]^n} \sum_{i=1}^{n} \int_0^1 \left( (1 - \kappa_i) \widetilde{v}_i(q_i) - \lambda \right) \cdot \left( \int_{\boldsymbol{q}_{-i}} \Phi_i(\boldsymbol{\beta}, \widetilde{\boldsymbol{v}}, \boldsymbol{q}) \, \mathrm{d}\boldsymbol{q}_{-i} \right) \mathrm{d}q_i + \sum_{i=1}^{n} \kappa_i \rho_i$$

$$= \min_{\boldsymbol{\kappa} \in [0,1]^n} \mathbb{E}_{\boldsymbol{q}} \left[ \max_{1 \leq i \leq n} \left\{ (1 - \kappa_i) \widetilde{v}_i(q_i) - \lambda \right\}^+ \right] + \sum_{i=1}^{n} \kappa_i \rho_i$$

$$= \min_{\boldsymbol{\kappa} \in [0,1]^n} \chi^{\text{BDF}}(\boldsymbol{\kappa}) = \text{OPT}^{\text{BDF}}.$$

Here the third line is due to $\beta_i \leq 1$. The fourth line follows a similar argument used when we prove Theorem C.1, specifically (7) and (8). As a result, we have $\text{OPT}^{\text{PF}} \leq \text{OPT}^{\text{BDF}}$. Since $\text{OPT}^{\text{PF}}$ is no less than the revenue of any budget-extracting PFPA, the theorem is proved.

*BROA $\succeq$ eBDSPA, BROA $\succeq$ ePSPA.* At last, for this result, by the terminology in Balseiro et al. [3], we only need to show that eBDSPA is budget-constrained incentive-compatible (BCIC). In fact, as already proved by Balseiro et al. [3], ePSPA is BCIC, and when each buyer's bidding qf is strictly regular, BROA dominates all other BCIC mechanisms.

In fact, we can show that BDSPA is BCIC in general. To see this, we fix some bid-discount multiplier tuple $\boldsymbol{\tau}$ and quantile profile $\boldsymbol{q}$. We consider two cases for any buyer $1 \leq i \leq n$.

- If $\tau_i \widetilde{v}_i(q_i) \geq \max_{i' \geq i} \{ \tau_{i'} \widetilde{v}_{i'}(q_{i'}), \lambda \}$, then $\max_{i' \geq i} \{ \tau_{i'} \widetilde{v}_{i'}(q_{i'}), \lambda \} / \tau_i \leq \widetilde{v}_i(q_i)$. For $i$, as long as she wins, her actual bid does not affect her payment, and her revenue remains unchanged at a non-negative value. If $i$ cuts her bid to lose, then her revenue becomes zero, which is no better than winning the item.
- If $\tau_i \widetilde{v}_i(q_i) < \max_{i' \geq i} \{ \tau_{i'} \widetilde{v}_{i'}(q_{i'}), \lambda \}$, then $\max_{i' \geq i} \{ \tau_{i'} \widetilde{v}_{i'}(q_{i'}), \lambda \} / \tau_i > \widetilde{v}_i(q_i)$. For $i$, as long as she loses, her revenue remains zero. On the other hand, if she raises her bid to win the item, then her payment becomes strictly larger than the value she receives, and $i$ will get a negative revenue, which is worse than losing.

As a result, BDSPA is unconditionally BCIC.

Combining all three parts together, the proof of Theorem A.4 is finished.

