# OpenReview forum: "Budget-Constrained Auctions with Unassured Priors: Strategic Equivalence and Structural Properties"
_ACM.org/TheWebConf/2024/Conference — TheWebConf24 Oral_

### Official Review · Reviewer_3JmV · 2023-11-21

**Novelty:** 6
**Technical Quality:** 6

**Review:**

The authors consider an auction problem inspired by an auto-bidding context where an auctioneer aiming to maximize revenue faces a set of buyers with public ex-ante budget constraints (i.e., expected payments must be less than a budget cap) and private values drawn from independent distributions who aim to maximize their quasi-linear utility subject to the budget constraint.  In contrast to the standard fully Bayesian model, the authors assume that these value distributions are unknown and the auctioneer instead only has access to the distribution of past bids (and, hence, the bidding quantile functions) of the participants.  The goal of this paper is then to investigate the equilibria and strategic properties of various well-studied auction formats in this model - namely, bid-discounting and pacing first and second price auctions as well as the Bayesian revenue-optimal auction for the fully Bayesian setting.

The central results in this paper are that the Bayesian revenue-optimal auction is strategically equivalent to the budget-extracting bid-discount first-price auction (i.e., there is a mapping between strategies in one auction and the other such that outcomes are the same).  Moreover, they demonstrate that the equilibrium outcomes in these two auctions are the same.  They further show strategic equivalence between first and second price auctions when bidders are symmetric.  Finally, they compare the revenue obtained by these auctions when bidders are not strategic and demonstrate the revenue superiority of the bid-discount first-price auction.

This paper has many strengths.  First, it considers a well-studied model of auto-bidding that is of interest to the WebConf community from a new, and seemingly natural, angle of unassured priors.  Second, it provides a very comprehensive set of results comparing many auctions proposed in the literature and in practice.  Third, the proofs are non-trivial and the results offer interesting insights regarding the use of “simpler” mechanisms (e.g., first-price auctions) versus “complex” ones (e.g., the Bayesian revenue-optimal auction) in auto-bidding settings with unassured priors by showing that the simpler mechanisms induce the same outcomes as more complex ones.

On the negative side, this paper is, in a sense, “purely” theoretical so it is not clear whether or not the theoretical insights are born out in practice on real-world data.  Second, while the unassured priors model is interesting and moves in a good direction, in my view, away from fully Bayesian assumptions, it isn’t totally clear that it is the “right” model for practice.  Finally, I think the paper would benefit from a higher level “roadmap” of the results and some intuition for the proofs (which are quite technical).  However, despite these (smaller) drawbacks, I am positive about this paper and think it makes a nice contribution to the literature on auctions in the auto-bidding world.

Smaller comments

Line 269-271: “there exists a maximum budget-feasible parameter…, and is budget-extracting” -> I think “and” should probably be replaced with “and this parameter” in this sentence.

Line 1294: I would suggest writing out what “qf” means here.  You mention it in the body of the paper after the statement of the lemma which this proof is for.

[After rebuttal]  Thank you for your responses.  I remain positive about this paper and would recommend acceptance.

**Questions:**

Can you comment on what would fundamentally change in your modelling and in the results if we required the budget constraint to hold ex-post, rather than in expectation?

**Reviewer Confidence:**

2: The reviewer is willing to defend the evaluation, but it is likely that the reviewer did not understand parts of the paper

**Scope:**

4: The work is relevant to the Web and to the track, and is of broad interest to the community

---

### Official Review · Reviewer_K4dc · 2023-11-23

**Novelty:** 6
**Technical Quality:** 7

**Review:**

The paper studies various budget-constrained parametrized mechanisms in a multiple buyer setting with unassured priors. The paper provides theoretical results showing strategic equivalence and revenue-dominance relationships among budget-constrained parametrized mechanisms.

Strengths:

The paper’s model is motivated by the fact that platforms do not know the prior value distributions or actual values of advertisers. In practice, platforms observe only advertisers’ bids and may not know their values. This setting is different from existing works on budget control methods.

The paper’s strategic equivalence results provide a justification for major platforms’ recent transition to first-price auctions. More specifically, they show a strategic equivalence between the Bayesian revenue-optimal mechanism and the budget-extracting bid-discount first-price mechanism in the unassured prior setting.

**Questions:**

What is a utility-revenue profile? Is this a pair of realized utility and revenue? In Definition 4.1, are the utility-revenue profile and revenue-utility profile the same thing?

**Reviewer Confidence:**

3: The reviewer is confident but not certain that the evaluation is correct

**Scope:**

4: The work is relevant to the Web and to the track, and is of broad interest to the community

---

### Official Review · Reviewer_EYLe · 2023-11-26

**Novelty:** 6
**Technical Quality:** 5

**Review:**

Summary:

This paper studies 5 different mechanism designs for budget constrained bidders: Bid-discount/Pacing x SPA/FPA, and Bayesian revenue-optimal auction, when the prior on the bidders value distributions is unknown.

Under mild assumptions, the authors show that

(1) budget-extracting bid-discount first-price auction is strongly strategic-equivalent to the Bayesian revenue-optimal auction.
(2) in the symmetric case, first-price and second-price auctions are weakly strategic-equivalent.
(3) without strategic bidding, bid-discount first- price auction dominates Bayesian revenue-optimal auction and pacing first-price auction, while Bayesian revenue-optimal auction outperforms two variants of second-price auctions.

Comments:

This paper is generally well-written and clear. The problem studied in this paper is interesting and relevant to auction design in online advertising. The collection of results is non-trivial and technically strong. The strong strategic equivalence between bid-discount first-price auction and Bayesian revenue-optimal auction is particularly interesting and demonstrates interesting insight that may be applicable for practice. As most proofs are deferred to appendix, the reviewer didn't check the details of the proofs but they look plausible in the hindsight.

**Questions:**

1. The reviewer wonders what happens if the bidders can manipulate their budgets as well? Do the strategic equivalence results continue to hold?

**Reviewer Confidence:**

3: The reviewer is confident but not certain that the evaluation is correct

**Scope:**

3: The work is somewhat relevant to the Web and to the track, and is of narrow interest to a sub-community

---

### Official Review · Reviewer_JHvH · 2023-11-29

**Novelty:** 6
**Technical Quality:** 6

**Review:**

Summary

This paper introduces a notion of “strategic equivalence” between classes of auctions for budget-constrained bidders with unknown priors, and characterizes when common classes of auctions are equivalent.

More specifically, the authors consider an auction setting where bidders know their own priors but the auctioneer does not. At the beginning of the game, the bidders can report (perhaps untruthfully) their priors to the auctioneer. The auctioneer then chooses a mechanism from a parameterized class of mechanisms (e.g. a FPA where bidder i’s bid is weighted by alpha_i) and runs an auction amongst all the bidders. Instead of trying to solve the resulting game, the paper asks which parameterized classes give rise to the same underlying game between bidders and auctioneer. For example, is there a simple reduction from the case where a bidder chooses to run a FPA (with arbitrary bid weights) vs a SPA (with arbitrary bid weights)?

The authors consider 5 different classes of auction: discounted and pacing versions of both FPAs and SPAs, and the Bayesian revenue-optimal auction, each parametrized by weighting the bid of bidder i (or in the revenue-optimal case, the virtual bid) by a scaling parameter. The authors also introduce two notions of equivalence to capture this relationship: weak equivalence, which can transform the parameters arbitrarily, and strong equivalence, which must act independently on each bidder’s parameter.

The main result is that (under some Lipschitz continuity / budget consumption assumptions) the class of  Bayesian-revenue optimal auctions is strongly equivalent to the class of bid-discounted FPAs. Furthermore, if all bidders are symmetric, it is possible to show that all these classes of auctions are weakly equivalent.

Evaluation

The growth of adoption of autobidding (with budget and target constraints) has made the classic auction design problem increasingly complex. This is an interesting result in auction theory as it (in some sense) simplifies the space of possible auctions to run, in a similar way that e.g. the revelation principle and Myerson’s lemma simplify the space of possible auctions in the classic prior-aware setting. With the caveat that I have not deeply thought about these questions before, I find it pretty interesting that the somewhat complex class of Bayesian revenue-optimal auctions (with virtual-value-reweighting) is equivalent to the class of first-price auctions (with bid discounts). I support this paper for acceptance -- I think it will be of interest to the more theoretical mechanism design / algorithmic economics crowd attending WebConf.

The paper was generally well-written and easy to read (although it is definitely a little tricky to parse the results when they first appear amidst the sea of acronyms).

**Questions:**

Feel free to respond to any comments / potential misunderstandings in the above review.

One thing which was not too clear to me from reading the paper (as someone who is not very familiar with the prior work) -- are these (or similar) strategic equivalence results already known in the setting where bidders are not budget-constrained, or is this notion of strategic equivalence entirely novel (even in that setting)? If it is, do any of the equivalence results hold in the budget-unconstrained setting? (I know some of the results require the parametrized auction to consume the entire budget -- perhaps this prevents some results from clearly extending).

**Reviewer Confidence:**

2: The reviewer is willing to defend the evaluation, but it is likely that the reviewer did not understand parts of the paper

**Scope:**

3: The work is somewhat relevant to the Web and to the track, and is of narrow interest to a sub-community

---

### Decision · Program_Chairs · 2024-01-22

**Decision:**

Accept (Oral)

**Comment:**

The review team uniformly appreciated the fundamental nature of the results presented in the paper. The question of revenue- and strategic-equivalence between auctions with unknown priors is a very interesting one, and has recently seen important results, but without budget constraints. This paper extends those results to a setting with budget constraints, and also generalizes by considering many different auction types. The writing is clear, and the results will definitely be of interest to the AGT community researchers and practitioners alike.

 A couple more references on the budget-pacing literature:

 1) The Best of Many Worlds: Dual Mirror Descent for Online Allocation Problems; Santiago Balseiro, Haihao Lu and Vahab Mirrokni
 2) Analysis of Dual-Based PID Controllers through Convolutional Mirror Descent; Santiago Balseiro, Haihai Lu, Vahab Mirrokni and Balasubramanian Sivan